# A Hydrodynamic Theory for Non-Equilibrium Full Counting Statistics in One-Dimensional Quantum Systems

Dávid X. Horváth[1][*], Benjamin Doyon[1] and Paola Ruggiero[1]

**1** Department of Mathematics, King's College London, Strand WC2R 2LS, London, U.K.
[*] david.horvath@kcl.ac.uk

July 30, 2025

## Abstract

We study the dynamics of charge fluctuations after homogeneous quantum quenches in one-dimensional systems with ballistic transport. For short but macroscopic times where the non-trivial dynamics is largely dominated by long-range correlations, a simple expression for the associated full counting statistics can be obtained by hydrodynamic arguments. This formula links the non-equilibrium charge fluctuation after the quench to the fluctuations of the associated current after a charge-biased inhomogeneous modification of the original quench which corresponds to the paradigmatic partitioning protocol. Under certain assumptions, the fluctuations in the latter case can be expressed by explicit closed form formulas in terms of thermodynamic and hydrodynamic quantities via the Ballistic Fluctuation Theory. In this work, we identify precise physical conditions for the applicability of a fully hydrodynamic theory, and provide a detailed analysis explicitly demonstrating how such conditions are met and how this leads to such hydrodynamic treatment. We discuss these conditions at length in non-relativistic free fermions, where calculations become feasible and allow for cross-checks against exact results. In physically relevant cases, strong long-range correlations can complicate the hydrodynamic picture, but our formula still correctly reproduces the first cumulants.

# 1   Introduction

In a generic quantum mechanical state, most relevant observables typically do not have a particular value but are rather described by a probability distribution specific to the state and the observable. The problem of understanding the corresponding fluctuations in many-body systems has therefore been a central problem in quantum mechanics. This topic has recently regained a large amount of attention, with a significant drive being the advent of experimental platforms that are capable of simulating nearly isolated interacting many-body systems and fluctuations therein [1, 2]. Particular focus was given to charges that are conserved quantities for the full system; when restricted to a subsystem they exhibit nontrivial fluctuations and dynamics. This has been studied especially in low dimensional systems [3–24].

When equilibrium scenarios in extended systems are regarded, fluctuations on large regions are entirely determined by thermodynamic functions. For instance, in an infinitely large system of particles at finite density, the cumulants $C_n(Q) = \langle Q^n \rangle^c$ of an extensive conserved quantity $Q$ (such the energy, particle number, electric charge, etc) when supported on a region of space of finite but large volume $V$ are encoded in the generating function $F(\lambda, \beta)$ consisting of the difference of Landau free energies $V f(\beta)$: $F(\lambda, \beta) = \sum_{n=1}^{\infty} \lambda^n C_n / n! = V(f(\beta) - f(\beta + \lambda))$, where $\beta$ is the generalised temperature, or Lagrange multiplier, associated with the conserved charge within the equilibrium ensemble, and $\lambda$ is referred to as the "counting field". The generating function $F(\lambda, \bar{\mu})$ is often called the "full counting statistics" (FCS), whose Legendre-Frenchel transform is known to give the probability distribution. This is true, more generally, in *steady-states*, including non-equilibrium steady-states (NESS), as long as they exhibit short-range correlations. Importantly in this setting a large-deviation principle holds, both in quantum and classical systems.

By contrast, characterising the fluctuations of dynamical quantities, such as the total amount of charge going through an interface in a finite but large time $T$, is not possible merely by thermodynamics. Nevertheless, in some cases, obtaining the large-deviation form of such fluctuations only requires the knowledge of the hydrodynamic theory associated with the many-body system. In particular, if the charge admits ballistic transport, Euler hydrodynamics via ballistic fluctuation theory (BFT) [25–29], or the more general ballistic macroscopic fluctuation theory (BMFT) [30, 31] can be used; while if transport is diffusive, one can rely on macroscopic fluctuation theory (MFT) [32, 33]. It is worthwhile to note that in non-stationary but slowly-varying states, where local entropy maximisation has already occured, hydrodynamic principles still hold, and BMFT or MFT apply [34]. However there are also certain situations when fluctuations are anomalous and the large-deviation principle is broken [35–40].

When genuinely out-of-equilibrium cases are regarded, much less is known and even the basic physical principles are poorly understood. A common and important non-equilibrium protocol is that of *quenches*, that is, sudden changes of coupling constants or other parameters determining the dynamics. It is known that in most systems entropy maximisation constrained by the extensive charges, i.e. (generalised) thermalisation, occurs at long times [41–48]. Although, in correspondence with the above, averages of simple, local or few body operators can therefore be obtained by thermodynamic (Gibbs or generalised Gibbs) ensembles, the characterisation of fluctuations (or correlation in general) is highly non-trivial, both technically and conceptually. For instance, when considering the problem of how conserved charges restricted to a large volume $X$ fluctuate after time $T$, difficulties are faced. First, the long-time steady-state only describes the regime $T \gg X$ where (generalised)-thermalization has occurred, but not the large-scale fluctuations for $T \propto X$ or $T \ll X$, because it misses long-range correlations that are known to emerge after the quench. Second, simple hydrodynamic fluctuation arguments based on linear response are not applicable far from equilibrium. Therefore, the main challenge is, on the one hand, to identify how much information of the quench should be kept and, on the other hand, incorporating this information in a theory of non-equilibrium fluctuations in a feasible way.

In this paper we reconsider the hydrodynamic picture introduced in a companion paper [24], for large-scale fluctuations after quenches, in one dimensional models with ballistic transport. The theory, in principle, is able to characterise the fluctuations of ballistically propagating Noether-type conserved charges in a large subsystem at ballistic but short times $T \ll X$, thus describing the initial time evolution of the associated FCS at such time scales. A cornerstone of the hydrodynamic picture is that it can take care of the strongest long-range correlations arising from quenches by exploiting the continuity equation. Crucially, this amounts to recognising that in the regime of short but macroscopic times, the dynamical fluctuation of the charge $\int_0^X dx \, q(x,T)$ specified by the generating function $\langle \Psi | \exp\left( \lambda \int_0^X dx \, q(x,T) \right) | \Psi \rangle$ after quenching from $| \Psi \rangle$ can be characterised by the FCS of the associated current $\int_0^T dt \, j(0,t)$ in the biased initial state $\exp\left( \frac{\lambda}{2} \int_0^\infty dx \, q(x,0) \right) | \Psi \rangle$ (upon normalising it). This corresponds to a version of the paradigmatic "partitioning protocol", hence the theory establishes an unexpected connection between the charge fluctuations after a homogeneous quench and the associated current fluctuation after a particular (inhomogeneous) partitioning protocol (cf. eq.(8) below). Further, under conditions of temporal clustering, the time-dependence of the FCS is determined in terms of the long-time non-equilibrium steady-state emerging from the aforementioned partioning protocol (see Fig. 1 and Sec. 2 (especially eq.(9)) for details). In this case, to characterise the current fluctuations in the non-equilibrium steady-state, the theory utilises the BFT flow equations, and thus is expected to apply to general interacting many-body systems with ballistically propagating conserved quantities. For certain (but not all) integrable models and exactly solvable quenches, time clustering indeed holds and the formula specialises to a recent conjecture from space-time swap techniques [20, 21]. For technical reasons, our results are shown for the particle number FCS, which is "ultra-local", but are expected to be valid for generic local charge.

Importantly, limitations of this hydrodynamic picture were also highlighted in [24]. In particular, while spatial long-range correlations are believed to be taken care of by using the continuity equations and by reformulating the problem of charge fluctuations in terms of the current fluctuations, the BFT framework only provides a legitimate way of computing the latter if no dynamical long-range correlations, arising from the bipartite inhomogeneous quench, are present. Accordingly, this requirement is one the main criteria for the applicability for the hydrodynamic picture, and we stress again that in realistic models, is not always satisfied [24]. Although this means that the simple hydrodynamic picture needs further refinements to be able to deal with a broader class of physical situations, in this work we do not initiate such an enterprise. Instead we, thoroughly discuss the emergence of this picture and discuss physical situations when it is readily applicable. Additionally, it is worth emphasising that corrections to the naive predictions can be "small" even when long-range correlation are present: for the particular problem studied in [24] it was found that first 5 cumulants of the charge fluctuations are unaffected by such corrections.

In this work, we review this hydrodynamic framework, provide precise conditions under which it is applicable (the viewpoint taken here is slightly modified wrt. that in [24], see the discussion about the characterisation of the non-equilibrium steady-state there), and demonstrate in details that they are met in a non-relativistic free fermion model. The aim is to give a thorough understanding of the emergence of this simple picture demonstrating the main guiding principles. Computations in free fermion models also allow for indirect cross-checks thanks to known exact results for their FCS [22].

The structure of the paper is as follows. In Section 2 we review the hydrodynamic theory and list three main physical conditions the system must satisfy in order for the hydrodynamic treatment to apply. The exposition and the discussion of the conditions are followed by the derivation of the theory using thermo- and hydrodynamic arguments and exploiting the three conditions. The derivation is reasonably compact, however, two more technical steps (cf. Factorisation Property 1 and 2) are relegated to Section 4. Additionally, we comment on the characterisation of the NESS:

this task is rather non-trivial but considerable simplifications are expected to occur which we shall discuss.

Section 3 is dedicated to free non-relativistic fermions and a specific family of solvable quench problems, and there are numerous reasons for their discussion. First, the demonstration of the two Factorisation Properties for theories with an infinite Lieb-Robinson velocity can been carried out for free theories. Second, for this specific class of quenches in free-fermion theories, exact analytic results are known for the time-dependent FCS [21], and these results offer an immediate benchmark for the hydrodynamic framework. In addition, this setting makes it possible to transparently check the fulfillment of the main physical conditions, but also to study the emergence and the properties of the NESS after the inhomogeneous charged-biased initial state. This latter study is particularly insightful, as it allows us to conjecture a generic and conceptually simple way of charaterising the NESS. After introducing the class of integrable quenches, the main results concerning the NESS are summarised in this section (detailed calculations are instead presented in Sec. 5, Sec. 6 and in some appendices, as explained there), and the non-equilibrium FCS is expressed explicitly in terms of the solution of the BFT Flow Equations.

Sections 2 and 3 are self-contained and demonstrate the key findings of the paper with little technicalities. The rest of the paper is instead more technical, and although important details are presented therein, the following sections are not necessary for obtaining a clear understanding of the main ideas behind the hydrodynamic theory and its applicability.

In Section 4, the aforementioned Factorisation Properties are discussed. We argue in this section that these properties are indeed present if the initial state is a strongly clustering one. We demonstrate this for the first property in complete generality (no restriction to free fermions is needed here) for systems with a finite Lieb-Robinson velocity. For the case of an infnite Lieb-Robinson velocity, we only deal with non-relativistic free fermions and apply a suitable series expansion wrt. the counting field $\lambda$. We demonstrate the presence of the second property for clustering initial states as well, but in this case, we focus only on free fermions. We perform a series expansion up to the 3rd order in the counting field and we argue for the behaviour of higher orders. In this section, we use the explicit case of particle number FCS, however we argue how our results are expected to be valid more generally.

In Section 5 we explicitly demonstrate via microscopic computations and by focusing on the particle current density operator that, for solvable quenches and non-relativistic free fermions, a NESS is indeed reached after the bipartite quench (the same is shown for generic ultra-local operators and for a representative class of quasi-local operators as well in Appendix D). Here we also show that the NESS emerging from the microscopic inhomogeneous initial state equals the NESS that develops after two maximal entropy states restricted to half-infinite subsystems are joint together. These maximal entropy states are the steady-state of the original quench problem, and the homogeneous steady-state that develops after the biased initial state, where now the charge-bias is made in a homogeneous way. We believe that the aforementioned equality of the two NESS-s is generally valid, although we can demonstrate it explicitly for free fermions. The computations in the section are carried out via a non-trivial systematic series expansion wrt. the bias in the inhomogeneous initial state, focusing on the first few orders. Finally, for solvable quenches it is possible to describe the maximal entropy state emerging after the homogeneously biased initial states. While this result is already present in [22], and was derived in [24] for generic interacting integrable quenches, here a microscopic derivation in the free fermion case is provided (see Appendix C).

Section 6 focuses on long-range temporal correlations of the current density. Such correlations are proved to be sufficiently weak after quenching from the charge-biased inhomogeneous state in free fermion systems if the original initial state is solvable. This check is again carried out by explicit microscopic computations applying the same series expansion wrt. the bias in the inhomogeneous initial state. In addition, we also demonstrate how the cumulants describing the charge fluctuation

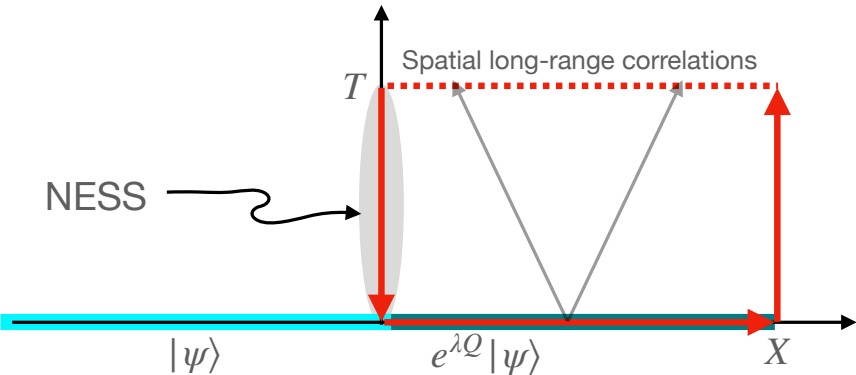

Figure 1: Illustration of the universal hydrodynamic principles for full counting statistics of the total charge $Q$ on length $X$ at time $T$ after a quench. The dotted red line is where the generator $F = \ln\langle\psi|e^{\lambda Q|_0^X(T)}|\psi\rangle$ is evaluated in space-time. The full red line represents the deformed path obtained by the local continuity equation. Turquoise bars represent the biased initial state (on the right locally $e^{\lambda Q}|\psi\rangle$ and on the left locally $|\psi\rangle$). The inhomogeneous state give rise to a partitioning protocol, in whose non-equilibrium steady-state (NESS) the current full counting statistics is evaluated to obtain the linear growth of $F$ with $T$.

converge to their values in the NESS, which is in accordance with the convergence of local and quasi-local operators to the NESS.

Finally we conclude in Section 7. Besides summarising our main findings, we comment on further important quench problems which fall inside the range of applicability of the presented hydrodynamic treatment, discuss a possible way go beyond that, and list a few particularly relevant open problems.

## 2   Main Result

### 2.1   Full counting statistics after a quench in the ballistic regime

The quantity of main interest of this paper is the initial growth of the FCS of a conserved charge after homogeneous quantum quenches in closed quantum systems undergoing Hamiltonian time-evolution. Our focus is on the large-scale fluctuations of a conserved charge lying on the region $[0, X] \subset \mathbb{R}$ a time $T$ after a quench, $\int_0^X dx\, q(x, T)$, where $q(x, t)$ is a parity-even and local density, which obeys the continuity equation

$$\partial_t q(x, t) + \partial_x j(x, t) = 0, \tag{1}$$

with some current $j$. Here we use the Heisenberg picture, i.e., $O(x, t) = e^{itH}O(x, 0)e^{-itH}$ where $H$ is the Hamiltonian of the system. Both subsystem size $X$ and time $T$ are large compared to microscopic and sub-ballistic scales: they are both on the Euler scale associated with ballistic propagation. But we consider "intermediate times", such that

$$X \gg T \gg 1. \tag{2}$$

The dynamics which results under this scaling is subject to long-time, emergent effects. The quench is specified by a parity and translation invariant initial state $|\Psi\rangle$ that is not an eigenstate or a simple superposition of a few energy-eigenstates of the Hamiltonian of the isolated many-body system – we provide more precise requirements on $|\Psi\rangle$ below.

For simplicity we shall concentrate on the charge $Q$ being the particle number,

$$Q = \int \mathrm{d}x\, \rho(x)$$

with $\rho$ the density of particles. However, most of the steps go through for any local and extensive charge (not to be confused with the extensivity of its fluctuation in the initial state, discussed below) and the result is argued to hold in general.

For the dynamics in the region (2), it turns out that the fluctuations of the conserved charge in the initial state $|\Psi\rangle$ itself play an important role. In particular, they can be extensive (that is, with cumulants growing linearly with the size $X$ of the region on which the charge is evaluated) or sub-extensive. We shall mostly deal with the extensive case throughout this paper, but comment on certain sub-extensive situations in our conclusions.

Results for the aforementioned problem have first been obtained in [20, 21] for free fermion and integrable systems, where the authors focused on initial states with non-extensive [20] and extensive [21] fluctuations for specific charges, using a framework called space-time duality [49]. Let us review the main structure of these results, which was clarified in [24].

Based on [20, 21, 24], the logarithm of the full counting statistics,

$$F(\lambda, X, T) = \ln\langle\Psi|e^{\lambda \int_0^X \mathrm{d}x\, q(x,T)}|\Psi\rangle \tag{3}$$

can be written as

$$F(\lambda, X, T) = \big(X f_\Psi(\lambda) + o(X)\big) + \big(T f_{\mathrm{dyn}}(\lambda) + o(T)\big) + o(X^0)_T \tag{4}$$

where we emphasize that the first parenthesis on the right-hand side is independent of $T$ and that the second parenthesis is independent of $X$. Here, $o(X^0)_T$ means terms that may depend on both $X$ and $T$, but that decay as $X \to \infty$ for $T$ fixed. Below we use the notations

$$Q|_0^X(T) = \int_0^X \mathrm{d}x\, q(x, T) \quad \text{and} \quad J_X|_0^T = \int_0^T \mathrm{d}t\, j(X, t). \tag{5}$$

In (4), $f_\Psi(\lambda)$ is the scaled cumulant generating function (SCGF) of the charge fluctuation in the initial state defined as

$$f_\Psi(\lambda) = \lim_{X\to\infty} \frac{1}{X}\ln\langle\Psi|e^{\lambda Q|_0^X(0)}|\Psi\rangle, \tag{6}$$

and $f_{\mathrm{dyn}}(\lambda)$ accounts for the dynamics of these fluctuations due to time evolution defined as

$$f_{\mathrm{dyn}}(\lambda) = \lim_{T\to\infty} \frac{1}{T} \lim_{X\to\infty} \ln \frac{\langle\Psi|e^{\lambda Q|_0^X(T)}|\Psi\rangle}{\langle\Psi|e^{\lambda Q|_0^X(0)}|\Psi\rangle}. \tag{7}$$

Recalling (2), we interpret (4) as follows. After sending the subsystem size to infinity, for any finite and large $T$, $\lim_{X\to\infty}(F(\lambda, X, T) - F(\lambda, X, 0))$ has a leading linear-in-$T$ growth with the corresponding coefficient $f_{\mathrm{dyn}}(\lambda)$. Loosely speaking this also means that for very large and fixed subsystem sizes the dominant change in $F(\lambda, X, T)$ is linear in $T$ as long as $T$ remains much smaller than $X$. In addition, for any finite and fixed $T \gg 1$, $F(\lambda, X, T)/X$ is finite and tends to $f_\Psi(\lambda)$ as $X$ is increased.

## 2.2 Main result

Our main result is the characterisation of the dynamics of fluctuations after a quench, encoded by $f_{\mathrm{dyn}}(\lambda)$ in (4).

**Formula for a hydrodynamic-type description**

*We show that, under a certain weak condition of spatial clustering, the following identity holds:*

$$f_{\mathrm{dyn}}(\lambda) = \lim_{T\to\infty} \frac{2}{T} \ln \mathrm{Tr}\left[ e^{\lambda J_0|_0^T} \rho_{\mathrm{in}}(\lambda) \right], \quad \text{with} \quad \rho_{\mathrm{in}}(\lambda) = \frac{e^{\lambda/2 Q|_0^\infty} |\Psi\rangle \langle\Psi| e^{\lambda/2 Q|_0^\infty}}{\langle\Psi| e^{\lambda Q|_0^\infty} |\Psi\rangle}. \tag{8}$$

*Further, we show, that under an additional condition of time-like clustering, this can be re-written in terms of a special non-equilibrium steady state (NESS) coming from a partitioning protocol as follows*

$$f_{\mathrm{dyn}}(\lambda) = \lim_{T\to\infty} \frac{2}{T} \ln \mathrm{Tr}\left[ e^{\lambda J_0|_0^T} \rho_{\mathrm{NESS}}(\lambda) \right]. \tag{9}$$

*We specify the conditions precisely below.*

Here $\rho_{\mathrm{NESS}}$ is the non-equilibrium steady-state that emerges after the "bipartite quench" from the initial state $e^{\lambda/2 Q|_0^\infty(0)} |\Psi\rangle$. It is defined as the maximal entropy state ((generalised) Gibbs ensemble, GGE) obtained by the limit

$$\lim_{t\to\infty} \mathrm{Tr}\left[ O(0,t) \rho_{\mathrm{in}}(\lambda) \right] = \mathrm{Tr}\left[ O(0,0) \rho_{\mathrm{NESS}}(\lambda) \right] \tag{10}$$

for every local operator $O$. The temporal clustering condition below guarantees that (19) holds. Additionally, it implies clustering of currents in the non-equilibrium steady-state,

$$|t - t'| \, \mathrm{Tr}\left( \rho_{\mathrm{NESS}}(\lambda) j(0,t) j(0,t') \right)^c \to 0, \quad \text{as} \quad |t - t'| \to \infty, \tag{11}$$

and similarly for higher-point functions. The latter is guaranteed if there are no zero-velocity modes coupling to the currents, see the discussion in [50]; and a counter example in [36]. Here and below, the superscript $c$ denotes connected correlation functions,

$$\mathrm{Tr}\left( \rho_{\mathrm{NESS}}(\lambda) j(0,t) j(0,t') \right)^c =$$
$$\mathrm{Tr}\left( \rho_{\mathrm{NESS}}(\lambda) j(0,t) j(0,t') \right) - \mathrm{Tr}\left( \rho_{\mathrm{NESS}}(\lambda) j(0,t) \right) \mathrm{Tr}\left( \rho_{\mathrm{NESS}}(\lambda) j(0,t') \right). \tag{12}$$

Crucially, Eq. (11) guarantees that the right-hand side of (9) can be evaluated by hydrodynamic principles via the Ballistic Fluctuation Theory (BFT) [25–29]. This therefore *provides a simple and general hydrodynamic-type theory for the full counting statistics after a quench $F(\lambda, X, T)$ in the regime* (2). We will show (9) in Subsection 2.3 order by order in $\lambda$: the FCS $f_{\mathrm{dyn}}(\lambda)$, Eq. (7), is seen as a generating function and $\lambda$ as a generating parameter, and the right-hand side of (9) is also to be expanded in $\lambda$.

Formula (9) is inspired by results of Refs. [21, 24]. However, the characterisation of $\rho_{\mathrm{NESS}}(\lambda)$ given here in terms of a bipartite quench is more accurate. We will explain below how one can argue that $\rho_{\mathrm{NESS}}(\lambda)$ is in fact a steady-state for a bipartitioning protocol of a more standard form coming from two maximal entropy states, as is obtained in [21] in a particular family of integrable models, and argued more generally in [24].

**Conditions**

An important part of our result is to determine precise conditions under which (8) and (9) hold. Eq. (8) requires only the first condition, while Eq. (9) holds only if all conditions are met. *The three conditions are as follows.*

  i) (even, homogeneous, clustering initial state). We require parity and translation invariance, and sufficiently strong clustering of the initial state. That is, connected correlation functions behave as

$$\langle\Psi| O(x) O'(y) |\Psi\rangle^c \to 0 \quad \text{fast enough as} \quad |x - y| \to \infty, \tag{13}$$

for every local operators $O$, $O'$ (and similarly for higher-order correlation functions). As above, the superscript $c$ denotes connected correlation functions,

$$\langle\Psi|O(x)O'(y)|\Psi\rangle^c = \langle\Psi|O(x)O'(y)|\Psi\rangle - \langle\Psi|O(x)|\Psi\rangle\langle\Psi|O'(y)|\Psi\rangle \tag{14}$$

We do not specify "fast enough", but exponential decay, or a power law decay with large enough exponents, is expected to work. This condition is sufficient for (8).

ii) (local relaxation). We require relaxation to a NESS $\rho_{\text{NESS}}(\lambda)$ (normalised) at $x = 0$ after a bipartite quench with initial state

$$|\tilde\Psi\rangle_\lambda = \frac{e^{\lambda/2Q|_0^\infty(0)}|\Psi\rangle}{\sqrt{\langle\Psi|e^{\lambda Q|_0^\infty(0)}|\Psi\rangle}}, \tag{15}$$

that is

$$\lim_{t\to\infty} {}_\lambda\langle\tilde\Psi|O(0,t)|\tilde\Psi\rangle_\lambda = \text{Tr}\left[\rho_{\text{NESS}}(\lambda)O(0,0)\right], \tag{16}$$

for every local or quasi-local operator $O$, where by "quasi-local" we mean in general products of local operators, which may be evolved in finite times.

iii) (temporal clustering). We require temporal clustering of current correlations after the bipartite quench above. That is, connected correlations for the current operators at different times decay as

$$|t - t'|\,{}_\lambda\langle\tilde\Psi|j(0,t)j(0,t')|\tilde\Psi\rangle^c_\lambda \to 0, \quad \text{as} \quad |t - t'| \to \infty, \tag{17}$$

uniformly for $t, t' > 0$ (and similarly for higher-order correlation functions). For technical reasons, we also assume that $|{}_\lambda\langle\tilde\Psi|j(0,t)j(0,t')|\tilde\Psi\rangle^c_\lambda|$ is uniformly bounded on $t, t' > 0$.

The discussion of whether and when these conditions are met is presented in the sections that follow, mostly concentrating on two-point clustering[1] and on the leading orders in $\lambda$. For this purpose, we will restrict our focus to the case of free non-relativistic fermions, but we discuss in Sec. 7 how the applicability of the hydrodynamic picture can extend to interacting systems (see the companion paper [24]).

Additionally, for technical reasons in some parts of the calculations we will use the assumption that a Lieb-Robinson (LR) bound [52] holds, with finite LR velocity $v_{\text{LR}} < \infty$. Note that as part of a Lieb-Robinson bound, the algebra of observables is typically assumed to have a finite norm $||\cdot||$; this implies the technical uniform bound in condition iii): $|{}_\lambda\langle\tilde\Psi|j(0,t)j(0,t')|\tilde\Psi\rangle^c_\lambda| \le ||j||$. This holds for spin chains and fermionic chains, but does not hold in many important situations, such as non-relativistic fermions on the continuum. Therefore we would like to regard the requirement of a Lieb-Robinson bound as an auxiliary condition facilitating a more transparent presentation of parts of the derivation of the results, but to be eliminated eventually, and possibly replaced by a weaker condition to be determined. We provide partial calculations in this direction in Sec. 4.1.2 (see also Sec. 7).

Our derivation of the main results (8), (9) using the above conditions is presented in Subsection 2.3 is general. This is so except for one step, which relates to an "Asymptotic Commutativity" phenomenon, that we discuss below. Asymptotic Commutativity is intuitively natural, but as far as we know, there is no general rigorous or semi-rigorous statements in the literature concerning this property – we provide supporting calculations in the context of free fermions in the sections that follow.

Let us comment on conditions i) - iii) and their physical meaning.

---

[1]We note that as per the results of [51], two-point clustering is sufficient in order to establish certain forms of higher-point clustering. However we have not shown that these forms of higher-point clustering are sufficient for our results, which is beyond the scope of this paper.

**Condition i): even, homogeneous, clustering initial state**

This condition is expected to hold in many physically relevant quench initial states. In particular, exponential clustering is known to occur in ground states of gapped Hamiltonians. The requirement of parity is for simplicity. Without parity, we have instead

$$f_{\text{dyn}}(\lambda) = \lim_{T \to \infty} \frac{1}{T} \left( \ln \text{Tr} \left[ e^{\lambda J_0|_0^T} \rho_{\text{NESS}}^+(\lambda) \right] + \ln \text{Tr} \left[ e^{-\lambda J_0|_0^T} \rho_{\text{NESS}}^-(\lambda) \right] \right), \quad (18)$$

where

$$
\begin{aligned}
\lim_{t \to \infty} \frac{\langle \Psi | e^{\lambda/2 Q|_0^\infty(0)} O(0,t) e^{\lambda/2 Q|_0^\infty(0)} | \Psi \rangle}{\langle \Psi | e^{\lambda Q|_0^\infty(0)} | \Psi \rangle} &= \text{Tr} \left[ O(0,0) \rho_{\text{NESS}}^+(\lambda) \right] \\
\lim_{t \to \infty} \frac{\langle \Psi | e^{\lambda/2 Q|_{-\infty}^0(0)} O(0,t) e^{\lambda/2 Q|_{-\infty}^0(0)} | \Psi \rangle}{\langle \Psi | e^{\lambda Q|_{-\infty}^0(0)} | \Psi \rangle} &= \text{Tr} \left[ O(0,0) \rho_{\text{NESS}}^-(\lambda) \right]
\end{aligned}
\quad (19)
$$

for every local operator $O$. In translation and parity invariant states and for charges which are even under parity (so that the associated currents are odd), these two contributions are equal (see [24] and below).

**Condition ii): local relaxation**

Condition ii) is also relatively weak, as local relaxation to some NESS is generically expected to occur. The characterisation of the NESS itself is a rather non-trivial task, but this is not necessary for the result. We nevertheless articulate an intuition regarding this problem, which we shall explicitly verify for a certain class of quenches.

Recall that the initial states $e^{\lambda/2 Q|_0^\infty(0)} | \Psi \rangle$ and $e^{\lambda/2 Q|_{-\infty}^0(0)} | \Psi \rangle$ are expected to uniquely define the NESS $\rho_{\text{NESS}}(\lambda) = \rho_{\text{NESS}}^\pm(\lambda)$ via (19). But in fact, it is natural to assume that the dynamics from the initial state $e^{\lambda/2 Q|_0^\infty(0)} | \Psi \rangle$ is such that the left and the right halves of the system at spacial points $|x| \gg \tau$ satisfy

$$\text{Tr} \left[ O(-x, \tau) \rho_{\text{L}} \right] \approx \text{Tr} \left[ O(0,0) \rho \right] \quad \text{and} \quad \text{Tr} \left[ O(x, \tau) \rho_{\text{R}} \right] \approx \text{Tr} \left[ O(0,0) \rho^{(\lambda)} \right] \quad (20)$$

after some microscopic time $\tau$, where $\rho$ and $\rho^{(\lambda)}$ are the maximal entropy or (generalised) Gibbs ensembles emerging after the homogeneous quenches from the initial states $|\Psi\rangle$ (i.e., the original quench problem) and $e^{\lambda/2 Q} |\Psi\rangle$, respectively. Then, the leading dynamical behaviour of local observables in space-time regions $x \approx 0$ and $t \gg \tau$ in the problem is captured by the NESS $\rho_{\text{bpp}}(\lambda)$ of the bipartite quench protocol built up from left and right maximal entropy states (Gibbs or Generalised Gibbs Ensembles) $\rho$ and $\rho^{(\lambda)}$, which is a paradigmatic problem in studying transport properties and non-equilibrium physics. That is, we expect that

$$\text{Tr}[O(0,0) \rho_{\text{NESS}}(\lambda)] = \text{Tr}[O(0,0) \rho_{\text{bpp}}(\lambda)], \quad (21)$$

with

$$\lim_{t \to \infty} \text{Tr} \left[ O(0,t) \left( (\rho)^L \otimes (\rho^{(\lambda)})^R \right) \right] = \text{Tr} \left[ O(0,0) \rho_{\text{bpp}}(\lambda) \right], \quad (22)$$

for every local operators $O$, where $(\rho)^{L/R}$ means a normalised density matrix that is restricted on the left/right half of the system. In other words, we expect that $\rho_{\text{NESS}}(\lambda) = \rho_{\text{bpp}}(\lambda)$, where the two NESS's originate from, strictly speaking, two different quench protocols, cf, Eqs. (19) and (22). We will explicitly verify this intuition for integrable quenches in free fermion systems and we note that (21) follows from the results of [21] also for the Rule 54 model and certain initial states.

**Condition iii): temporal clustering**

Whereas the fulfillment of i) - ii) is naturally expected, condition iii), by contrast, is a rather non-trivial requirement, which is not necessarily fulfilled (except for the technical uniform bound on $|_\lambda \langle \tilde{\Psi} | j(0, t) j(0, t') | \tilde{\Psi} \rangle^c_\lambda |$, which is expected to hold[2]). The physical meaning of iii) is that strong long-range temporal correlations for the current must be absent in the microscopic bipartite quench problem (from the initial state $|\tilde{\Psi}\rangle_\lambda$). This is non-trivial, because after quantum quenches such strong correlations generically develop, which may lead to the violation of iii) [24, 53]. We shall demonstrate that iii) is fulfilled in the physically important class of integrable quenches in free fermion models. An example where iii) is broken was given in [24], in which long-range temporal correlations were found of the form $|t - t'|^3 / (t + t')^4$ in the current-current 2-point function after a similar quench problem. An important part of our result is that *indeed,* (9) *is not expected to hold if condition iii) is broken, something which happens in certain physical situations.* However, as we argued in [24], even if iii) is violated, the simple hydrodynamic framework we discuss in this work can be applied and can yield exact prediction for the scaled cumulants up to a given order. We shall elaborate more on this issue in our conclusions, Section 7.

An important additional remark is that, as we show in Section 2.3, condition iii) guarantees that the scaled cumulants $\tilde{\kappa}^s_j(n)$ associated with the current in the state $|\tilde{\Psi}\rangle_\lambda$ are finite:

$$\tilde{\kappa}^s_j(n) := 2 \lim_{T \to \infty} \frac{1}{T} \int_0^T \prod_{i=1}^n dt_i \, _\lambda \langle \tilde{\Psi} | \prod_{i=1}^n j(0, t_i) | \tilde{\Psi} \rangle^c_\lambda < \infty \quad \forall n \geq 1. \tag{23}$$

Further, combined with condition ii), the result (11) holds, and scaled cumulants may be evaluated within $\rho_{\text{NESS}}$,

$$\tilde{\kappa}^s_j(n) = \tilde{\kappa}^s_{\text{dyn}}(n) \tag{24}$$

with

$$\tilde{\kappa}^s_{\text{dyn}}(n) := \partial^n_{\lambda'} \left( 2 \lim_{T \to \infty} \frac{1}{T} \ln \text{Tr} \left[ e^{\lambda' J_0 |_0^T} \rho_{\text{NESS}}(\lambda) \right] \right) \Big|_{\lambda' = 0}$$
$$= 2 \lim_{T \to \infty} \frac{1}{T} \int_0^T \prod_{i=1}^n dt_i \text{Tr} \left( \rho_{\text{NESS}}(\lambda) \prod_{i=1}^n j(0, t_i) \right)^c. \tag{25}$$

Note how we distinguished the counting field in the exponential and the inhomogeneity parameter of the density matrix of the NESS to obtain a partial expansion of $f_{\text{dyn}}$ wrt. the scaled cumulants $\tilde{\kappa}^s_{\text{dyn}}$ (which themselves depend on $\lambda$). Also note, that these are not the cumulant associated to the charge in the original problem, as for the latter equating $\lambda' = \lambda$ first is necessary.

That is, the cumulants in $|\tilde{\Psi}\rangle_\lambda$ grow at most extensively[3] with $T$, and their leading behaviour is equal to that of the cumulants in $\rho_{\text{NESS}}$.

Finally, as we mentioned, we remark that in order to evaluate (9) from hydrodynamic and thermodynamic data using the BFT, the absence of temporal correlations is required [26]. However, we note that at least in integrable systems, there exists a theoretical framework, the Ballistic Macroscopic Fluctuation Theory (BMFT) [30], which can take into account the effect of temporal correlations in quantities such as (9), relying the thermodynamic and hydrodynamic properties of the system as well as the knowledge of initial correlations [24]. The rather challenging task of integrating the BMFT into our current hydrodynamic picture is, however, out of the scope of this work.

---

[2]As mentioned, if the algebra of observables has a norm, the bound is immediate. If not, then a bound in terms of cumulants of currents is $|C^{jj}_{\tilde{\Psi}}| \leq \left( \sup_{t \in \mathbb{R}^+} \sqrt{_\lambda \langle \tilde{\Psi} | j(0, t) j(0, t) | \tilde{\Psi} \rangle^c_\lambda} \right)^2$. In field theory, one would need a finite-separation or UV regularisation.

[3]It is natural to also demand that $\tilde{\kappa}^s_{\text{dyn}}(2) > 0$, as if $\tilde{\kappa}^s_{\text{dyn}}(2) = 0$ then $\tilde{\kappa}^s_{\text{dyn}}(n) = 0$ for all $n \geq 2$; that is, we ask that the large-deviation principle is valid for current fluctuation with a linear-in-$T$ behaviour.

## 2.3 Derivation

We now show how conditions i)-iii), along with a principle of Asymptotic Commutativity (which we will describe), yield the identity (8) and hydrodynamic formula (9). We recall that this is to hold order by order in $\lambda$.

**Contour deformation**

The prime difficulty in computing the FCS of $Q|_0^X(T)$ after quenches, is that the quench generally gives rise to long-range dynamical correlations, which may be interpreted as "entangled quasi-particles" emitted by the initial state. Expanding the exponential in the cumulant generating function $\langle \Psi | e^{\lambda Q|_0^X(T)} | \Psi \rangle$ in powers of $\lambda$, we see that we need to compute connected $n$-point correlation functions of the time evolved density of the conserved charge and perform a spatial integration. In particular, considering the $n$-th derivative we have

$$\frac{\partial^n}{\partial \lambda^n} F(\lambda, X, T)|_{\lambda=0} = \int_0^X \prod_{j=1}^n \mathrm{d}x_j \, \langle \Psi | \prod_{j=1}^n q(x_j, T) | \Psi \rangle^c \,. \tag{26}$$

The problem is that it is not known how to directly evaluate such correlation functions. If $T$ is much larger than $X$, then the system converges to the maximal entropy state (MES) associated to $|\Psi\rangle$ and the problem is reduced to that of correlation functions within it. This is a simpler problem, namely that of the thermodynamics of the MES, for which many techniques are available. However, for the opposite regime (2), long-range correlations appear due to the quench, and modify the thermodynamic result. This is what makes the dynamical problem more difficult, and for which the present paper proposes a general theory.

    The main idea is to circumvent the problem of long-range correlations via suitable contour-manipulations inspired by [53]. As we shortly demonstrate such contour manipulations eventually allow for the usage of specific non-equilibrium states more likely to lack strong long-range correlations and hence make it possible to evaluate the integrated correlation functions by much simpler means, in particular using the BFT. The contour manipulations are based on the continuity equation of conserved quantities (1) and by rewriting the integrated charge appearing in the FCS as

$$Q|_0^X(T) = \int_0^X \mathrm{d}x \, q(x, T) = \int_0^X \mathrm{d}x \, q(x, 0) + \int_0^T \mathrm{d}t \, j(0, t) - \int_0^T \mathrm{d}t \, j(X, t) \,. \tag{27}$$

We modified the integration contour over space-time points as $(0, T) \to (0, 0) \to (X, 0) \to (X, T)$. Performing this step we avoid the correlations along the contour $(0, T) \to (X, T)$ due to entangled quasi-particles emitted by the initial state.

**Large-$X$: separation between left- and right-contributions (Factorisation Property 1)**

To proceed we recall the separation of spatial and temporal scales (2) and use the cluster property of the initial state i) and the Lieb-Robinson condition a). From this we may rewrite the SCGF as

$$F(\lambda, X, T) = \left( X \tilde{f}_\Psi + o(X) \right) + \left( T \tilde{f}_{\mathrm{dyn}}^{\mathrm{L}}(\lambda) + T \tilde{f}_{\mathrm{dyn}}^{\mathrm{R}}(\lambda) + o(T) \right) + o(X^0)_T \tag{28}$$

where

$$\begin{aligned} T \tilde{f}_{\mathrm{dyn}}^{\mathrm{L}}(\lambda) &:= \lim_{X \to \infty} \ln \frac{\langle \Psi | e^{\lambda J_0|_0^T + \lambda Q|_0^{X/2}(0)} | \Psi \rangle}{\langle \Psi | e^{\lambda Q|_0^{X/2}(0)} | \Psi \rangle} + o(T) \\[2mm] T \tilde{f}_{\mathrm{dyn}}^{\mathrm{R}}(\lambda) &:= \lim_{X \to \infty} \ln \frac{\langle \Psi | e^{\lambda Q|_{X/2}^X(0) - \lambda J_X|_0^T} | \Psi \rangle}{\langle \Psi | e^{\lambda Q|_{X/2}^X} | \Psi \rangle} + o(T) \end{aligned} \tag{29}$$

and

$$X\tilde{f}_\Psi(\lambda) := \ln\langle\Psi|e^{\lambda Q|_0^{X/2}(0)}|\Psi\rangle + \ln\langle\Psi|e^{\lambda Q|_{X/2}^X(0)}|\Psi\rangle + o(X).\tag{30}$$

By translation invariance, it is clear that $\tilde{f}_\Psi(\lambda) = f_\Psi(\lambda)$ according to definition (6). Further, in the second line of (29), we may shift every spatial argument by $-X$, and use parity transformation, under which $|\Psi\rangle$ is invariant (by condition i)) and under which $Q|_{-X/2}^0(0) \to Q|_0^{X/2}(0)$ and $J_0|_0^T \to -J_0|_0^T$, showing that $\tilde{f}_{\mathrm{dyn}}^{\mathrm{L}}(\lambda) = \tilde{f}_{\mathrm{dyn}}^{\mathrm{R}}(\lambda) =: \tilde{f}_{\mathrm{dyn}}(\lambda)$.

A sketch of a rigorous proof of the asymptotic form (28), at all orders in $\lambda$, is provided in Sec. 4.1. There, we use exponential clustering of two-point functions, and the recent results for clustering of higher-point correlation functions [51] that follow from it. The proof fails at large orders in $\lambda$ if only power-law clustering is assumed, however, many of the bounds can be made tighter, and we expect power-law clustering still to be sufficient at all orders.

The intuition behind (28) is as follows. We note that on the right-hand side of (27), $Q|_0^X(0)$ may be split into $Q|_0^{X/2}(0) + Q|_{X/2}^X(0)$. The resulting set of operators are all commuting in the regime of interest:

$$[Q|_0^{X/2}(0), Q|_{X/2}^X(0)] = 0\tag{31}$$

from ultra-locality, and assuming finite LR velocity momentarily,

$$[J_X|_0^T, Q|_0^{X/2}(0)] = [J_0|_0^T, Q|_{X/2}^X(0)] = 0, \quad \text{and} \quad [J_0|_0^T, J_X|_0^T] = 0,\tag{32}$$

when $X \gg T$, as the operators will be supported on different regions; for instance, $J_0|_0^T$ is supported on a region of length at most $2v_{\mathrm{LR}}T$ around $x = 0$. Thus operators can be re-ordered (the exponential of the right-hand side of (27) separates as a product of exponentials). But most importantly, the pairs of operators in (32) are all far from each other. Thus by clustering, the averages involving these, and their powers, factorise – these give corrections terms $o(X^0)_T$ in (28). Operators in (31) are not far from each other, but clustering guarantees that the resulting averages only give finite, $O(1)$ contributions that are independent of $T$ – these are parts of correction terms $o(X)$ in (28). This intuition is worked out in details in Sec. 4.1.

If there is no Lieb-Robinson bound, then one can still argue that as long as the occupation of high velocity quasi-particle is strongly suppressed the contribution of fast excitation can be neglected, and similar arguments can be made. In Sec. 4.1, we also argue more explicitly that (28) is justified up to 2nd order in $\lambda$.

**The modified quench state: Asymptotic Commutativity (Factorisation Property 2)**

From now on, let us focus on $\tilde{f}_{\mathrm{dyn}}^{\mathrm{L}}(\lambda)$. The next step is thus rewriting $\tilde{f}_{\mathrm{dyn}}^{\mathrm{L}}(\lambda)$ in (29) as

$$T\tilde{f}_{\mathrm{dyn}}^{\mathrm{L}}(\lambda) = \ln\frac{\langle\Psi|e^{\lambda J_0|_0^T}e^{\lambda Q|_0^\infty(0)}|\Psi\rangle}{\langle\Psi|e^{\lambda Q|_0^\infty(0)}|\Psi\rangle} + o(T) = \ln\frac{\langle\Psi|e^{\lambda/2 Q|_0^\infty(0)}e^{\lambda J_0|_0^T}e^{\lambda/2 Q|_0^\infty(0)}|\Psi\rangle}{\langle\Psi|e^{\lambda Q|_0^\infty(0)}|\Psi\rangle} + o(T).\tag{33}$$

That is, the exponential in (29) may be written as a product of exponentials, for the leading-in-$T$ asymptotic behaviour. This is the very non-trivial step of "Asymptotic Commutativity", which says that for the sake of the large-$T$ behaviour, currents and charge may be considered to be commuting observables. We provide in Sec. 4.2 a check in the context of free fermions, relying on condition i). Note that this does not require a Lieb-Robinson bound. Our checks are performed for the case of the particle number, where higher order terms can be treated in a transparent way, but the calculations make (33) plausible for generic conserved charges. We therefore have that

$$f_{\mathrm{dyn}} = \lim_{T\to\infty}\frac{2}{T}\ln\mathrm{Tr}\left[e^{\lambda J_0|_0^T}\rho_{\mathrm{in}}\right]\tag{34}$$

where

$$\rho_{\mathrm{in}} = |\tilde{\Psi}\rangle_\lambda {}_\lambda\langle\tilde{\Psi}|\tag{35}$$

with $|\tilde{\Psi}\rangle_\lambda$ defined in (15) yielding the hydrodynamic identity (8).

**Emergence of the NESS, and its temporal decay**

The last step in our derivation is to use conditions ii) and iii) in order to show that $f_{\mathrm{dyn}}$, from (34), can be evaluated in the state $\rho_{\mathrm{NESS}}$ instead of $\rho_{\mathrm{in}}$, as in Eq. (9), and that currents have fast-enough vanishing correlations in $\rho_{\mathrm{NESS}}$, Eq. (11).

Eq. (11) follows from conditions ii) and iii), and Lieb-Robinson bound. Indeed, for every $s, s' \in \mathbb{R}$, by the Lieb-Robinson bound, the observables $O_1(0,0) = j(0,s)j(0,s')$, $O_2(0,0) = j(0,s)$ and $O_3(0,0) = j(0,s')$ are quasi-local at the origin $x = 0$. Hence on the left-hand side of (16) (condition ii)), we may use $O_1(0,t) = j(0,t+s)j(0,t+s')$, $O_2(0,t) = j(0,t+s)$ and $O_3(0,t) = j(0,t+s')$, and we get, with the definition (35),

$$
\begin{aligned}
\mathrm{Tr}\Big(\rho_{\mathrm{in}} j(0,t+s)j(0,t+s')\Big)^c &= \mathrm{Tr}\Big(\rho_{\mathrm{in}} O_1(0,t)\Big) - \mathrm{Tr}\Big(\rho_{\mathrm{in}} O_2(0,t)\Big)\mathrm{Tr}\Big(\rho_{\mathrm{in}} O_3(0,t)\Big) \\
&\stackrel{t\to\infty}{\to} \mathrm{Tr}\Big(\rho_{\mathrm{NESS}} O_1(0,0)\Big) - \mathrm{Tr}\Big(\rho_{\mathrm{NESS}} O_2(0,0)\Big)\mathrm{Tr}\Big(\rho_{\mathrm{NESS}} O_3(0,0)\Big) \\
&= \mathrm{Tr}\Big(\rho_{\mathrm{NESS}} j(0,t+s)j(0,t+s')\Big)^c.
\end{aligned}
\tag{36}
$$

Then, applying condition iii) with the replacements $t \to t+s$ and $t' \to t+s'$ in (17), the uniformity requirement in condition iii) implies that the vanishing in the limit $|s-s'| \to \infty$ holds uniformly in $t$, and we recover Eq. (11) by taking the limit $t \to \infty$.

Now we show that if conditions ii) and iii) hold, that is, if the system after quenching from $|\tilde{\Psi}\rangle_\lambda$ locally converges to a NESS around $x = 0$ and no temporal long-range correlations are present in the current multi-point functions, then integrated temporal current correlation functions in $\rho_{\mathrm{in}}$ converge to those of the NESS $\rho_{\mathrm{NESS}}$. For simplicity, we shall focus on the 2-pt function and the associated cumulant, that is

$$
\begin{aligned}
\tilde{\kappa}_2(T) &= \int_0^T\int_0^T \mathrm{d}t\,\mathrm{d}t'\,{}_\lambda\langle\tilde{\Psi}|j(0,t)j(0,t')|\tilde{\Psi}\rangle_\lambda^c \\
&= \int_0^{\sqrt{2}T/2}\mathrm{d}\sigma\int_{-\sigma}^{\sigma}\mathrm{d}\tau\,C_{\tilde{\Psi}}^{jj}(\sigma,\tau) + \int_{\sqrt{2}T/2}^{\sqrt{2}T}\mathrm{d}\sigma\int_{-\sqrt{2}T+\sigma}^{\sqrt{2}T-\sigma}\mathrm{d}\tau\,C_{\tilde{\Psi}}^{jj}(\sigma,\tau),
\end{aligned}
\tag{37}
$$

where we introduced new integration variables $\sigma = (t+t')/\sqrt{2}$ and $\tau = (t-t')/\sqrt{2}$, and

$$
C_{\tilde{\Psi}}^{jj}(\sigma,\tau) = {}_\lambda\langle\tilde{\Psi}|j(0,t)j(0,t')|\tilde{\Psi}\rangle_\lambda^c.
\tag{38}
$$

Now we introduce $0 < c < \sqrt{2}T$ and rewrite the integral as

$$
\begin{aligned}
\tilde{\kappa}_2(T) &= \int_c^{\sqrt{2}T-c}\mathrm{d}\sigma\int_{-c}^{c}\mathrm{d}\tau\,C_{\tilde{\Psi}}^{jj}(\sigma,\tau) + \int_c^{\sqrt{2}T/2}\mathrm{d}\sigma\left(\int_{-\sigma}^{-c}+\int_c^{\sigma}\right)\mathrm{d}\tau\,C_{\tilde{\Psi}}^{jj}(\sigma,\tau) \\
&+ \int_{\sqrt{2}T/2}^{\sqrt{2}T-c}\mathrm{d}\sigma\left(\int_{-\sqrt{2}T+\sigma}^{-c}+\int_c^{\sqrt{2}T-\sigma}\right)\mathrm{d}\tau\,C_{\tilde{\Psi}}^{jj}(\sigma,\tau) + B(c,T),
\end{aligned}
\tag{39}
$$

where $B(c,T)$ corresponds to the part of the integral on regions of linear lengths $\propto c$, independent of $T$. By the uniform bound on $|C_{\tilde{\Psi}}^{jj}(\sigma,\tau)|$ of condition iii), we therefore have

$$
|B(c,T)| \le B_0 c^2 \sup_{t,t'>0} |C_{\tilde{\Psi}}^{jj}(\sigma,\tau)|
\tag{40}
$$

for some constant ($c, T$-independent) $B_0 > 0$.

In the second and third terms on the right-hand side of (39), for clarity we write

$$
C_{\tilde{\Psi}}^{jj}(\sigma,\tau) = A(\sigma,\tau)/\tau,
\tag{41}
$$

where, by condition iii), Eq. (17), $\lim_{\tau \to \infty} A(\sigma, \tau) = 0$ uniformly in $\sigma$. We note that for any $u = u(\sigma, T)$,

$$\left| \int_{-u}^{-c} d\tau \frac{A(\sigma, \tau)}{\tau} \right| \leq \int_{-u}^{-c} d\tau \frac{|A(\sigma, \tau)|}{\tau} \leq \int_{-\infty}^{-c} d\tau \frac{|A(\sigma, \tau)|}{\tau} \to 0 \qquad (c \to \infty) \qquad (42)$$

uniformly in $\sigma, T$. This is because by the condition on $A(\sigma, \tau)$, the integral of $\frac{|A(\sigma,\tau)|}{\tau}$ over $\tau \in (-\infty, -c)$ exists, and as the integrand is non-negative, the integral is non-increasing and its limit is 0 as $c \to \infty$ uniformly in $\sigma$. A similar result holds for $\left| \int_c^u d\tau \frac{A(\sigma,\tau)}{\tau} \right|$. Therefore,

$$\tilde{\kappa}_2(T) = \int_c^{\sqrt{2}T-c} d\sigma \int_{-c}^c d\tau \, C_{\tilde{\Psi}}^{jj}(\sigma, \tau) + (T/\sqrt{2} - c)o(1) + B(c, T) \qquad (c \to \infty) \qquad (43)$$

uniformly in $T$.

By (36) we have, for every $\tau$,

$$\lim_{\sigma \to \infty} C_{\tilde{\Psi}}^{jj}(\sigma, \tau) = \text{Tr}\Big(\rho_{\text{NESS}} j(0, t) j(0, t')\Big)^c, \qquad (44)$$

which is independent of $\sigma$ by stationarity of $\rho_{\text{NESS}}$. Hence, for every $c$,

$$T^{-1} \int_c^{\sqrt{2}T-c} d\sigma \int_{-c}^c d\tau \, C_{\tilde{\Psi}}^{jj}(\sigma, \tau) = T^{-1} \int_c^{\sqrt{2}T-c} d\sigma \int_{-c}^c d\tau \left( \text{Tr}\Big(\rho_{\text{NESS}} j(0, t) j(0, t')\Big)^c + \varepsilon(\sigma, \tau) \right)$$

$$\to \sqrt{2} \int_{-c}^c d\tau \, \text{Tr}\Big(\rho_{\text{NESS}} j(0, t) j(0, t')\Big)^c \qquad (T \to \infty)$$

$$= \int_{-c}^c dt \, \text{Tr}\Big(\rho_{\text{NESS}} j(0, t) j(0, 0)\Big)^c. \qquad (45)$$

In the second line, we used the fact that $\lim_{\sigma \to \infty} \varepsilon(\sigma, \tau) = 0$, which implies

$$\left| T^{-1} \int_c^{\sqrt{2}T-c} d\sigma \int_{-c}^c d\tau \, \varepsilon(\sigma, \tau) \right| \leq T^{-1} \int_c^b d\sigma \int_{-c}^c d\tau \, |\varepsilon(\sigma, \tau)|$$

$$+ \left( \sqrt{2} - \frac{c+b}{T} \right) \int_{-c}^c d\tau \sup_{\sigma \in [b,\infty]} |\varepsilon(\sigma, \tau)|$$

$$\Rightarrow \lim_{T \to \infty} \left| T^{-1} \int_c^{\sqrt{2}T-c} d\sigma \int_{-c}^c d\tau \, \varepsilon(\sigma, \tau) \right| \leq \sqrt{2} \int_{-c}^c d\tau \sup_{\sigma \in [b,\infty]} |\varepsilon(\sigma, \tau)|. \qquad (46)$$

We then use the bounded convergence theorem: as $b$ can be taken as large as desired, $|\varepsilon(\sigma, \tau)|$ is uniformly bounded, and $\lim_{b \to \infty} \sup_{\sigma \in [b,\infty]} |\varepsilon(\sigma, \tau)| = 0$, the limit in (46) vanishes, and the second line of (45) follows.

Using this result in (43), along with (40) and uniformity in $T$ of the $c \to \infty$ decaying term $o(1)$, we obtain (recall (23))

$$\tilde{\kappa}_j^s(2) = 2 \lim_{T \to \infty} T^{-1} \tilde{\kappa}_2(T) \to 2 \int_{-c}^c dt \, \text{Tr}\Big(\rho_{\text{NESS}} j(0, t) j(0, 0)\Big)^c \qquad (c \to \infty). \qquad (47)$$

Because the left-hand side is independent of $c$, we may take the limit $c \to \infty$ and we find

$$\tilde{\kappa}_j^s(2) = 2 \int_{-\infty}^{\infty} dt \, \text{Tr}\Big(\rho_{\text{NESS}} j(0, t) j(0, 0)\Big)^c. \qquad (48)$$

Due to (11), which we have shown above, the derivation above can be repeated with $C_{\tilde{\Psi}}^{jj}(\sigma, \tau)$ replaced by $\mathrm{Tr}\big(\rho_{\mathrm{NESS}}j(0,t)j(0,t')\big)^c$ (in which case $\varepsilon(\sigma, \tau) = 0$). Recalling the definition (25), we therefore obtain

$$\tilde{\kappa}_{\mathrm{dyn}}^s(2) = 2 \int_{-\infty}^{\infty} \mathrm{d}t \, \mathrm{Tr}\big(\rho_{\mathrm{NESS}}j(0,t)j(0,0)\big)^c. \tag{49}$$

This shows (24) in the case $n = 2$.

Therefore, using (34), we have shown (9) at second order in $\lambda$.

Finally, one can show that (24) also holds for higher cumulants. The way to show this is analogous to what has been presented. In particular, one can introduce a "centre of mass" $\sigma = (t_1 + t_2 + ...t_n)/n$ and $n-1$ relative coordinates $\tau_i$ and first apply a cut-off $c$ in the integration wrt. to the relative time-coordinates. Then it can be argued that for large $\sigma$, the non-vanishing contribution to the cumulant comes from the bounded region for the relative integration coordinates in which the $n$-point connected correlation function can be replaced by its value in the NESS.

# 3   Free non-relativistic fermions and integrable quenches

We now explain the example of the free fermion and its integrable quenches. In later subsections, we will confirm that the three conditions above hold, and justify some of the steps above leading to our main result, using this model.

## 3.1   Model and charges

The free fermion Hamiltonian is defined as

$$H_{\mathrm{FF}} = \int \mathrm{d}x \, \psi^\dagger(x)\big[-\partial_x^2\big]\psi(x) = \int \mathrm{d}p \, E_p a_p^\dagger a_p \,, \tag{50}$$

where we have set the mass to $1/2$, integration is implicitly over $\mathbb{R}$, and $\psi^\dagger(x), \psi(x)$ are canonical fermionic operators, with anti-commutation relation $\{\psi(x), \psi^\dagger(y)\} = \delta(x - y)$. The Hamiltonian can be diagonalised using the Fourier decomposition of the fundamental field resulting in the rhs. of (50), where $\{a_p, a_{p'}^\dagger\} = \delta(p - p')$, and $E_p = E(p) = p^2$. The diagonalisation is achieved by

$$a_k = \frac{1}{\sqrt{2\pi}} \int \mathrm{d}x e^{-\mathrm{i}kx}\psi(x), \quad a_k^\dagger = \frac{1}{\sqrt{2\pi}} \int \mathrm{d}x e^{\mathrm{i}kx}\psi^\dagger(x) \tag{51}$$

and by the corresponding inverse transformation

$$\psi(x,t) = \frac{1}{\sqrt{2\pi}} \int \mathrm{d}k \, e^{\mathrm{i}kx - \mathrm{i}E_k t}a_k, \quad \psi^\dagger(x,t) = \frac{1}{\sqrt{2\pi}} \int \mathrm{d}k \, e^{-\mathrm{i}kx + \mathrm{i}E_k t}a_k^\dagger. \tag{52}$$

The conserved charges in the model are also worth introducing. The local conserved charges can be written by the densities

$$
\begin{aligned}
q_{2n}(x) &= \partial_x^n \psi^\dagger(x)\partial_x^n \psi(x) \\
q_{2n+1}(x) &= \mathrm{i}\big(\partial_x^{n+1}\psi^\dagger(x)\partial_x^n\psi(x) - \partial_x^n\psi^\dagger(x)\partial_x^{n+1}\psi(x)\big) \,,
\end{aligned}
\tag{53}
$$

where $Q_i = \int \mathrm{d}x \, q_i(x)$ and in particular $Q_0$ is the total particle number, $Q_1$ is the total momentum, and $Q_2$ is the total energy and charges with an even/odd index are even/odd under spatial parity transformation $\hat{P}$. In addition, in free models it is also true that the total current associated with an even charge $Q_{2n}$ is the total, odd charge $Q_{2n+1}$, and vice versa; for instance, the total momentum

equals the total particle number current and the total momentum current equals the total energy. The current densities can be obtained from the charge densities of (53) by applying the $i\overset{\leftrightarrow}{\partial}_x$ operation. The specific gauge we have chosen for the charge densities also guarantees that the current density associated with an even charge $Q_{2n}$ equals the charge density of the odd charge $Q_{2n+1}$ (but not vice versa in this case due to differences accounting for a total derivative).

It is possible to define quasi-local charges in a simple a manner [54, 55] using

$$q_\alpha(x) = \psi^\dagger(x)\psi(x+\alpha), \tag{54}$$

where $\alpha$ is a real and continuous parameter, and which, when combined as

$$q_\alpha^e(x,t) = \frac{q_\alpha(x) + q_\alpha^\dagger(x)}{2}, \qquad q_\alpha^o(x,t) = \frac{q_\alpha(x) - q_\alpha^\dagger(x)}{2i}, \tag{55}$$

are made self-adjoint, and parity even and odd, respectively, as well.

In fact, one can define the full space of extensive charges by integrating $\int d\alpha\, f(\alpha)q_\alpha$ with a suitable $L^2$ space of functions $f$; the $q_\alpha$'s form a "scattering basis", and the $q_n$'s span a dense subspace.

We may characterise the conserved charge densities in Fourier space, that is, using the fermionic oscillator modes. They can be written as

$$q_\kappa^{e/o}(x,t) = \frac{1}{2\pi} \int dk dp\, e^{i(E_k - E_p)t} e^{-i(k-p)x} q_\kappa^{e/o}(k,p)\, a_k^\dagger a_p, \tag{56}$$

where $\kappa$ can be an integer $m$ (even or odd depending on the parity of the charge) or a continuous real number $\alpha$, and with a slight abuse of the notation $q_\kappa^{e/o}(k,p)$ are defined as

$$
\begin{aligned}
q_{2n}^e(k,p) &= (kp)^n, \quad q_{2n+1}^o(k,p) = (k+p)(kp)^n \\
q_\alpha^e(k,p) &= \frac{e^{ik\alpha} + e^{-ip\alpha}}{2}, \quad q_\alpha^o(k,p) = \frac{e^{ik\alpha} - e^{-ip\alpha}}{2i}.
\end{aligned}
\tag{57}
$$

Using the shorthand $q_\kappa^{e/o}(k) = q_\kappa^{e/o}(k,k)$, the total charges can be written as

$$Q_\kappa^{e/o} = \int dk\, q_\kappa^{e/o}(k) a_k^\dagger a_k, \tag{58}$$

where $q_\kappa^{e/o}(k)$ is the 1-particle eigenvalue of the charge, that is,

$$Q_\kappa^{e/o}|k\rangle = Q_\kappa^{e/o} a_k^\dagger|0\rangle = q_\kappa^{e/o}(k)|k\rangle, \tag{59}$$

where $|0\rangle$ is the free fermion vacuum, and which read for the local and quasi-local charges as

$$q_{2n}^e(k) = k^{2n}, \quad q_{2n+1}^o(k) = 2k^{2n+1}, \quad q_\alpha^e(k) = \cos(\alpha k), \quad q_\alpha^o(k) = \sin(\alpha k). \tag{60}$$

The current densities of the local parity even operators expressed via the fermionic creation and annihilation operators clearly equal that of (56), but the currents of the odd charges are also easy to express

$$
\begin{aligned}
j_{2n}(x,t) &= \frac{1}{2\pi} \int dk dp\, e^{i(E_k - E_p)t} e^{-i(k-p)x} \frac{E_k - E_p}{k-p} q_{2n}(k,p)\, a_k^\dagger a_p \\
&= \frac{1}{2\pi} \int dk dp\, e^{i(E_k - E_p)t} e^{-i(k-p)x} (k+p) q_{2n}(k,p)\, a_k^\dagger a_p \\
&= \frac{1}{2\pi} \int dk dp\, e^{i(E_k - E_p)t} e^{-i(k-p)x} q_{2n+1}(k,p)\, a_k^\dagger a_p \\
j_{2n+1}(x,t) &= \frac{1}{2\pi} \int dk dp\, e^{i(E_k - E_p)t} e^{-i(k-p)x} (k+p) q_{2n+1}(k,p)\, a_k^\dagger a_p
\end{aligned}
\tag{61}
$$

as a simple consequence of the continuity equation (1) and by demanding that the current vanishes $|x| \to \infty$.

## 3.2 Integrable quenches and hydrodynamic predictions for FCS

For the out-of-equilibrium dynamics we shall consider integrable quenches [56] in the free fermion model, specified by the initial state, which we denote by $|\Omega\rangle$ to distinguish them from the non-integrable counterparts. The reason for this choice is multifaceted. Besides belonging to an important class of quenches, integrable initial states admit the exact characterisation of the steady-state GGEs [44, 57]. Regarding the microscopic bipartite quench from $e^{\lambda/2Q|_0^\infty}|\Omega\rangle$, the integrability of $|\Omega\rangle$ in a free fermion theory allows us to perform numerous microscopic computations to check our derivation and the fulfillment of the criteria. In particular for such quenches one can not only show the fulfillment of conditions i), but it is also possible to demonstrate explicitly the rapid convergence to $\rho_{\text{NESS}}$ and the suppression of temporal correlations in current multi-point functions (i.e., conditions ii) and iii)). In other words, we can demonstrate the validity of both (8) and (9) together with how the latter expression emerges from the former. The NESS for the microscopic problem $\langle\Omega|e^{\lambda/2Q|_0^\infty}O(0,t)e^{\lambda/2Q|_0^\infty}|\Omega\rangle$ can also be specified explicitly: $\rho_{\text{NESS}}$ can be shown to be identical to the NESS $\rho_{\text{bpp}}$ after the bipartitioning protocol with $\rho_{\text{GGE}}$ and $\rho_{\text{GGE}}^{(\lambda)}$ (see the discussion around Eq. (20)). Crucially, it is also possible to determine the GGE of the modified quench problem as well, that is, $\rho_{\text{GGE}}^{(\lambda)}$ emerging after quenching from $e^{\lambda/2Q}|\Omega\rangle$. This GGE can be written as $\rho_{\text{GGE}}^{(\lambda)} \propto \rho_{\text{GGE}}e^{2\lambda Q}$, which is a rather non-trivial finding. Interestingly, this latter GGE also encodes the FCS of the *initial state* $f_\Omega(\lambda)$ which is a further surprising non-trivial statement and was demonstrated in [21, 22]. Last but not least, the original non-equilibrium problem of computing the FCS after integrable quenches can be solved by correlation matrix techniques [20, 21] and the corresponding results allow us to benchmark our theory.

Integrable quenches have the distinctive property that the initial states have non-vanishing overlaps with only those eigenstates of the post-quench Hamiltonian that consist of pairs of particles with opposite momentum. For free fermions we have an infinite family of such initial states defined as

$$|\Omega\rangle = N^{-1/2}\exp\left(\int_0^\infty \frac{dk}{2\pi}K(k)a_k^\dagger a_{-k}^\dagger\right)|0\rangle\,, \tag{62}$$

where $|0\rangle$ is the vacuum, $a_k^\dagger$ and $a_k$ are fermionic creation and annihilation operators in momentum space defined in (51). In (62), $N$ ensures the normalisation of the initial state and $K(k)$ is an arbitrary function which decays fast enough for large $k$ and has the property $K(-k)=K(k)$ which is required for the consistent definition of (62). Additionally and for simplicity we shall also assume that the $K$-function has a zero at the origin and it is (complex) analytic. For these initial states, it possible to carry out microscopic computations as well as to compute the steady-state spectral densities corresponding to the GGE due to the Bogolyubov transformation. The Bogolyubov transformation for the quench problem can be defined as

$$a_k = u_k\tilde{a}_k + v_k\tilde{a}_{-k}^\dagger\,, \qquad a_k^\dagger = v_k^*\tilde{a}_{-k} + u_k\tilde{a}_k^\dagger \tag{63}$$

with

$$u_k = \frac{1}{\sqrt{1+|K(k)|^2}}\,, \quad v_k = \frac{K(k)}{\sqrt{1+|K(k)|^2}} \tag{64}$$

ensuring that

$$\tilde{a}_k|\Omega\rangle = 0\,, \forall k\,. \tag{65}$$

Thanks to the above, it is immediate to show that the initial state satisfy the strong enough cluster property i) (cf. Appendix A), moreover, the GGE after the quench can be written explicitly (cf. Appendix C) as

$$\rho_{\text{GGE}} \propto e^{\int dk\left(\ln|K(k)|^2 a_k^\dagger a_k\right)}\,. \tag{66}$$

The GGE can be unambiguously characterised by the corresponding fermion occupation number or filling $\theta(k)$ $(1 \geq \theta(k) \geq 0)$ which in this case reads as

$$\theta(k) = \frac{|K(k)|^2}{1 + |K(k)|^2} \, . \tag{67}$$

The initial states $|\Omega\rangle$ allow us to show that conditions ii) and iii) hold for these particular quenches. Regarding ii) we recall that it requires that every local observable relaxes to a NESS after quenching from the bipartite state $|\tilde{\Omega}\rangle_\lambda$. Because of the inhomogeneous initial state, rigorously demonstrating ii) in full generality is rather complicated even for free systems, although this condition is generally assumed to hold. Therefore our analysis is restricted to the class of conserved densities (53) and (55) (the details of the calculation can be found in Appendix C), since the relaxation of conserved densities is expected to be sufficient for the relaxation of local and quasi-local operators. For instance, in theories such as BMFT [30], a core assumption is that non only the steady-state expectation values of local operators can be expressed as a functional of the conserved densities as $O(x, t) = O[\{q_i(x, t)\}]$, but also their fluctuations via the same functional. Regarding now the behaviour of conserved densities, via series expansion wrt. $\lambda$ in $_\lambda\langle\tilde{\Omega}|(.)|\tilde{\Omega}\rangle_\lambda$ with

$$|\tilde{\Omega}\rangle_\lambda = \frac{e^{\lambda/2\tilde{Q}}|\Omega\rangle}{\sqrt{\langle\Omega|e^{\lambda\tilde{Q}}|\Omega\rangle}} \quad \text{and} \quad \tilde{Q} = Q|_0^\infty(0) = \int_0^\infty dx\, q(x, 0), \tag{68}$$

and performing explicit microscopic calculations we have checked in the first three orders for parity even conserved densities, and in the first two non-trivial orders for the current operator and for parity odd charge densities, that the corresponding expectation values at $x = 0$ indeed match with that of a GGE $\rho_{\text{bpp}}$ specified by the filling function

$$\theta_{\text{NESS}}(k, \lambda) = \theta_{\text{bpp}}(k, \lambda) = \Theta_{\text{H}}(v(k))\theta[|K|^2](k) + \Theta_{\text{H}}(-v(k))\theta[|K|^2 e^{2\lambda q}](k), \tag{69}$$

up to the specific orders in the counting field and where

$$v(k) = E'(k) = 2k \tag{70}$$

is the velocity of the free fermion excitations (note the factor of 2 due to the definition of $H_{\text{FF}}$ (50) with $E(k) = k^2$). For the sake of transparency, brevity, and due to its special role we discuss the case of the current density (despite not being a charge density) in the main text in Sec. 5. The analogous treatment of parity-odd densities and eventually the parity-even ones requiring a slightly different approach are presented in Appendix D. Additionally, the approach to the dictated value of $\rho_{\text{NESS}}$ is also identified and is found to be $\mathcal{O}(t^{-3})$ in both orders wrt. $\lambda$ for the current and for parity odd conserved densities, and $\mathcal{O}(t^{-3/2})$ for the parity even conserved densities in the first non-trivial order (which is $\lambda^2$ in the case of densities). This result also ensures that for integrated current $\int_0^T dt\, j(0, t)$ in the exponential (9), the power-law corrections originating from the convergence to the GGE vanish for large $T$.

Importantly, we also show in Sec. C.2 based on exact microscopic calculations for the modified *homogeneous* quench problem with

$$|\bar{\Omega}\rangle_\lambda = \frac{e^{\lambda/2Q}|\Omega\rangle}{\sqrt{\langle\Omega|e^{\lambda Q}|\Omega\rangle}} \left(\neq |\tilde{\Omega}\rangle_\lambda\right), \tag{71}$$

that

$$\lim_{T\to\infty} \langle\bar{\Omega}|O(t, x)|\bar{\Omega}\rangle = \frac{\text{Tr}\left[O(0, 0)\rho_{\text{GGE}} e^{2\lambda Q}\right]}{\text{Tr}\left[\rho_{\text{GGE}} e^{2\lambda Q}\right]} \tag{72}$$

where

$$\rho_{\text{GGE}}^{(\lambda)} \propto \rho_{\text{GGE}} e^{2\lambda Q} \propto e^{\int dk \ln(|K(k)|^2 e^{2\lambda q(k)}) a_k^\dagger a_k}, \tag{73}$$

that is, $\rho_{\mathrm{GGE}}^{(\lambda)} \propto e^{2\lambda Q} \rho_{\mathrm{GGE}}$ for integrable quenches. The emergence of this GGE is highly non-trivial, and relies on the integrability of both the initial state and the model (see a derivation for arbitrary integrable model and initial states via the Quench Action method in [24]). However, from this finding and from (69), it also immediate that $\rho_{\mathrm{NESS}} = \rho_{\mathrm{bpp}}$ as $\theta_{\mathrm{NESS}}(k, \lambda) = \theta_{\mathrm{bpp}}(k, \lambda)$, that is, we have demonstrated the intuition we phrased in (21) (and also in (22)).

Additionally, for integrable initial states the fulfillment of iii) can be explicitly checked as well assuring the absence of strong long-range correlations in current 2-pt functions and the detailed computation can be found in Sec. 6. In particular using the same expansion i.e., expanding the initial states wrt. $\lambda$ and computing the microscopic time evolution and the first two non-trivial orders both predict a separate $|t - t'|^{-3}$ and $(t + t')^{-3}$ behaviours, asymptotically. As $t, t > 0$, clearly the entire correlation function vanishes fast enough for large separations, i.e., for large $|t - t'|$ (as for positive times $t + t' \geq |t - t'|$), hence iii) is satisfied, at least for 2-pt functions in the first two non-trivial orders in $\lambda$. Importantly, the terms that account for the $|t - t'|^{-3}$, that is, time-translation invariant asymptotic behaviours match exactly with the behaviour predicted by $\rho_{\mathrm{NESS}}$ as we show in Sec. 5. This finding is in accordance with (44). It is also easy to see that $|t - t'|^{-3}$ term gives rise to an extensive cumulant, that is, $\kappa(2) \propto T$ which is captured by the NESS. On the contrary, the $(t + t')^{-3}$ term gives an $\mathcal{O}(1)$ correction to the cumulant. These findings strongly suggest that when considering higher orders, the behaviour of current correlation functions remain weak enough to fulfill iii).

Finally, as was already shown in [20, 21], for integrable quenches the dynamical SCGF $f_{\mathrm{dyn}}$ of (7) in free fermion systems can be expressed in an explicit way via the BFT Flow Equations, which we have also justified above within our theoretical framework. In the hydrodynamic picture, the use of these equations is based on the fact that the NESS $\rho_{\mathrm{NESS}}$ of the inhomogeneous quench problem is a maximum entropy steady-state, a GGE which is known to characterise explicitly, and that in integrable systems the current SCGF in such GGEs can be computed explicitly via these Flow Equations. In other words, the hydrodynamic formula (9) can be evaluated via the BFT Flow Equations which take following shape in our case:

$$f_{\mathrm{dyn}}(\lambda) = 2f_{\mathrm{j}}^{\mathrm{bpp}}(\lambda) = 2 \int_0^\lambda \mathrm{d}\beta \int \frac{\mathrm{d}k}{2\pi} E'(k) \bar{\theta}_\beta(k) q(k), \tag{74}$$

where $q(k)$ denotes the aforementioned 1-particle eigenvalue of the conserved charge under consideration, and finally $\bar{\theta}_\beta(k)$ denotes fermion occupation functions in specific macro-states. The quantities in the integrand of (74) satisfy

$$\partial_\beta \epsilon_\beta(k) = -\mathrm{sgn}(v(k)) q(k), \quad \text{or} \quad \partial_\beta \bar{\theta}_\beta(k) = \bar{\theta}_\beta(k) (1 - \bar{\theta}_\beta(k)) \mathrm{sgn}(v(k)) q(k)$$
$$\bar{\theta}_\beta(k) = \frac{1}{e^{\epsilon_\beta(k)} + 1}, \quad \text{with} \quad \bar{\theta}_0(k) = \theta_{\mathrm{NESS}}(k, \lambda) = \frac{1}{e^{\epsilon_0(k, \lambda)} + 1}, \tag{75}$$

where the initial condition is specified via the filling function of the NESS ($x \ll t$) $\theta_{\mathrm{NESS}}(k, \lambda)$ after the bipartitioning protocol (19) given by (69) and $v(k)$ is the velocity of the fermionic modes cf. (70). We note that in free fermion systems we can explicitly perform the $\beta$-integration in the Flow Equations after solving the equation for $\epsilon_\beta(k)$ or equivalently, $\bar{\theta}_\beta(k)$

$$\bar{\theta}_\beta(k) = \frac{E'(k) q(k) \theta_{\mathrm{bpp}}(k, \lambda)}{(1 - \theta_{\mathrm{bpp}}(k, \lambda)) e^{-\beta \, \mathrm{sgn}(k) q(k)} + \theta_{\mathrm{bpp}}(k, \lambda)}, \tag{76}$$

from which

$$f_{\mathrm{dyn}}(\lambda) = 2 \int \frac{\mathrm{d}k}{2\pi} |E'(k)| \ln \left[ (1 - \theta_{\mathrm{bpp}}(k, \lambda)) + \theta_{\mathrm{bpp}}(k, \lambda) e^{\lambda \, \mathrm{sgn}(k) q(k)} \right]. \tag{77}$$

In summary, for free fermions after integrable quenches we have explicitly shown how the BFT equations Eqs. (74) and (75) and more generally, hydrodynamic quantities allow the computation non-equilibrium fluctuations and we also specified the necessary thermodynamic data (69) in terms of $\rho_{\text{GGE}}$ and $\rho_{\text{GGE}} e^{2\lambda Q}$.

*Remark.* We note that in principle, the microscopic treatment of the inhomogeneous quench ${}_\lambda \langle \tilde{\Omega} | O(x,t) | \tilde{\Omega} \rangle_\lambda$ is tractable via an appropriate inhomogeneous Bogolyubov transformation, at least when $Q$ is the particle number, and the reader may wonder about the need for series expansion when ii) and iii) are checked. The reason for this, as discussed in Appendix E, is that the inverse of this transformation, which is necessary for computations, cannot be obtained analytically. The problem of computing the inverse transformation boils down to inverting the object $(1 - \mathcal{K})$ with an integral operator $\mathcal{K}$, and this task is known to be accomplished only via series expansion in general.

## 4 Factorisation properties

In this section we argue for the validity of two non-trivial steps in the derivation: the factorisation of exponentials as in Eq. (28) and (29) (Factorisation Property 1); and the Asymptotic Commutativity (Factorisation Property 2) in Eq. (33). We show that these properties are indeed satisfied if the initial state $|\Psi\rangle$ is clustering and translation invariant (Condition i)). We consider two setups: models with a Lieb-Robinson bound (for Factorisation Property 1), and the non-relativistic free fermion model, Section 3 (for both Factorisation Properties). We specify more precisely the type of clustering that we need depending on the setup and the property to prove: either we are able to assume only power law decay, or we need exponential decay. Our claims are based on expanding exponentials in $\lambda$, either to all orders (for Property 1) or only to the first leading orders.

### 4.1 Factorisation Property 1

In this section we argue that

$$\ln \langle \Psi | e^{\lambda \int_0^T dt\, j(0,t) + \lambda \int_0^X dx\, q(x,0) - \lambda \int_0^T dt\, j(X,t)} | \Psi \rangle = \ln \langle \Psi | e^{\lambda J_0|_0^T + \lambda Q|_0^X (0) - \lambda J_X|_0^T} | \Psi \rangle \tag{78}$$

can be split into the sum

$$\ln \frac{\langle \Psi | e^{\lambda J_0|_0^T + \lambda Q|_0^{X/2}(0)} | \Psi \rangle}{\langle \Psi | e^{\lambda Q|_0^{X/2}(0)} | \Psi \rangle} + \ln \frac{\langle \Psi | e^{\lambda Q|_{X/2}^X(0) - \lambda J_X|_0^T} | \Psi \rangle}{\langle \Psi | e^{\lambda Q|_{X/2}^X(0)} | \Psi \rangle} + \ln \langle \Psi | e^{\lambda Q|_0^{X/2}(0)} | \Psi \rangle + \ln \langle \Psi | e^{\lambda Q|_{X/2}^X(0)} | \Psi \rangle \tag{79}$$

up to sub-leading errors which vanish faster than $\propto X$ as $X \to \infty$, order by order in $\lambda$. That is, defining

$$A = J_0|_0^T + Q|_0^{X/2}, \quad B = Q|_{X/2}^X - J_X|_0^T, \tag{80}$$

we must show

$$\ln \langle \Psi | e^{\lambda(A+B)} | \Psi \rangle = \ln \langle \Psi | e^{\lambda A} | \Psi \rangle + \ln \langle \Psi | e^{\lambda B} | \Psi \rangle + o(X) \tag{81}$$

where $o(X)$ is to be understood order by order in $\lambda$.

This factorisation property is physically natural: as velocities of propagation should be finite, $J_0|_0^T$, for instance, should be supported on a region of length $\propto T$ around the point $x = 0$, and therefore, by clustering of the state and quantum locality (vanishing commutators of operators supported on regions at non-zero distances from each other), should separate from $Q|_{X/2}^X + J_X|_0^T$. Further, by ultra-locality of the $U(1)$ charge density,

$$[q(x), q(y)] = 0 \ \forall x, y, \tag{82}$$

and the homogeneous part of $Q|_0^{X/2}$ and $Q|_{X/2}^{X}$ should give rise to linear-in-$X$ contribution well captured by the last two terms in (79). Ultra-locality simplifies the argument, but we expect it not to be strictly necessary, as for generic local conserved densities, $[q(x), q(y)] = O(x)\delta'(x - y) + \ldots$ with higher-order derivatives of $\delta$-function involved (or the equivalent in systems on discrete space); these would give rise to additional local observables around $X/2$, which by quantum locality and clustering again, should have sub-leading contributions.

However, working out more precisely these expected properties requires some analysis.

We consider two situations: first we assume the model admits a Lieb-Robinson bound and that the state is exponentially clustering. This holds for lattice models with finite local Hilbert spaces, where operators are bounded. (Hence, for this case, the space variable $x$ takes values in $\mathbb{Z}$ and $\int dx \to \sum_x$; but this does not conceptually affect the calculations.) Second, we consider the case of the free fermion model (50); there is no Lieb-Robinson bound associated to this, as (1) operators are unbounded (for instance, the fermion density is not bounded), and (2) the dispersion relation does not admit a maximal velocity. There, we assume a weaker power law clustering. We will use different techniques in order to argue (less rigorously, and only to leading order) that the factorisation holds.

### 4.1.1 In the presence of a Lieb-Robinson bound

We assume that a Lieb-Robinson bound is available, and that the state $|\Psi\rangle$ has exponentially decaying correlations (this is a more precise version of (13)): there is $\mu > 0$ such that for every local $O, O'$, there is $C > 0$ such that

$$\langle\Psi|O(x)O'(x')|\Psi\rangle^c \le C e^{-\mu|x-x'|}. \tag{83}$$

We write

$$
\begin{aligned}
&\ln\langle\Psi|e^{\lambda(A+B)}|\Psi\rangle - \ln\langle\Psi|e^{\lambda A}|\Psi\rangle - \ln\langle\Psi|e^{\lambda B}|\Psi\rangle \\
&= \sum_{n=1}^{\infty} \frac{\lambda^n}{n!}\left(\langle\Psi|(A+B)^n|\Psi\rangle^c - \langle\Psi|A^n|\Psi\rangle^c - \langle\Psi|B^n|\Psi\rangle^c\right) \\
&= \sum_{n=1}^{\infty} \frac{\lambda^n}{n!} \sum_{\substack{E_i \in \{A,B\}\,\forall\, i=1,\ldots,n \\ \{E_i\} \neq \{A,\ldots,A\},\{B,\ldots,B\}}} \langle\Psi|E_1 \cdots E_n|\Psi\rangle^c.
\end{aligned}
\tag{84}
$$

Note how the sum in the last line keeps the appropriate operator ordering, and avoids the two configurations where only $A$'s and only $B$'s appear – these two configurations are those that are subtracted by the two terms with negative sign on the left-hand side. We use the results of [51], which show that, in the presence of a Lieb-Robinson bound and for exponentially clustering states, connected correlation functions, which may involve operators evolved to finite times, where a group of operators is far from the rest of the operators, vanish exponentially with the distance between the groups, no matter the ordering of the product of operators. As we assume that $|\Psi\rangle$ is clustering strongly enough – say exponentially – we may use these results.

In order to do so, we expand both $A$ and $B$ into their two terms in (80), and further, expand the operators $Q|_0^{X/2} = \int_0^{X/2} dx\, q(x)$ and $Q|_{X/2}^{X} = \int_{X/2}^{X} dx\, q(x)$ into their integrands. We note that in (84) there is at least one operator $A$ and one operator $B$. The idea of the argument is to take the term(s) with the "worst behaviour", that is, whose upper bound is the largest, and bound all terms by this.

Let us first take for at least one of the $A$'s the operator $J_0|_0^T$ (and the same argument will have to be done, instead, for the $B$ operator). Then, from the operator $B$, there will be terms $q(x)$ for $x > X/2$, and / or $J_X|_0^T$, and from the other possible $A$'s, there will be terms $q(x)$ for $x \in [0, X/2]$, and / or $J_0|_0^T$. The situation with the closest operators is that with $q(X/2)$ from the $B$'s, and $n-2$

operators $q(x)$'s for $x \in [0, X/2]$ from the $A$'s, distributed uniformly. In this case, there are two groups of operators a distance at least $X/(2(n-1))$, hence the correlation function is bounded by $Ce^{-\mu X/(2(n-1))}$ for some $C > 0$, and for $\mu > 0$ as above. Taking this bound, the integrals over $x$ give contributions that grow at most proportionally to $X^{n-1}$. Because for each of $A$ and $B$ we have two choices (currents or local density), and because for each $E_i$ we have two choices, this gives a total of at most $4^n$ such terms. The only term that we have not considered, for given $E_i$'s, is that where we take for all $A$'s the term $Q|_0^{X/2}$ and for all $B$'s the term $Q|_{X/2}^X$. Therefore,

$$
\left| \sum_{\substack{E_i \in \{A,B\} \, \forall i = 1,\ldots,n \\ \{E_i\} \neq \{A,\ldots,A\}, \{B,\ldots,B\}}} \langle \Psi | E_1 \cdots E_n | \Psi \rangle^c \right|
$$

$$
\leq \quad g_n(T) 4^n X^{n-1} e^{-\mu X/(2(n-1))} \quad + \quad \left| \sum_{\substack{E_i \in \{Q|_0^{X/2}, Q|_{X/2}^X\} \, \forall i = 1,\ldots,n \\ \{E_i\} \neq \{Q|_0^{X/2},\ldots,Q|_0^{X/2}\}, \{Q|_{X/2}^X,\ldots,Q|_{X/2}^X\}}} \langle \Psi | E_1 \cdots E_n | \Psi \rangle^c \right|, \quad (85)
$$

where we have indicated explicitly the dependence on $n$ and $T$ of the bounding constant $g_n(T)$.

The last term would require a more detailed analysis. Here we simply note that, for instance, $\langle \Psi | \left( Q|_0^{X/2} \right)^n | \Psi \rangle^c$ grows like $X$, and that this growth is due to the "diagonal" part of the integration of $\langle \Psi | q(x_1) \cdots q(x_n) | \Psi \rangle^c$ over $x_1, \ldots, x_n$, that is, $x_1 \approx x_2 \approx \cdots \approx x_n \approx x_c \in [0, X/2]$. For $\langle \Psi | \left( Q|_0^{X/2} \right)^{n-1} Q|_{X/2}^X | \Psi \rangle^c$, this diagonal integration gives a finite contribution because of the presence of operators on $x \in [X/2, X]$, which provide an exponential decay for $x_c \ll X/2$. For the case $n = 2$ the calculation is worked out in (105) below.

Therefore, we have found that

$$
\ln \langle \Psi | e^{\lambda(A+B)} | \Psi \rangle - \ln \langle \Psi | e^{\lambda A} | \Psi \rangle - \ln \langle \Psi | e^{\lambda B} | \Psi \rangle = \sum_{n=1}^{\infty} \frac{\lambda^n}{n!} C_n \quad (86)
$$

with

$$
|C_n| \leq g_n(T) 4^n X^{n-1} e^{-\mu X/(2(n-1))} + f_n(X) \quad (87)
$$

and $f_n(X) = \mathcal{O}(1)$ as $X \to \infty$ for all $n$. Hence, the quantity

$$
\ln \langle \Psi | e^{\lambda J_0|_0^T + \lambda Q|_0^X(0) - \lambda J_X|_0^T} | \Psi \rangle \quad (88)
$$

equals (79), up to correction terms, at order $\lambda^n$, bounded by

$$
g_n(T) \frac{4^n X^{n-1}}{n!} e^{-\mu X/(2(n-1))} + f_n(X) \quad \text{with} \quad f_n(X) = \mathcal{O}(1). \quad (89)
$$

This bound indeed satisfies (81) order by order in $\lambda$.

### 4.1.2 Infinite Lieb-Robinson velocity: free non-relativistic fermions

In order to illustrate how Factorisation Property 1 does not necessarily need a Lieb-Robinson bound and exponential clustering, we now consider the non-relativistic free fermion model of Section 3, for which there is no Lieb-Robinson bound, and, instead of the exponential decay (83), we assume power law decay.

**Assumptions.**

For simplicity, we assume that the state $|\Psi\rangle$ satisfies Wick's theorem with zero average of single-fermion fields (which is the case, for instance, for the integrable quench state of Section 3),

and that the state is translation and parity invariant. Thus we have

$$\langle\Psi|\psi^\dagger(y)\psi(z)|\Psi\rangle = C_{+-}(y-z) = C_{+-}(z-y) = \delta(y-z) - \langle\Psi|\psi(y)\psi^\dagger(z)|\Psi\rangle$$
$$\langle\Psi|\psi^\dagger(y)\psi^\dagger(z)|\Psi\rangle = C_{++}(y-z) = C_{++}(z-y) \tag{90}$$
$$\langle\Psi|\psi(y)\psi(z)|\Psi\rangle = C_{--}(y-z) = C_{++}^*(y-z).$$

We further impose conditions on these functions as follows. First, there is power law decay: there is $C > 0$ and $\epsilon > 0$ such that for every $x$,

$$|C_{\eta\eta'}(x)| \le \frac{C}{|x|^{1+\epsilon}}, \quad \eta, \eta' \in \{\pm\}. \tag{91}$$

Second, we require conditions on the short-distance behaviour of these functions. We assume that $C_{\eta\eta'}(x)$ is twice continuously differentiable for every $x$, and further that there is $r > 0, A_0 > 0$ and bounded functions $A_1(y) > 0, A_2(y) > 0$ such that for all $|x| \le r$ and all $y \in \mathbb{R}$

$$|C_{\eta\eta'}(x)| \le A_0, \quad |C_{\eta\eta'}(y+x) - C_{\eta\eta'}(y) - xA_1(y)| \le x^2 A_2(y), \quad \eta, \eta' \in \{\pm\}. \tag{92}$$

Recall that, for this model, the charge $Q$ is taken to be the fermion particle number for simplicity. In particular, by Wick's theorem, we have

$$\langle\Psi|q(x)q(x')|\Psi\rangle^c \le \frac{2C^2}{|x-x'|^{2+2\epsilon}}. \tag{93}$$

**Derivation.**

For what follows, it is useful to re-write $j(x,t)$ using the real-space fermionic fields, yielding

$$j(x,t) = \mathrm{i}\left((\partial_x\psi^\dagger(x,t))\psi(x,t) - \psi^\dagger(x,t)\partial_x\psi(x,t)\right) = \int \mathrm{d}y\mathrm{d}z\,\psi^\dagger(y)F(y-x,z-x,t)\psi(z), \tag{94}$$

where

$$F(x,y,t) = \frac{1}{4\pi^2}\int \mathrm{d}k\mathrm{d}p\,e^{\mathrm{i}(kx-py)+\mathrm{i}(E_k-E_p)t}(k+p) = -\frac{e^{-\mathrm{i}\frac{x^2-y^2}{4t}}(x+y)}{8\pi t^2}, \tag{95}$$

with

$$F(x,y,0) = \mathrm{i}\delta'(x)\delta(y) - \mathrm{i}\delta(x)\delta'(y). \tag{96}$$

The time integration of the current can be performed as well, and in order to proceed we use the following expression for the integrated current

$$\tilde{J} = \int_0^T \mathrm{d}t\,j(0,t) = \lim_{\epsilon\to 0}\int_\epsilon^T \mathrm{d}t\int \mathrm{d}y\int \mathrm{d}z\,F(y,z,t)\psi^\dagger(y)\psi(z) = \lim_{\epsilon\to 0}\int \mathrm{d}y\int \mathrm{d}z\,G(y,z,T)_\epsilon\psi^\dagger(y)\psi(z), \tag{97}$$

where

$$G(y,z,T)_\epsilon = -\mathrm{i}\frac{e^{-\mathrm{i}\frac{y^2-z^2}{4T}} - e^{-\mathrm{i}\frac{y^2-z^2}{4\epsilon}}}{2\pi(y-z)}, \tag{98}$$

Additionally, $G_\epsilon$ has the property that $G(y,z,t)_\epsilon = G(z,y,t)_\epsilon^*$, and it is possible to perform the $\epsilon \to 0$ limit resulting in

$$\lim_{\epsilon\to 0}G(y,z,T)_\epsilon = -\mathrm{i}\frac{1}{2\pi}\mathrm{P}\frac{e^{-\mathrm{i}\frac{y^2-z^2}{4T}}}{y-z} + \frac{1}{2}\delta(y-z)\,\mathrm{sgn}(y). \tag{99}$$

although when the product of two or more $G_\epsilon$ functions are regarded, this identity cannot be directly used.

Turning to the main task of this subsection, it is convenient to organise the calculation by using the Zassenhaus formula to write

$$\ln\langle\Psi|e^{\lambda A+\lambda B}|\Psi\rangle = \ln\langle\Psi|e^{\lambda A}e^{\lambda B}e^{-\frac{\lambda^2}{2}[A,B]}e^{\frac{\lambda^3}{6}(2[B,[A,B]]+[A,[A,B]])}\times(...)|\Psi\rangle\,. \tag{100}$$

We want to demonstrate that

$$\begin{aligned}\ln\langle\Psi|e^{\lambda A+\lambda B}|\Psi\rangle =\,&\ln\langle\Psi|e^{\lambda A}|\Psi\rangle+\ln\langle\Psi|e^{\lambda B}|\Psi\rangle+\ln\langle\Psi|e^{-\frac{\lambda^2}{2}[A,B]}|\Psi\rangle\\&+\ln\langle\Psi|e^{\frac{\lambda^3}{6}(2[B,[A,B]]+[A,[A,B]])}|\Psi\rangle+\dots,\end{aligned} \tag{101}$$

and that only the first two terms on the right hand side of (101) are the leading ones (we must show that the other ones decay faster than $X$ as $X\to\infty$). Demonstrating this fact (Factorisation Property 1) is achieved via a series expansion: we consider the expansion up to second order in $\lambda$ for simplicity.

More precisely, we first spell out the expansion of the l.h.s. of (101) and compare it with the expansion of its r.h.s., showing that they are identical for any finite $T$ as $X\to\infty$ up to $o(X)$ terms. As a second step, we show that the r.h.s. of (101) indeed equals $\ln\langle\Psi|e^{\lambda A}|\Psi\rangle+\ln\langle\Psi|e^{\lambda B}|\Psi\rangle$ under the aforementioned limit.

Therefore, considering first the l.h.s. of (101), we have that

$$\begin{aligned}\ln\langle\Psi|e^{\lambda(A+B)}|\Psi\rangle =\,&\lambda\langle\Psi|(A+B)|\Psi\rangle+\frac{\lambda^2}{2}(\langle\Psi|(A+B)^2|\Psi\rangle-\langle\Psi|(A+B)|\Psi\rangle^2)+\mathcal{O}(\lambda^3)\\=\,&\lambda\langle\Psi|A|\Psi\rangle+\frac{\lambda^2}{2}(\langle\Psi|A^2|\Psi\rangle-\langle\Psi|A|\Psi\rangle^2)+\lambda\langle\Psi|B|\Psi\rangle+\frac{\lambda^2}{2}(\langle\Psi|B^2|\Psi\rangle-\langle\Psi|B|\Psi\rangle^2)\\&+\lambda^2(\langle\Psi|AB|\Psi\rangle-\langle\Psi|A|\Psi\rangle\langle\Psi|B|\Psi\rangle)-\frac{\lambda^2}{2}\langle\Psi|[A,B]|\Psi\rangle+\mathcal{O}(\lambda^3)\,.\end{aligned} \tag{102}$$

On the other hand, the r.h.s. reads

$$\begin{aligned}&\ln\langle\Psi|e^{\lambda A}|\Psi\rangle+\ln\langle\Psi|e^{\lambda B}|\Psi\rangle+\ln\langle\Psi|e^{-\frac{\lambda^2}{2}[A,B]}|\Psi\rangle+\mathcal{O}(\lambda^3)\\=\,&\lambda\langle\Psi|A|\Psi\rangle+\frac{\lambda^2}{2}(\langle\Psi|A^2|\Psi\rangle-\langle\Psi|A|\Psi\rangle^2)+\lambda\langle\Psi|B|\Psi\rangle+\frac{\lambda^2}{2}(\langle\Psi|B^2|\Psi\rangle-\langle\Psi|B|\Psi\rangle^2)\\&-\frac{\lambda^2}{2}\langle\Psi|[A,B]|\Psi\rangle+\mathcal{O}(\lambda^3)\,.\end{aligned} \tag{103}$$

In order to show their equality, one must demonstrate that $(\langle\Psi|A\,B|\Psi\rangle-\langle\Psi|A|\Psi\rangle\langle\Psi|B|\Psi\rangle)$ is $o(X)$. We first deal with the integrated charges in $A$ and $B$ (cf. (80)) and we check that

$$\langle\Psi|Q|_0^{X/2}Q|_{X/2}^{X}|\Psi\rangle-\langle\Psi|Q|_0^{X/2}|\Psi\rangle\langle\Psi|Q|_{X/2}^{X}|\Psi\rangle=o(X) \tag{104}$$

holds. Its analysis is completely analogous to the case of finite Lieb-Robinson velocity. In particular, the second expression in (104) equals $(X/2)^2\langle\Psi|q(0)|\Psi\rangle^2$ due to translation invariance. For the first term, we have that

$$\begin{aligned}&\int_0^{X/2}\mathrm{d}x\int_{X/2}^{X}\mathrm{d}y\,\langle\Psi|q(x)q(y)|\Psi\rangle=\\&\int_0^{X/2}\mathrm{d}x\int_{X/2}^{X}\mathrm{d}y\,\langle\Psi|q(x)|\Psi\rangle\langle\Psi|q(y)|\Psi\rangle+\mathcal{O}(|x-y|^{-1-\epsilon})=(X/2)^2\langle\Psi|q(0)|\Psi\rangle^2+o(X)\end{aligned} \tag{105}$$

due to the strong cluster property and translation invariance of the state $|\Psi\rangle$, and hence (104) indeed holds.

To continue with the analysis of $(\langle\Psi|A\,B|\Psi\rangle - \langle\Psi|A|\Psi\rangle\langle\Psi|B|\Psi\rangle)$, we have to consider three more terms, namely 2-pt functions $\langle\Psi|J_0|_0^T Q|_{X/2}^X(0)|\Psi\rangle$, $-\langle\Psi|Q|_0^{X/2}(0)J_X|_0^T|\Psi\rangle$ and $\langle\Psi|J_0|_0^T J_X|_0^T|\Psi\rangle$ (since $\langle\Psi|J_0|_0^T|\Psi\rangle = 0$) and we have to show that they are sub-leading for $X \gg 1$ and finite $T$. To show that $(\langle\Psi|A\,B|\Psi\rangle - \langle\Psi|A|\Psi\rangle\langle\Psi|B|\Psi\rangle)$ is $o(X)$ and we start by studying

$$
\langle\Psi|J_0|_0^T J_X|_0^T|\Psi\rangle = \langle\Psi|J_0|_0^T J_X|_0^T|\Psi\rangle^c =
$$
$$
= \int_0^T \mathrm{d}t \int_0^T \mathrm{d}t' \int \mathrm{d}y\,\mathrm{d}z\,\mathrm{d}y'\,\mathrm{d}z' F(y,z,t)F(y'-X,z'-X,t')\langle\Psi|\psi^\dagger(y)\psi(z),\psi^\dagger(y')\psi(z')|\Psi\rangle^c
$$
$$
= \lim_{\epsilon\to 0}\int \mathrm{d}y\,\mathrm{d}z\,\mathrm{d}y'\,\mathrm{d}z' G(y,z,T)_\epsilon G(y'-X,z'-X,T)_\epsilon\langle\Psi|\psi^\dagger(y)\psi(z),\psi^\dagger(y')\psi(z')|\Psi\rangle^c
$$

$$(106)$$

where on the right-hand side, we put a comma in order to specify the connected correlation function taken. Now, we use the simplifying feature that the state satisfies Wick's theorem and is translation invariant, see (90). According to Wick's theorem, we may rewrite the integrated current 2-pt function as

$$
\langle\Psi|J_0|_0^T J_X|_0^T|\Psi\rangle = \langle\Psi|J_0|_0^T J_X|_0^T|\Psi\rangle^c =
$$
$$
= \lim_{\epsilon\to 0}\int \mathrm{d}y\,\mathrm{d}z\,\mathrm{d}y'\,\mathrm{d}z' G(y,z,T)_\epsilon G(y'-X,z'-X,T)_\epsilon\big[C_{+-}(y-z')(\delta(y'-z)-C_{+-}(y'-z'))
$$
$$
-C_{++}(y-y')C_{--}(z-z')\big]
$$
$$
= J_{X,1}^2 + J_{X,2}^2 + J_{X,3}^2\,.
$$

$$(107)$$

Let us start by the evaluation of $J_{X,1}^2$. We employ two sets of changes of variables: first $y' \to y'+X$, $z' \to z'+X$ yielding

$$
J_{X,1}^2 = \lim_{\epsilon\to 0}\int \mathrm{d}y\,\mathrm{d}z\,\mathrm{d}y'\,\mathrm{d}z' G(y,z,T)_\epsilon G(y',z',T)_\epsilon C_{+-}(y-z'-X)\delta(z-y'-X)
$$
$$
= \lim_{\epsilon\to 0}\int \mathrm{d}y\,\mathrm{d}z\,\mathrm{d}w\, G(y,z,T)_\epsilon G(z-X,w,T)_\epsilon C_{+-}(y-w-X) \qquad (108)
$$
$$
= \lim_{\epsilon\to 0}\int \mathrm{d}y\,\mathrm{d}z\,\mathrm{d}w\, G(y,z+X/2,T)_\epsilon G(z-X/2,w,T)_\epsilon C_{+-}(y-w-X)
$$

and then in the second line we renamed $z'$ to $w$ and afterwards performed the $z \to z+X/2$ change of variable as well. Now introducing the variables

$$
\kappa_1 = y-z-X/2,\quad \kappa_2 = z-w-X/2,\quad \sigma = y+w \qquad (109)
$$

accounting for a Jacobian $1/2$, as well as the notation for function $G_\epsilon$ written in terms of these new variables as

$$
\tilde{G}(\kappa,\sigma,T)_\epsilon = -\mathrm{i}\frac{e^{-\mathrm{i}\frac{\kappa\sigma}{4T}} - e^{-\mathrm{i}\frac{\kappa\sigma}{4\epsilon}}}{2\pi\kappa}\,, \qquad (110)
$$

we have that

$$
\begin{aligned}
J_{X,1}^2 &= \lim_{\epsilon \to 0} \frac{1}{2} \int d\kappa_1 \, d\kappa_2 \, d\sigma \, \tilde{G}(\kappa_1, \sigma + \kappa_2 + X, T)_\epsilon \, \tilde{G}(\kappa_2, \sigma - \kappa_1 - X, T)_\epsilon \, C_{+-}(\kappa_1 + \kappa_2) \\
&= \lim_{\epsilon \to 0} \frac{1}{2} \int d\kappa_1 \, d\kappa_2 \, d\sigma \, \tilde{G}(\kappa_1, \sigma + \kappa_2 + X, T)_\epsilon \, \tilde{G}(\kappa_2, \sigma - \kappa_1 - X, T)_\epsilon \, C_{+-}(\kappa_1 + \kappa_2) \\
&= \lim_{\epsilon \to 0} 2 \int d\kappa_1 \, d\kappa_2 \left( \frac{\delta\left(\frac{\kappa_1}{\epsilon} + \frac{\kappa_2}{T}\right) e^{i\left(\frac{\kappa_2(\kappa_1 + X)}{4T} - \frac{\kappa_1(\kappa_2 + X)}{4\epsilon}\right)}}{2\pi \kappa_1 \kappa_2} + \frac{\delta\left(\frac{\kappa_1}{T} + \frac{\kappa_2}{\epsilon}\right) e^{-i\left(\frac{\kappa_1(\kappa_2 + X)}{4T} - \frac{\kappa_2(\kappa_1 + X)}{4\epsilon}\right)}}{2\pi \kappa_1 \kappa_2} \right. \\
&\qquad\qquad\qquad \left. - \frac{\delta\left(\frac{\kappa_1 + \kappa_2}{T}\right) e^{-i\frac{X(\kappa_1 - \kappa_2)}{4T}}}{2\pi \kappa_1 \kappa_2} - \frac{\delta\left(\frac{\kappa_1 + \kappa_2}{\epsilon}\right) e^{-i\frac{X(\kappa_1 - \kappa_2)}{4\epsilon}}}{2\pi \kappa_1 \kappa_2} \right) C_{+-}(\kappa_1 + \kappa_2),
\end{aligned}
\tag{111}
$$

since the integration wrt. $\sigma$ can be carried out explicitly, resulting in Dirac-$\delta$ functions. Taking them into account, we end up with a single variable integral

$$
\begin{aligned}
J_{X,1}^2 &= \lim_{\epsilon \to 0} 2 \int d\kappa \, T \, \frac{C_{+-}(0) - C_{+-}\left(\kappa - \kappa\frac{\epsilon}{T}\right) e^{-i\left(\frac{\kappa^2}{4T} - \frac{\kappa^2 \epsilon}{4T^2}\right)}}{2\pi \kappa^2} e^{-i\frac{\kappa X}{2T}} \\
&\quad + \lim_{\epsilon \to 0} 2 \int d\kappa \, \epsilon \, \frac{C_{+-}(0) - C_{+-}\left(\kappa - \kappa\frac{T}{\epsilon}\right) e^{-i\left(\frac{\kappa^2}{4\epsilon} - \frac{\kappa^2 T}{4\epsilon^2}\right)}}{2\pi \kappa^2} e^{-i\frac{\kappa X}{2\epsilon}} \\
&= c + c^*,
\end{aligned}
\tag{112}
$$

where it is easy to see that the two terms are complex conjugates of each other, after changing integration variables in the second line as $\kappa \to -\kappa\epsilon/T$. Now we can send $\epsilon$ to zero inside the integrand for $c$. This holds by the bounded convergence theorem, and because the integrand is bounded by a function that is integrable in $\kappa$, uniformly for $\epsilon$ small enough. The latter is immediate by using (91) and the second bound of (92), and separating the integral into $\kappa \leq r$ and $\kappa > r$ (here we replace the regularisation parameter $\epsilon \to \epsilon'$ as $\epsilon$ is used in the bound (91)):

$$
\left| \frac{C_{+-}(0) - C_{+-}\left(\kappa - \kappa\frac{\epsilon'}{T}\right) e^{-i\left(\frac{\kappa^2}{4T} - \frac{\kappa^2 \epsilon'}{4T^2}\right)}}{\kappa^2} e^{-i\frac{\kappa X}{2T}} \right|
\tag{113}
$$

$$
\leq \left(1 - \frac{\epsilon'}{T}\right)^2 A_2(0) + |C_{+-}(0)| \frac{\left|1 - e^{-i\left(\frac{\kappa^2}{4T} - \frac{\kappa^2 \epsilon'}{4T^2}\right)}\right|}{\kappa^2} \ (\kappa \leq r), \quad \frac{|C_{+-}(0)|}{\kappa^2} + \frac{1}{(1 - \frac{\epsilon'}{T})^{1+\epsilon}} \frac{C}{\kappa^{3+\epsilon}} \ (\kappa > r).
$$

This way we arrive at

$$
c = 2T \int d\kappa \, \frac{C_{+-}(0) - C_{+-}(\kappa) e^{-i\frac{\kappa^2}{4T}}}{2\pi \kappa^2} e^{-i\frac{\kappa X}{2T}}
\tag{114}
$$

This is the Fourier transform of the function $f(\kappa) = \frac{C_{+-}(0) - C_{+-}(\kappa) e^{-i\frac{\kappa^2}{4T}}}{2\pi \kappa^2}$, with $X$ the Fourier parameter. As we have assumed that $C_{+-}(\kappa)$ is twice continuously differentiable, and as it is an even function, $f(\kappa)$ is continuous everywhere. Further, by the bound above, it is integrable. Therefore by a general result of Fourier analysis, $c$ decays as $X \to \infty$, and we find

$$
\lim_{X \to \infty} J_{X,1}^2 = 0.
\tag{115}
$$

We now continue with the evaluation of $J_{X,2}^2$. We employ a change of variables: first $y' \to y'+X$, $z' \to z'+X$ yielding

$$J_{X,2}^2 = -\lim_{\epsilon \to 0} \int dy \, dz \, dy' \, dz' \, G(y,z,T)_\epsilon G(y',z',T)_\epsilon C_{+-}(y-z'-X) C_{+-}(z-y'-X) \quad (116)$$

and afterwards we introduce the variables

$$\kappa_1 = y - z, \quad \kappa_2 = y' - z', \quad \kappa_3 = z - y', \quad \sigma = z + y' \quad (117)$$

accounting for a Jacobian $\mathrm{abs}(-1/2)$. Via (110) $J_{X,2}^2$ can now be rewritten as we have that

$$J_{X,2}^2 = -\lim_{\epsilon \to 0} \frac{1}{2} \int d\kappa_1 \, d\kappa_2 \, d\kappa_3 \, d\sigma \, \tilde{G}(\kappa_1, \kappa_1 + \kappa_3 + \sigma, T)_\epsilon \tilde{G}(\kappa_2, \sigma - \kappa_2 - \kappa_3, T)_\epsilon \times$$
$$\times C_{+-}(\kappa_1 + \kappa_2 + \kappa_3 - X) C_{+-}(\kappa_3 - X), \quad (118)$$

and just like in the previous case, the integration wrt. $\sigma$ can be performed yielding Dirac-$\delta$-s:

$$J_{X,2}^2 = -\lim_{\epsilon \to 0} 2 \int d\kappa_1 \, d\kappa_2 \, d\kappa_3 \, C_{+-}(\kappa_1 + \kappa_2 + \kappa_3 - X) C_{+-}(\kappa_3 - X) \left( \frac{\delta\left(\frac{\kappa_1}{\epsilon} + \frac{\kappa_2}{T}\right) e^{i\left(\frac{\kappa_2(\kappa_2+\kappa_3)}{4T} - \frac{\kappa_1(\kappa_1+\kappa_3)}{4\epsilon}\right)}}{2\pi \kappa_1 \kappa_2} \right.$$

$$\left. + \frac{\delta\left(\frac{\kappa_1}{T} + \frac{\kappa_2}{\epsilon}\right) e^{-i\left(\frac{\kappa_1(\kappa_1+\kappa_2)}{4T} - \frac{\kappa_2(\kappa_2+\kappa_3)}{4\epsilon}\right)}}{2\pi \kappa_1 \kappa_2} - \frac{\delta\left(\frac{\kappa_1+\kappa_2}{T}\right) e^{-i\frac{(\kappa_1-\kappa_2)(\kappa_1+\kappa_2+\kappa_3)}{4T}}}{2\pi \kappa_1 \kappa_2} - \frac{\delta\left(\frac{\kappa_1+\kappa_2}{\epsilon}\right) e^{-i\frac{(\kappa_1-\kappa_2)(\kappa_1+\kappa_2+\kappa_3)}{4\epsilon}}}{2\pi \kappa_1 \kappa_2} \right)$$

$$= -\lim_{\epsilon \to 0} 2 \int d\kappa_1 d\kappa_3 \, T \frac{C_{+-}(\kappa_3-X) - C_{+-}\left(\kappa_1 - \kappa_1 \frac{\epsilon}{T} + \kappa_3 - X\right) e^{-i\left(\frac{\kappa_1^2}{4T} - \frac{\kappa_1^2 \epsilon}{4T^2}\right)}}{2\pi \kappa_1^2} C_{+-}(\kappa_3-X) e^{-i\frac{\kappa_1 \kappa_3}{2T}}$$

$$- \lim_{\epsilon \to 0} 2 \int d\kappa_1 d\kappa_3 \, \epsilon \frac{C_{+-}(\kappa_3-X) - C_{+-}\left(\kappa_1 - \kappa_1 \frac{T}{\epsilon} + \kappa_3 - X\right) e^{-i\left(\frac{\kappa_1^2}{4\epsilon} - \frac{\kappa_1^2 T}{4\epsilon^2}\right)}}{2\pi \kappa_1^2} C_{+-}(\kappa_3-X) e^{-i\frac{\kappa_1 \kappa_3}{2\epsilon}}$$

$$= c' + c'^*, \quad (119)$$

where the two terms are again complex conjugates of each other, which can be seen after changing integration variables in the second line as $\kappa_1 \to -\kappa_1 \epsilon / T$. Therefore, we only have to deal with

$$c' = -\lim_{\epsilon \to 0} 2 \int d\kappa_1 d\kappa_3 \, T \frac{C_{+-}(\kappa_3-X) - C_{+-}\left(\kappa_1 - \kappa_1 \frac{\epsilon}{T} + \kappa_3 - X\right) e^{-i\left(\frac{\kappa_1^2}{4T} - \frac{\kappa_1^2 \epsilon}{4T^2}\right)}}{2\pi \kappa_1^2} C_{+-}(\kappa_3-X) e^{-i\frac{\kappa_1 \kappa_3}{2T}}$$

$$= -2T \int d\kappa_1 d\kappa_3 \frac{C_{+-}(\kappa_3-X) - C_{+-}(\kappa_1 + \kappa_3 - X) e^{-i\frac{\kappa_1^2}{4T}}}{2\pi \kappa_1^2} C_{+-}(\kappa_3-X) e^{-i\frac{\kappa_1 \kappa_3}{2T}}$$

$$= -2T \int d\kappa_1 d\kappa_3 \frac{C_{+-}(\kappa_3) - C_{+-}(\kappa_1 + \kappa_3) e^{-i\frac{\kappa_1^2}{4T}}}{2\pi \kappa_1^2} C_{+-}(\kappa_3) e^{-i\frac{\kappa_1(\kappa_3+X)}{2T}}$$

$$= -2T \int d\kappa_1 \int_{>0} d\kappa_3 \frac{2\cos\left(\frac{\kappa_1 \kappa_3}{2T}\right) C_{+-}(\kappa_3) - \left(C_{+-}(\kappa_3 + \kappa_1) e^{-i\frac{\kappa_1 \kappa_3}{2T}} + C_{+-}(\kappa_3 - \kappa_1) e^{i\frac{\kappa_1 \kappa_3}{2T}}\right)}{2\pi \kappa_1^2} \times$$

$$\times e^{-i\frac{\kappa_1^2}{4T}} C_{+-}(\kappa_3) e^{-i\frac{\kappa_1 X}{2T}}$$

$$(120)$$

where in the second line we sent $\epsilon$ to zero[4], and in the 3rd line we performed a change of variables shifting $\kappa_3$.

Consider the function

$$f(\kappa_3) = \int d\kappa_1 \frac{2\cos(\frac{\kappa_1\kappa_3}{2T})C_{+-}(\kappa_3) - \left(C_{+-}(\kappa_3+\kappa_1)e^{-i\frac{\kappa_1\kappa_3}{2T}} + C_{+-}(\kappa_3-\kappa_1)e^{i\frac{\kappa_1\kappa_3}{2T}}\right)e^{-i\frac{\kappa_1^2}{4T}}}{2\pi\kappa_1^2} \times$$
$$\times C_{+-}(\kappa_3)e^{-i\frac{\kappa_1 X}{2T}}.$$
(121)

Note that because $C_\pm(\kappa_3+\kappa_1)$ is twice continuously differentiable in $\kappa_1$, then the integrand is finite and continuous at $\kappa_1 = 0$. Using (91) and (92), and in particular the constants $C > 0$, $r > 0$, $\epsilon > 0$, $A_0 > 0$ and the bounded functions $A_1(y)$, $A_2(y)$ involved there, we find that for every $\kappa_3$,

$$|f(\kappa_3)| \leq \frac{|C_{+-}(\kappa_3)|}{\pi}\left(\int_{|\kappa_1|>r} d\kappa_1 \frac{|C_{+-}(\kappa_3)|}{\kappa_1^2} + \frac{1}{2}\sum_\pm \int_{\substack{|\kappa_1|>r \\ |\kappa_3\pm\kappa_1|>r}} d\kappa_1 \frac{C}{|\kappa_3\pm\kappa_1|^{1+\epsilon}\kappa_1^2}\right.$$

$$+ |C_{+-}(\kappa_3)|\int_{|\kappa_1|\leq r} d\kappa_1 \frac{|1-e^{-i\frac{\kappa_1^2}{4T}}|}{\kappa_1^2} + A_1(\kappa_3)\int_{|\kappa_1|\leq r} d\kappa_1 \frac{|\sin(\frac{\kappa_1\kappa_3}{2T})|}{|\kappa_1|} + 2rA_2(\kappa_3)$$

$$\left.+ \frac{1}{2}\sum_\pm \int_{\substack{|\kappa_1|>r \\ |\kappa_3\pm\kappa_1|\leq r}} \frac{A_0}{\kappa_1^2}\right)$$

$$\leq \frac{|C_{+-}(\kappa_3)|}{\pi}\left(\left(|C_{+-}(\kappa_3)| + \frac{C}{r^{1+\epsilon}} + A_0\right)\int_{|\kappa_1|>r} d\kappa_1 \frac{1}{\kappa_1^2}\right.$$

$$\left.+ |C_{+-}(\kappa_3)|\int_{|\kappa_1|\leq r} d\kappa_1 \frac{|1-e^{-i\frac{\kappa_1^2}{4T}}|}{\kappa_1^2} + A_1(\kappa_3)\int d\kappa_1 \frac{|\sin(\frac{\kappa_1}{2T})|}{|\kappa_1|} + 2rA(\kappa_3)\right).$$
(122)

Because $|C_{+-}(\kappa_3)|$ is integrable on $\kappa_3$, we find that $f(\kappa_3)$ is uniformly bounded by an integrable function of $\kappa_3$. Further, because $C_{+-}(\kappa_3+\kappa_1)$ is twice continuously differentiable in $\kappa_1$ for all $\kappa_3$, then for every $\kappa_3$ the expression (121) is the Fourier transform of a continuous function, and hence vanishes as $X \to \infty$ for every $\kappa_3$. By the bounded convergence theorem, we conclude that

$$\lim_{X\to\infty} c' = -2T \lim_{X\to\infty} \int_{>0} d\kappa_3 f(\kappa_3) = -2T\int_{>0} d\kappa_3 \lim_{X\to\infty} f(\kappa_3) = 0$$
(123)

whence $\lim_{X\to\infty} J_{X,2}^2 = 0$.

Finally, the treatment of $J_{X,3}^2$ is completely analogous to that of $J_{X,2}^2$, merely the correlation functions $C_{+-}(x)C_{+-}(y)$ are to be replaced by $C_{++}(x)C_{--}(y)$, hence we find $\lim_{X\to\infty} J_{X,3}^2 = 0$.

In summary, we have demonstrated that

$$\lim_{X\to\infty}\langle\Psi|J_0|_0^T J_X|_0^T|\Psi\rangle = \lim_{X\to\infty}\langle\Psi|J_0|_0^T J_X|_0^T|\Psi\rangle^c = \lim_{X\to\infty}(J_{X,1}^2 + J_{X,2}^2 + J_{X,3}^2) = 0, \quad \forall\, 0 < T < \infty.$$
(124)

As a next step we move onto the other two multi-point functions, and in particular the combination $\langle\Psi|J_0|_0^T Q|_{X/2}^X(0)|\Psi\rangle - \langle\Psi|Q|_0^{X/2}(0)J_X|_0^T|\Psi\rangle$ as it turns out to be useful to group these terms.

---

[4]We note that by a rescaling of $\kappa_1$, the parameter $\epsilon$ simply rescales the parameters of the integrand – the various occurrences of $T$ in this expression – by a scaling factor that tends to 1 as $\epsilon \to 0$. In order to take the limit $\epsilon \to 0$ inside the integral, we have used the bounded convergence theorem, along with the bounds established below, which are uniform in these parameters.

For the second term, we use the following manipulations and write

$$
-\langle\Psi|Q|_0^{X/2}(0)J_X|_0^T|\Psi\rangle =
$$
$$
=-\int_0^T dt\int_0^{X/2} dy\,\langle\Psi|\psi^\dagger(y)\psi(y)i\big[(\partial_x\psi^\dagger(X,t))\psi(X,t)-\psi^\dagger(X,t)\partial_x\psi(X,t)\big]|\Psi\rangle
$$
$$
=\int_0^T dt\int_0^{X/2} dy\,\langle\Psi|\psi^\dagger(-y)\psi(-y)i\big[(\partial_x\psi^\dagger(-X,t))\psi(-X,t)-\psi^\dagger(-X,t)\partial_x\psi(-X,t)\big]|\Psi\rangle
$$
$$
=\int_0^T dt\int_{-X/2}^0 dy\,\langle\Psi|\psi^\dagger(y)\psi(y)i\big[(\partial_x\psi^\dagger(-X,t))\psi(-X,t)-\psi^\dagger(-X,t)\partial_x\psi(-X,t)\big]|\Psi\rangle
$$
$$
=-\int_0^T dt\int_{X/2}^X dy\,\langle\Psi|\psi^\dagger(y)\psi(y)i\big[(\partial_x\psi^\dagger(0,t))\psi(0,t)-\psi^\dagger(0,t)\partial_x\psi(0,t)\big]|\Psi\rangle.
$$

$$(125)$$

In the above, on the second line we applied a spatial parity transformation $\hat{P}$, which we defined as $\hat{P}\psi(x)\hat{P}=-\psi(-x)$ and $\hat{P}\partial_x\psi(x)\hat{P}=\partial_x\psi(-x)$, and we exploited that the state is invariant under this transformation. In the fifth line translational invariance of the state was exploited. This way, we showed that

$$
\langle\Psi|J_0|_0^T Q|_{X/2}^X(0)|\Psi\rangle-\langle\Psi|Q|_0^{X/2}(0)J_X|_0^T|\Psi\rangle=\langle\Psi|\Big[J_0|_0^T,Q|_{X/2}^X(0)\Big]|\Psi\rangle \tag{126}
$$

which is a term that also needs to be analysed as it appears in Zassenhaus contribution $[A,B]$ as well.

Now we demonstrate the last step, that is, that the relevant Zassenhaus terms $\langle\Psi|[A,B]|\Psi\rangle$ are also $o(X)$. This necessitates the study of the current-current and and the current-charge commutators. Focusing on the current-charge commutator $[j(0,t),q(x,0)]$, we may write that

$$
[j(0,t),q(x,0)]=-\frac{1}{4\pi^2}\int dk\,dp\,dk'\,dp'\,e^{-i(k'-p')x+i(E_k-E_p)t}(k+p)[a_{k'}^\dagger a_{p'},a_k^\dagger a_p]
$$
$$
=\int dy\,\psi^\dagger(y)F(y,x,t)\psi(x)-\int dy\,\psi^\dagger(x)F(x,y,t)\psi(y), \tag{127}
$$

where $F(x,y,t)$ is defined in (95). Integrating this expression we end up with

$$
\Big[J_0|_0^T,Q_{X/2}^X(0)\Big]=\int_0^T dt\int_{X/2}^X dx\int dy\,\psi^\dagger(y)F(y,x,t)\psi(x)-\psi^\dagger(x)F(x,y,t)\psi(y). \tag{128}
$$

We perform the $T$ integral using (99) to get

$$
\langle\Psi|\Big[J_0|_0^T,Q_{X/2}^X(0)\Big]|\Psi\rangle=\frac{i}{\pi}\int_{X/2}^X dx\int dy\,C_{+-}(x-y)\mathrm{P}\frac{\cos\left(\frac{x^2-y^2}{4T}\right)}{x-y} \tag{129}
$$

where we used symmetry of $C_{+-}(x-y)$ under $x\leftrightarrow y$ and the fact that the anti-symmetrisation of $\delta(x-y)\mathrm{sgn}(x)$ is $\frac{1}{2}\delta(x-y)(\mathrm{sgn}(x)-\mathrm{sgn}(y))=0$. This is

$$
\langle\Psi|\Big[J_0|_0^T,Q_{X/2}^X(0)\Big]|\Psi\rangle=\frac{i}{\pi}\int_{X/2}^X dx\left(\int_X^\infty+\int_{-\infty}^{X/2}\right)dy\,C_{+-}(x-y)\frac{\cos\left(\frac{x^2-y^2}{4T}\right)}{x-y} \tag{130}
$$

where we used the fact that the part with $y$ integral on $[X/2,X]$ vanishes by anti-symmetry under $x\leftrightarrow y$. The principal value is not required anymore, because the singularity at $x=y$ is reached

only at two points $(x,y) = (X,X)$ and $(x,y) = (X/2,X/2)$, hence it is integrable under the remaining double integral. By (91), (92) there is $C'$ such that

$$|C_{+-}(x-y)| \leq \frac{C'}{(|x-y|+1)^{1+\epsilon}}. \tag{131}$$

Therefore

$$\left| \langle \Psi | \left[ J_0 \big|_0^T, Q_{X/2}^X(0) \right] | \Psi \rangle \right| \leq \frac{C'}{\pi} \int_{X/2}^X dx \left( \int_X^\infty + \int_{-\infty}^{X/2} \right) dy \frac{1}{(|x-y|+1)^{1+\epsilon}|x-y|}. \tag{132}$$

The integration over $y$ is integrable at $\pm\infty$ and the resulting integral over $X$ vanishes as $X^{-\epsilon}$. We conclude that

$$\lim_{X\to\infty} \langle \Psi | \left[ J_0 \big|_0^T, Q_{X/2}^X(0) \right] | \Psi \rangle = 0. \tag{133}$$

For the current-current commutator, we have already shown the $o(X)$ behaviour when its separate 2-pt functions are considered, but for the commutator we can also argue as follows: on a parity- and translation-invariant, $\langle \Psi | J_0 \big|_0^T J_X \big|_0^T | \Psi \rangle = \langle \Psi | J_{-X} \big|_0^T J_0 \big|_0^T | \Psi \rangle = \langle \Psi | J_X \big|_0^T J_0 \big|_0^T | \Psi \rangle$, where we use the fact that the current is parity odd (this works also for parity even). Thus

$$\langle \Psi | \left[ J_0 \big|_0^T, J_X \big|_0^T \right] | \Psi \rangle = 0. \tag{134}$$

**Differentiability requirement and filling function.**

Finally, we briefly comment on the requirement that $C_{+-}$ (cf. (90)) shall be two-times differentiable in order for $J_{X,1}^2$ to be finite. To do so, for physical interpretation, let us study the temporal behaviour of the 2nd cumulant of the current in a GGE, that is, when Wick's theorem holds and only the $\langle \psi^\dagger(x)\psi(y) \rangle$ 2-pt function is non-vanishing. In particular, setting $X = 0$ in (114) and (120), we can easily see that

$$\begin{aligned}
\lim_{T\to\infty} T^{-1} \langle (J_0 \big|_0^T)^2 \rangle_{\text{GGE}} = {} & \lim_{T\to\infty} 4 \int d\kappa \, \frac{C_{+-}(0) - C_{+-}(\kappa)\cos(\frac{\kappa^2}{4T})}{2\pi\kappa^2} \\
& - 2 \lim_{T\to\infty} \int d\kappa_1 d\kappa_3 \, \frac{C_{+-}(\kappa_3) - C_{+-}(\kappa_1 + \kappa_3) e^{-i\frac{\kappa_1^2}{4T}}}{2\pi\kappa_1^2} C_{+-}(\kappa_3) e^{-i\frac{\kappa_1\kappa_3}{2T}} \\
& - 2 \lim_{T\to\infty} \int d\kappa_1 d\kappa_3 \, \frac{C_{+-}(\kappa_3) - C_{+-}(\kappa_1 + \kappa_3) e^{i\frac{\kappa_1^2}{4T}}}{2\pi\kappa_1^2} C_{+-}(\kappa_3) e^{i\frac{\kappa_1\kappa_3}{2T}},
\end{aligned} \tag{135}$$

which we may rewrite as

$$\begin{aligned}
\lim_{T\to\infty} T^{-1} \langle \Psi | (J_0 \big|_0^T)^2 | \Psi \rangle = {} & 4 \int d\kappa \, \frac{C_{+-}(0) - C_{+-}(\kappa)}{2\pi\kappa^2} + 4 \lim_{T\to\infty} \int d\kappa \, \frac{C_{+-}(\kappa)(1 - \cos(\frac{\kappa^2}{4T}))}{2\pi\kappa^2} \\
& - \lim_{T\to\infty} 4 \int d\kappa d\tau \, \hat{C}_{+-}(\frac{\kappa}{2T} - \tau) \hat{C}_{+-}(\tau) \frac{1 - \cos(\frac{\kappa^2}{4T} - \kappa\tau)}{(2\pi)^2\kappa^2} \\
= {} & 4 \int d\kappa \, \frac{C_{+-}(0) - C_{+-}(\kappa)}{2\pi\kappa^2} - 4 \int d\kappa d\tau \, \hat{C}_{+-}(-\tau) \hat{C}_{+-}(\tau) \frac{1 - \cos(\kappa\tau)}{(2\pi)^2\kappa^2} \\
= {} & \int \frac{d\tau}{2\pi} |2\tau| \hat{C}_{+-}(\tau) - \int \frac{d\tau}{2\pi} |2\tau| \hat{C}_{+-}^2(\tau) \\
= {} & \int \frac{d\tau}{2\pi} |2\tau| \hat{C}_{+-}(\tau) \left(1 - \hat{C}_{+-}(\tau)\right) = \int \frac{d\tau}{2\pi} |2\tau| \theta(\tau)(1 - \theta(\tau)),
\end{aligned} \tag{136}$$

where in the second line we used similar considerations as previously, and in particular the bounded convergence theorem; and $\hat{C}_{+-}(\tau)$ denotes the Fourier transform of the 2-pt correlation function $C_{+-}(\kappa)$. Importantly, in free fermion GGEs $\hat{C}_{+-}(\tau) = \theta(\tau)$, where $\theta(\tau)$ is the filling function associated with the GGE. Now we can see how the fact that $C_{+-}$ is even (just like its Fourier transform) and two-times differentiable guarantees that the second current cumulant is finite; the finiteness of (114) equivalently means that $\theta(\tau)$ asymptotically decays stronger than $\tau^{-2}$, making the $\int d\tau |2\tau| \theta(\tau)$ integral finite. This finding can be interpreted as the sufficiently fast decay of quasi-particle occupation wrt. the momentum or rapidity, which implies finite current cumulants, is guaranteed by the smoothness of the correlation functions in the initial state.

## 4.2 Factorisation Property 2 for free non-relativistic fermions

In this subsection relying on free non-relativistic fermions, we demonstrate that under generic conditions, i.e., the sufficiently strong clustering of a given state the following equality holds:

$$\ln \frac{\langle \Psi | e^{\lambda J_0|_0^T + \lambda Q|_0^\infty(0)} | \Psi \rangle}{\langle \Psi | e^{\lambda Q|_0^\infty(0)} | \Psi \rangle} = \ln \frac{\langle \Psi | e^{\lambda J_0|_0^T} e^{\lambda Q|_0^\infty(0)} | \Psi \rangle}{\langle \Psi | e^{\lambda Q|_0^\infty(0)} | \Psi \rangle} + o(T), \tag{137}$$

when $T$ is large, and we shall further simplify our notations by

$$\tilde{J} = J_0|_0^T = \int_0^T dt \, j(0,t), \quad \tilde{Q} = Q|_0^\infty(0) = \int_0^\infty dx \, q(x,0). \tag{138}$$

Note that for brevity and simplicity, we restrict our analysis to the case of the asymmetric case above (137), however, from this one

$$\ln \frac{\langle \Psi | e^{\lambda/2 Q|_0^\infty(0)} e^{\lambda J_0|_0^T} e^{\lambda/2 Q|_0^\infty(0)} | \Psi \rangle}{\langle \Psi | e^{\lambda_i Q_i|_0^\infty(0)} | \Psi \rangle} = \ln \frac{\langle \Psi | e^{\lambda J_0|_0^T} e^{\lambda Q|_0^\infty(0)} | \Psi \rangle}{\langle \Psi | e^{\lambda Q|_0^\infty(0)} | \Psi \rangle} + o(T), \tag{139}$$

also follows. To see this, one can group $\lambda \tilde{J} + \lambda \tilde{Q}$ as $\lambda \tilde{J} + \lambda/2 \tilde{Q} + \lambda/2 \tilde{Q}$ in the exponent, first show the factorisation of $\lambda \tilde{J} + \lambda/2 \tilde{Q}$ and $\lambda/2 \tilde{Q}$, and afterwards that of $\lambda \tilde{J} + \lambda/2 \tilde{Q}$ based on the procedure we immediately explain. However, for preciseness, we shall also demonstrate in the Sections 5 and 6 that regarding the time evolution in the modified, inhomogeneous quench problem, the symmetric ($\langle \Psi | e^{\lambda/2\tilde{Q}} O_1 ... O_n e^{\lambda/2\tilde{Q}} | \Psi \rangle$) and asymmetric initial states ($\langle \Psi | O_1 ... O_n e^{\lambda\tilde{Q}} | \Psi \rangle$) yield the same result, at least for free fermionic systems and integrable quenches.

The main idea to demonstrate the validity of (137) is performing an expansion wrt. $\lambda$ and check if certain correlation functions, specified below, give negligible contributions. For the sake of simplicity we will focus on non-relativistic free fermions and the particle number and the associated current. Nevertheless the calculations presented below are expected to generalise to other charges or currents, which are still bilinears of the fermion creation and annihilation operators. In the following we shall carry out the expansion wrt. the counting field directly of (137) up to the 3rd order and show that in each order, the leading $T$ behaviour is identical. For further brevity, the details of the 3rd order computations are relegated to Appendix B, but the treatment of the second order together with the main techniques we use are presented here.

### 4.2.1 Series expansion of (137)

Using the shorthand (138), let us rewrite the lhs. of (137) using the Zassenhaus formulas:

$$\ln \frac{\langle \Psi | e^{\lambda \tilde{J} + \lambda \tilde{Q}} | \Psi \rangle}{\langle \Psi | e^{\lambda \tilde{Q}} | \Psi \rangle} = \ln \langle \Psi | e^{\lambda \tilde{J}} e^{\lambda \tilde{Q}} e^{-\frac{\lambda^2}{2} [\tilde{J}, \tilde{Q}]} e^{\frac{\lambda^3}{6}(2[\tilde{Q},[\tilde{J},\tilde{Q}]] + [\tilde{J},[\tilde{J},\tilde{Q}]])} \times (...) | \Psi \rangle - \ln \langle \Psi | e^{\lambda \tilde{Q}} | \Psi \rangle. \tag{140}$$

In the above expression, up to 3rd order, we have the following terms

$$\lambda: \quad \langle\Psi|\tilde{J}|\Psi\rangle$$

$$\lambda^2: \quad \frac{1}{2}\left\{\langle\Psi|\tilde{J}^2|\Psi\rangle - \langle\Psi|\tilde{J}|\Psi\rangle^2 + 2\langle\Psi|\tilde{J}\tilde{Q}|\Psi\rangle - 2\langle\Psi|\tilde{J}|\Psi\rangle\langle\Psi|\tilde{Q}|\Psi\rangle - \langle\Psi|[\tilde{J},\tilde{Q}]|\Psi\rangle\right\}$$

$$\lambda^3: \quad \frac{1}{6}\left\{\langle\Psi|\tilde{J}^3|\Psi\rangle - 3\langle\Psi|\tilde{J}^2|\Psi\rangle\langle\Psi|\tilde{J}|\Psi\rangle + 2\langle\Psi|\tilde{J}|\Psi\rangle^3 + 3\langle\Psi|\tilde{J}^2\tilde{Q}|\Psi\rangle - 6\langle\Psi|\tilde{J}\tilde{Q}|\Psi\rangle\langle\Psi|\tilde{J}|\Psi\rangle\right.$$

$$+3\langle\Psi|\tilde{J}\tilde{Q}^2|\Psi\rangle - 6\langle\Psi|\tilde{J}\tilde{Q}|\Psi\rangle\langle\Psi|\tilde{Q}|\Psi\rangle - 3\langle\Psi|\tilde{Q}[\tilde{J},\tilde{Q}]|\Psi\rangle + 3\langle\Psi|\tilde{Q}|\Psi\rangle\langle\Psi|[\tilde{J},\tilde{Q}]|\Psi\rangle$$

$$\left. -3\langle\Psi|\tilde{J}[\tilde{J},\tilde{Q}]|\Psi\rangle + 3\langle\Psi|\tilde{J}|\Psi\rangle\langle\Psi|[\tilde{J},\tilde{Q}]|\Psi\rangle + 2\langle\Psi|\big[\tilde{Q},[\tilde{J},\tilde{Q}]\big]|\Psi\rangle + \langle\Psi|\big[\tilde{J},[\tilde{J},\tilde{Q}]\big]|\Psi\rangle\right\}. \tag{141}$$

Up to the same order, the expansion of the rhs. of (137), that is, $\ln\langle\Psi|e^{\lambda\tilde{J}}e^{\lambda\tilde{Q}}|\Psi\rangle - \ln\langle\Psi|e^{\lambda\tilde{Q}}|\Psi\rangle$ reads instead as

$$\lambda: \quad \langle\Psi|\tilde{J}|\Psi\rangle$$

$$\lambda^2: \quad \frac{1}{2}\left\{\langle\Psi|\tilde{J}^2|\Psi\rangle - \langle\Psi|\tilde{J}|\Psi\rangle^2 + 2\langle\Psi|\tilde{J}\tilde{Q}|\Psi\rangle - 2\langle\Psi|\tilde{J}|\Psi\rangle\langle\Psi|\tilde{Q}|\Psi\rangle\right\}$$

$$\lambda^3: \quad \frac{1}{6}\left\{\langle\Psi|\tilde{J}^3|\Psi\rangle - 3\langle\Psi|\tilde{J}^2|\Psi\rangle\langle\Psi|\tilde{J}|\Psi\rangle + 2\langle\Psi|\tilde{J}|\Psi\rangle^3 + 3\langle\Psi|\tilde{J}^2\tilde{Q}|\Psi\rangle - 6\langle\Psi|\tilde{J}\tilde{Q}|\Psi\rangle\langle\Psi|\tilde{J}|\Psi\rangle\right.$$

$$\left. +3\langle\Psi|\tilde{J}\tilde{Q}^2|\Psi\rangle - 6\langle\Psi|\tilde{J}\tilde{Q}|\Psi\rangle\langle\Psi|\tilde{Q}|\Psi\rangle\right\}, \tag{142}$$

with the same terms as in (141), except for the last term on the 2nd order, and the last six terms in the 3rd order in (141). That is, in other words, it is sufficient to show for (137) that

$$\lambda^2: \quad \langle\Psi|[\tilde{J},\tilde{Q}]|\Psi\rangle = o(T)$$

$$\lambda^3: \quad \langle\Psi|\tilde{J}[\tilde{J},\tilde{Q}]|\Psi\rangle = \langle\Psi|\tilde{Q}[\tilde{J},\tilde{Q}]|\Psi\rangle - \langle\Psi|\tilde{Q}|\Psi\rangle\langle\Psi|[\tilde{J},\tilde{Q}]|\Psi\rangle = \tag{143}$$

$$= \langle\Psi|\big[\tilde{Q},[\tilde{J},\tilde{Q}]\big]|\Psi\rangle = \langle\Psi|\big[\tilde{J},[\tilde{J},\tilde{Q}]\big]|\Psi\rangle = o(T)$$

hold, where we assume that $|\Psi\rangle$ is a translationally invariant parity even state with short range correlations, hence $\langle\Psi|\tilde{J}|\Psi\rangle$ and consequently $\langle\Psi|\tilde{J}|\Psi\rangle\langle\Psi|[\tilde{J},\tilde{Q}]|\Psi\rangle$ are zero. Although it may happen that in the 3rd order only combinations of the above terms are sub-leading wrt. $T$ we anticipate that this is not the case and shall demonstrate it for each term, with the exception of $\langle\Psi|\tilde{Q}[\tilde{J},\tilde{Q}]|\Psi\rangle$ and $\langle\Psi|\tilde{Q}|\Psi\rangle\langle\Psi|[\tilde{J},\tilde{Q}]|\Psi\rangle$, which are sub-linear in $T$ but also extensive wrt. the volume, hence only their difference is finite in a strict sense.

To show the vanishing of the above terms, we shall work with non-relativistic free fermions and assume that the initial state is exponentially clustering and is translation and parity invariant. For the sake of simplicity, we also assume that it has a definite fermion number parity, which means that $\langle\Psi|\psi^\dagger(y)|\Psi\rangle = 0$ as well as $\langle\Psi|\psi(y)|\Psi\rangle = 0$. Because of the aforementioned properties, we may write the fermion 2-pt functions according to (90) since $|\Psi\rangle$ is translation invariant, the 2-pt correlation functions can depend only on the difference of spatial arguments, and additionally, due to $|\Psi\rangle$ being parity even and since the fermionic field operators transform as $\hat{P}\psi(x)\hat{P} = -\psi(-x)$, the 2-pt functions must be even functions as well. Additionally, $C_{+-}(y-z)$ is non-divergent and when $z = y$ it should give the charge density $\langle\Psi|q(0)|\Psi\rangle$ which we denote by the shorthand $\langle q(0)\rangle$. For simplicity, we shall also assume that the other 2-pt functions $C_{++}(x)$, $C_{--}(x)$ are also regular.

To proceed we use (97) for the integrated current in which we shall often split and rewrite the function $G$ defined in (98) as

$$G(y,z,T)_\epsilon = G_1(y,z,T) - G_1(y,z,\epsilon) = -\mathrm{i}\frac{e^{-\mathrm{i}\frac{y^2-z^2}{4T}}-1}{2\pi(y-z)} + \mathrm{i}\frac{e^{-\mathrm{i}\frac{y^2-z^2}{4\epsilon}}-1}{2\pi(y-z)}, \tag{144}$$

where the second term is independent of $T$ which we can make advantage of. When a series expansion is regarded, $G_1$ can be easier to deal with, although the integration of $G_1$ on specific regions may be divergent which has to be taken into account. Additionally, both $G_\epsilon$, and $G_1$ have the properties that $G(y, z, t) = G(z, y, t)^*$. A further important property is that for $G_1(y, z, \epsilon)$ one can show that in a distribution-sense

$$\lim_{\epsilon \to 0} -\mathrm{i}\frac{e^{-\mathrm{i}\frac{y^2-z^2}{4\epsilon}} - 1}{2\pi(y-z)} = \lim_{\epsilon \to 0} -\mathrm{i}\mathrm{P}\frac{e^{-\mathrm{i}\frac{y^2-z^2}{4\epsilon}} - 1}{2\pi(y-z)} = -\frac{1}{2}\delta(y-z)\,\mathrm{sgn}(y) + \mathrm{i}\frac{1}{2\pi}\mathrm{P}\frac{1}{y-z}\,, \tag{145}$$

which follows from (179) (cf. Section 5).

Now, exploiting that

$$\left[\psi^\dagger(x)\psi(y), \psi^\dagger(z)\psi(w)\right] = \psi^\dagger(x)\psi(w)\delta(y-z) - \psi^\dagger(z)\psi(y)\delta(x-w)\,, \tag{146}$$

we can move onto the analysis of the first commutator appearing in the second order.

### 4.2.2 Second order in $\lambda$

For the commutator itself, i.e., the sole term appearing at the 2nd order, we have that

$$[\tilde{J}, \tilde{Q}(0)] = \lim_{\epsilon \to 0} \int \mathrm{d}y \int \mathrm{d}z \int_0^\infty \mathrm{d}x\, G(y, z, T)_\epsilon [\psi^\dagger(y)\psi(z), \psi^\dagger(x)\psi(x)]$$

$$= \lim_{\epsilon \to 0} \int \mathrm{d}y \int_0^\infty \mathrm{d}x \left\{ G(y, x, T)_\epsilon \psi^\dagger(y)\psi(x) - G(x, y, T)_\epsilon \psi^\dagger(x)\psi(y) \right\} \tag{147}$$

$$= \lim_{\epsilon \to 0} \int_{-\infty}^0 \mathrm{d}y \int_0^\infty \mathrm{d}x \left\{ G(y, x, T)_\epsilon \psi^\dagger(y)\psi(x) - G(x, y, T)_\epsilon \psi^\dagger(x)\psi(y) \right\}.$$

In order to evaluate the expectation value of (147) in exponentially clustering initial states, we proceed as follows. As a first step we split $G_\epsilon$ into a $T$- and an $\epsilon$-dependent bits according to (144)

$$\langle \Psi | [\tilde{J}, \tilde{Q}(0)] | \Psi \rangle = \lim_{\epsilon \to 0} \int_{-\infty}^0 \mathrm{d}y \int_0^\infty \mathrm{d}x \left\{ G(y, x, T)_\epsilon C_{+-}(y-x) - G(x, y, T)_\epsilon C_{+-}(x-y) \right\}$$

$$= \int_{-\infty}^0 \mathrm{d}y \int_0^\infty \mathrm{d}x \left\{ G_1(y, x, T) C_{+-}(y-x) - G_1(x, y, T) C_{+-}(x-y) \right\} \tag{148}$$

$$- \lim_{\epsilon \to 0} \int_{-\infty}^0 \mathrm{d}y \int_0^\infty \mathrm{d}x \left\{ G_1(y, x, \epsilon) C_{+-}(y-x) - G_1(x, y, \epsilon) C_{+-}(x-y) \right\},$$

where the integrals including $G_1(.,.,T)$ and $G_1(.,.,\epsilon)$ are finite. The next observation is that, due to the integration range and the exponential decay of the 2-pt function, the dominant contribution comes from the origin. In addition, $T$ is large, therefore, we perform a double-expansion of the $T$-dependent oscillatory $G$-functions. The crucial property is that, at any order in the expansion, we obtain polynomials of $x$ and $y$ multiplied by growing negative powers of $T$ and the exponentially decaying correlation function. Due to the decay of the latter and the range of integration, we obtain finite integrals at any order, as

$$-\infty < \int_{-\infty}^0 \mathrm{d}y \int_0^\infty \mathrm{d}x\, (x+y)^n (x-y)^m C_{+-}(y-x) < \infty\,. \tag{149}$$

This implies that the obtained series, eventually consisting of the inverse powers of $T$, is a well-defined asymptotic expansion. In particular, performing the expansion of $G_1(.,.,T)$ we have

$$
\langle\Psi|[\tilde{J},\tilde{Q}(0)]|\Psi\rangle = \int_{-\infty}^{0} dy \int_{0}^{\infty} dx \left\{ \left[ (y+x)T^{-1} + (\text{higher powers of x,y}) \times T^{-n} \right] C_{+-}(y-x) \right.
$$
$$
\left. - \left[ (y+x)T^{-1} + (\text{higher powers of x,y}) \times T^{-n} \right] C_{+-}(x-y) \right\}
$$
$$
- \lim_{\epsilon \to 0} \int_{-\infty}^{0} dy \int_{0}^{\infty} dx \left\{ G_1(y,x,\epsilon) - G_1(x,y,\epsilon) \right\} C_{+-}(x-y)
$$
$$
= \mathcal{O}(T^{-2}) + \mathcal{O}(1) = o(T), \tag{150}
$$

since in (150) we expect the eventual cancellation of the $\mathcal{O}(T^{-1})$ term, as $C_{+-}$ is even, and $T^{-n}$ means growing negative powers of $T$ wrt. the order of the expansion. The second integral gives $-2ic_1 + \sqrt{\epsilon}2c_2$, $(c_2 \approx 0.3989\,\mathrm{i})$ with

$$
c_1 = \int_{-\infty}^{0} dy \int_{0}^{\infty} dx \, \frac{C_{+-}(y-x)}{2\pi(y-x)} < \infty, \tag{151}
$$

according to (145); due to the specific integration range, the Dirac-$\delta$ term in (145) is vanishing, and no divergence emerges from the region $x \approx y \approx 0$ hence the taking the principal value is not necessary. That is, we obtain also a $\mathcal{O}(1)$ quantity, which is non-vanishing despite having a difference in (150). We note that thanks to the range of integration, in the distribution sense we have that

$$
\lim_{\epsilon \to 0} \int_{-\infty}^{0} dy \int_{0}^{\infty} dz \, \frac{-i\left(e^{-i\frac{y^2-z^2}{4\epsilon}} - 1\right)}{2\pi(y-z)} = \int_{-\infty}^{0} dy \int_{0}^{\infty} dz \, i\frac{1}{2\pi(y-z)}, \tag{152}
$$

where the rhs. does not account for divergences due to integration range.

### 4.2.3 Summary of the third order in $\lambda$

The analysis of the 3rd order terms in (143) are carried out explicitly in Appendix B and here we merely summarise the results of the corresponding computations. We have managed to demonstrate that each term in (143) are $o(T)$ in exponentially clustering, parity even and translation invariant states. We note that the precise requirement for the clustering of higher-point function is as follows:

$$
|\langle\Psi|\psi^{\dagger}(x_1)\psi(x_2)\psi^{\dagger}(x_3)\psi(x_4)|\Psi\rangle^c| \le C e^{-c\ell} \tag{153}
$$

for some positive $c$ and $C$, where

$$
\ell = \max_{\substack{A \cup B = \{x_1,x_2,x_3,x_4\} \\ A \cap B = \emptyset, A \ne \emptyset, B \ne \emptyset}} \text{dist}(A,B), \tag{154}
$$

that is, the 4-pt correlation functions decay exponentially wrt. the maximum distance of all possible bipartitions of the arguments $x_1, x_2, x_3, x_4$ excluding empty sets. We anticipate that the above clustering condition can be relaxed to demanding the decay of the connected function only when one argument is far away from the others, and when the decay is slower than exponential, i.e., it is a high enough power of the appropriate distance. However, (153) facilitates a transparent demonstration of the factorisation property and relaxing (153), which would include more technicalities, is left for future studies.

For simplicity, we also assumed that the connected 4-pt functions do not have any singularities when carrying out our calculations in Appendix B. In continuous models, such as non-relativistic

free fermions, this may not be the case. However, upon implementing suitable short distance regularisations, the 4-pt and higher point functions can be regularised, therefore we expect that and our results remain unchanged.

To summarise our findings, in Section 4.2.2 and in Appendix B, we have shown that

$$\lambda^2 : \langle\Psi|[\tilde{J},\tilde{Q}]|\Psi\rangle = \mathcal{O}(1) + o(T^{-1})$$
$$\lambda^3 : \langle\Psi|\tilde{J}[\tilde{J},\tilde{Q}]|\Psi\rangle = \mathcal{O}(1) + \mathcal{O}(T^{-1}), \ \langle\Psi|\tilde{Q}[\tilde{J},\tilde{Q}]|\Psi\rangle - \langle\Psi|\tilde{Q}|\Psi\rangle\langle\Psi|[\tilde{J},\tilde{Q}]|\Psi\rangle = \mathcal{O}(1) + \mathcal{O}(T^{-1}),$$
$$\langle\Psi|[\tilde{Q},[\tilde{J},\tilde{Q}]]|\Psi\rangle = 0, \ \langle\Psi|[\tilde{J},[\tilde{J},\tilde{Q}]]|\Psi\rangle = 0,$$

(155)

i.e., that

$$\ln\frac{\langle\Psi|e^{\lambda\tilde{J}+\lambda\tilde{Q}}|\Psi\rangle}{\langle\Psi|e^{\lambda\tilde{Q}}|\Psi\rangle} = \ln\frac{\langle\Psi|e^{\lambda\tilde{J}}e^{\lambda\tilde{Q}}|\Psi\rangle}{\langle\Psi|e^{\lambda\tilde{Q}}|\Psi\rangle}$$

(156)

for large $T$ up to $o(T)$ corrections and up to the 3rd order wrt. the counting field $\lambda$.

### 4.2.4 Higher orders

In this part we discuss the behaviour of certain terms coming from higher orders upon expanding the left and right sides of (137). In particular, we can argue in a transparent way, that all expectation values containing fully nested commutators give not stronger than $o(T)$ contributions. Whereas we refrain from the study of multi-point functions of nested commutators and $\tilde{Q}$ and/or $\tilde{J}$ operators, the vanishing expectation value of nested commutators is a strong indication for validity of (137), at any order.

Let us concentrate on a generic nested commutator,

$$\left[X_1,\left[X_2,...,\left[X_{n-2},[\tilde{J},\tilde{Q}]\right]...\right]\right],$$

(157)

where each $X_i$ can be either $\tilde{J}$ or $\tilde{Q}$. The first observation is that, by symmetry, the expectation value of such nested commutators is always zero in even parity states, if the parity of the number of $\tilde{J}$ operators denoted by $k$ and the parity of the number of $\tilde{Q}$ operators denoted by $l$ are different ($k + l = n$). This is because under spatial reflection ($\hat{P}$), $\tilde{J}$ transforms as $\tilde{J} \to -\tilde{J}$, and $\tilde{Q}$ as $\tilde{Q} \to Q - \tilde{Q}$. Since (for any charge) an arbitrary nested commutator always consists of precisely one fermion creation field $\psi^\dagger$ and one annihilation field $\psi$, any nested commutator vanishes which includes the total charge. That is, we have that under parity

$$\langle\Psi|\left[X_1,\left[X_2,...\left[X_{n-1},[\tilde{J},\tilde{Q}]\right]...\right]\right]|\Psi\rangle \xrightarrow{\hat{P}} (-1)^{k+l}\langle\Psi|\left[X_1,\left[X_2,...\left[X_{n-1},[\tilde{J},\tilde{Q}]\right]...\right]\right]|\Psi\rangle.$$ (158)

If the parity of $k$ and $l$ is the same, we can use the fact that the expectation value of the nested commutator can be written in the form

$$\sum_{\substack{k_1\geq 1, k_2\geq 1 \\ k_1+k_2=k+1}} \sum_{\sigma\in S_c(\{+_{k_1},-_{k_2}\})} d(k,l,\sigma)\left(\prod_{i=1}^{k+1}\int_{\sigma_i(+,-)}dx_i\right)\prod_{j=1}^{k}G(x_j,x_{j+1},T)_\epsilon C_{+-}(x_{k+1}-x_1)$$
$$-\prod_{j=1}^{k}G(x_{k+2-j},x_{k+1-j},T)_\epsilon C_{+-}(x_1-x_{k+1}),$$

(159)

where $S_c(\{+_{k_1},-_{k_2}\})$ means all different cyclic permutations of the $k_1 \geq 1$ " + " symbols, and $k_2 \geq 1$ " − " symbols, and $d(k,l,\sigma)$ denotes numerical factors that depend on the number and order of $\tilde{J}$ and $\tilde{Q}$ operators. Here the symbols $\sigma_i(+,-)$ denote the range of integrations, which are either $[0,\infty)$ (+), or $(-\infty,0]$ (-).

We can now argue by a scaling argument that the corresponding expectation value is always $o(T)$: we use new variables $x_i = \ell \bar{x}_i$ and $T = \ell \bar{T}$ and after this change of variables, (159) can be rewritten as

$$
\ell \sum_{\substack{k_1+k_2=k+1 \\ k_1 \geq 1, k_2 \geq 1}} \sum_{\sigma \in S_c(\{+_{k_1}, -_{k_2}\})} d(k,l,\sigma) \left( \prod_{i=1}^{k+1} \int_{\sigma_i(+,-)} d\bar{x}_i \right) \prod_{j=1}^{k} G(\ell\bar{x}_j, \ell\bar{x}_{j+1}, \ell T)_\epsilon C_{+-}(\ell\bar{x}_{k+1} - \ell\bar{x}_1)
$$
$$
- \prod_{j=1}^{k} G(\ell\bar{x}_{k+2-j}, \ell\bar{x}_{k+1-j}, \ell T)_\epsilon C_{+-}(\ell\bar{x}_1 - \ell\bar{x}_{k+1}),
$$
(160)

where $\ell$ can be taken large and there is one overall $\ell$ factor in (160), as we have $k+1$ integration variables (accounting for $\ell^{k+1}$) and $k$ $G$ functions (accounting for $\ell^{-k}$). Exploiting that we have one multiplicative $\ell$ factor in (160), we can write that that

$$
\ell \int d\bar{x}\, C_{+-}(\ell\bar{x}) = \text{const}, \tag{161}
$$

that is, when $\ell$ is taken large we may approximate $C_{+-}$ as

$$
\ell C_{+-}(\ell\bar{x}) \approx \delta(\bar{x}), \tag{162}
$$

that is, the correlation function in the integrals can be replaced by a Dirac-$\delta$ at large spatio-temporal scales and the resulting expression shall account for the same temporal scaling behaviour wrt. $T$. Using $\delta$-functions in (160), we can see that if the range of integration for the $x_1$ and $x_{k+1}$ integrals are different $+,-$ or $-,+$, then the resulting expression is 0, hence $o(T)$ by the definition of the $\delta$-functions. In general, the integration range for the $x_1$ and $x_{k+1}$ variable can be the same ($+,+$ or $-,-$) as well. However, also in this case we expect $o(T)$ contribution: considering a single term from (159), we have an expression

$$
d(k,l,\sigma) \left( \prod_{i=1}^{k} \int_{\sigma_i(+,-)} dx_i \right) \left[ \prod_{j=1}^{k} G(x_j, x_{j+1}, T)_\epsilon - \prod_{j=1}^{k} G(x_{k+2-j}, x_{k+1-j}, T)_\epsilon \right] = 0, \tag{163}
$$

where we switched back to the original variables after performing the integration including the $\delta$-function. In the expression above $x_{k+1} = x_1$, and the integral is zero, since in the two products consisting of the same functions, the same integration with the same range is carried out but in a reversed order wrt. each other. That is, given the reasonings including the assertion that the $\delta$-correlated 2-pt function gives the same $o(T)$ scaling behaviour as exponentially decaying 2-pt function, we can argue that the expectation values of all fully nested commutators are $o(T)$.

## 5 Local relaxation to the NESS: microscopic results for free fermions

The aim of this section is twofold. First of all we demonstrate that quenching from $|\tilde{\Omega}\rangle_\lambda$ (cf. (68)), local operators relax to a NESS for free fermions and integrable initial states. Second, we show that $\rho_{\text{NESS}}(\lambda)$ characterising the NESS equals $\rho_{\text{bpp}}(\lambda)$, where $\rho_{\text{bpp}}(\lambda)$ is the NESS of a slightly different partitioning protocol, where the initial states on the left/right are assumed to be the generalised Gibbs ensembles emerging from the left/right microscopic initial states (see (20) and below). In addition, we shall also investigate the asymptotic time dependence, which provides valuable insight into how fast the stationary state is reached.

Whereas these claims should be shown for any local operator and a certain class a quasi-local operators (see Sec. 3.1) as well, as we argued earlier, it is sufficient to consider the relaxation of conserved densities, which are expected to imply that of the former ones. For brevity, in this section we shall only concentrate on the behaviour of the current $j$ associated with the charge under investigation (which, for the case of particle number is proportional to the momentum density) but in Appendix D we demonstrate the convergence in the more general case namely for local operators and a representative class of quasi-local operators as well.

Our strategy consists of performing a series expansion of the correlation functions in the counting field $\lambda$. In particular, we need to consider two states: $|\tilde{\Omega}\rangle_\lambda$, defined in (68), and $|\tilde{\Omega}^*\rangle_{2\lambda}$, defined as

$$|\tilde{\Omega}^*\rangle_{2\lambda} = \frac{e^{\lambda\tilde{Q}}|\Omega\rangle}{\langle\Omega|e^{\lambda\tilde{Q}}|\Omega\rangle}\,, \tag{164}$$

(note the absence of the square root in the denominator, that will be clear below). Such states allow to define the "symmetric" and "asymmetric" expectation values via, respectively,

$$_\lambda\langle\tilde{\Omega}|O(x,t)|\tilde{\Omega}\rangle_\lambda = \frac{\langle\Omega|e^{\lambda/2\tilde{Q}}O(x,t)^{\lambda/2\tilde{Q}}|\Omega\rangle}{\langle\Omega|e^{\lambda\tilde{Q}}|\Omega\rangle} \quad \text{and} \quad \langle\Omega|O(x,t)|\tilde{\Omega}^*\rangle_{2\lambda} = \frac{\langle\Omega|O(x,t)e^{\lambda\tilde{Q}}|\Omega\rangle}{\langle\Omega|e^{\lambda\tilde{Q}}|\Omega\rangle}\,, \tag{165}$$

In the series expansion, we shall use the fact that (from the definitions (165))

$$\partial_\lambda\left(\langle\Omega|O(x,t)|\tilde{\Omega}^*\rangle_{2\lambda}\right)|_{\lambda=0} = \langle\Omega|O(t,x)\tilde{Q}|\Omega\rangle^c$$
$$\partial_\lambda^2\left(\langle\Omega|O(x,t)|\tilde{\Omega}^*\rangle_{2\lambda}\right)|_{\lambda=0} = \langle\Omega|O(t,x)\tilde{Q}^2|\Omega\rangle^c\,, \tag{166}$$

and

$$\partial_\lambda\left(_\lambda\langle\tilde{\Omega}|O(x,t)|\tilde{\Omega}\rangle_\lambda\right)|_{\lambda=0} = \frac{\langle\Omega|O(t,x)\tilde{Q}|\Omega\rangle^c + \langle\Omega|\tilde{Q}O(t,x)|\Omega\rangle^c}{2}$$
$$\partial_\lambda^2\left(_\lambda\langle\tilde{\Omega}|O(x,t)|\tilde{\Omega}\rangle_\lambda\right)|_{\lambda=0} = \frac{\langle\Omega|O(t,x)\tilde{Q}^2|\Omega\rangle^c + 2\langle\Omega|\tilde{Q}O(t,x)\tilde{Q}|\Omega\rangle^c + \langle\Omega|\tilde{Q}^2O(t,x)|\Omega\rangle^c}{4}\,, \tag{167}$$

where we can utilise connected correlation function to account for the normalisation. The computations of the aforementioned quantities are straightforward but the formulae quickly become lengthy. In the following, we will concentrate on the case $O(x,t) = j(0,t)$ (so from the above formulae, we will evaluate correlation functions of $j$ and $\tilde{Q}$).

It has to be noted that applying such series expansion is not entirely trivial. At time $t = 0$ the initial state is discontinuous at $x = 0$, thus expectation values (at $x = 0$) are expected to be ill-defined and the expansion may break down. As we shall see, this is indeed the case, at $t = 0$ some terms from the expansion diverge. However, for finite positive times $t > 0$ the initial configuration smoothens out and the series expansion yields finite terms that converge to the expected steady-state order by order which are cross-checked by the analogous results obtained via expanding the expectation value in the NESS (cf. Appendix C.3).

In order to set up these calculations, let us start by recalling the following expressions for the current

$$j(0,t) = \frac{1}{2\pi}\int dk\, dp\, f_{k,p}(t)a_k^\dagger a_p\,, \tag{168}$$

where we introduce the short-hands

$$f_{k,p}(t) = e^{i(E_k-E_p)t}(k+p)\,, \quad f_{k,p} = f_{k,p}(t=0) = k+p\,, \tag{169}$$

and for the integrated charge

$$
\begin{aligned}
\tilde{Q} &= \lim_{\epsilon \to 0} \int_0^\infty \mathrm{d}x \, q(x,0) e^{-\epsilon x} = \lim_{\epsilon \to 0} \frac{1}{2\pi} \int \mathrm{d}k \mathrm{d}p \frac{-\mathrm{i}}{(k-p) - \mathrm{i}\epsilon} q^{\mathrm{e}}(k,p) \, a_k^\dagger a_p \\
&= \frac{1}{2\pi} \int \mathrm{d}k \mathrm{d}p \left( \pi \delta(k-p) - \mathrm{i} \mathrm{P} \frac{1}{k-p} \right) q^{\mathrm{e}}(k,p) \, a_k^\dagger a_p \,,
\end{aligned}
\tag{170}
$$

where the limit $\epsilon \to 0$ regularises the integration (note the last equality is not obvious in some of the relevant cases for our purposes, cf. the discussions in Sec. 5.1.1 and Sec. 5.1.3). In the following, we shall drop the function $q^{\mathrm{e}}(k,p)$ from $\tilde{Q}$ to shorten the expressions, and just keep in mind that $q^{\mathrm{e}}(k,p)$ is an even function ($q^{\mathrm{e}}(-k,-p) = q^{\mathrm{e}}(k,p)$).

Below, in order to evaluate the building blocks of our calculations, we are also going to use that both quantities have the structure

$$
\frac{1}{2\pi} \int \mathrm{d}k \, \mathrm{d}p \, H_{k,p} a_k^\dagger a_p
\tag{171}
$$

for some function $H_{k,p}$ associated to the bilinear operator $H$.

To proceed we now recall the Bogolyubov transformation that simplifies the *homogeneous* initial state $|\Omega\rangle$, i.e.,

$$
a_p = u_p \tilde{a}_p + v_p \tilde{a}_{-p}^\dagger, \qquad a_p^\dagger = v_p^* \tilde{a}_{-p} + u_p \tilde{a}_p^\dagger
\tag{172}
$$

where the $\tilde{a}$ operators annihilate the state $\tilde{a}_k |\Omega\rangle = 0$ and where

$$
u_k = \frac{1}{\sqrt{1 + |K(k)|^2}}, \quad v_k = \frac{K(k)}{\sqrt{1 + |K(k)|^2}} \,.
\tag{173}
$$

Importantly, $|\Omega\rangle$ is Gaussian, hence according to (173), expectation values involving the original creation and annihilation operators are simply obtained by the contractions rules (cf. Appendix C.2)

$$
\begin{aligned}
\langle \Omega | a_k^\dagger a_p | \Omega \rangle &= 2\pi \delta(k-p) \frac{|K(k)|^2}{1 + |K(k)|^2} \,, \quad \text{and} \\
\langle \Omega | a_k^\dagger a_p^\dagger | \Omega \rangle &= 2\pi \delta(k+p) \frac{K(k)^*}{1 + |K(k)|^2} \,, \quad \langle \Omega | a_k a_p | \Omega \rangle = 2\pi \delta(k+p) \frac{K(-k)}{1 + |K(k)|^2}
\end{aligned}
\tag{174}
$$

and by applying Wick's theorem. It is therefore immediate that for one bilinear $H^{(1)}$ we have the simple expectation value

$$
\langle \Omega | H^{(1)} | \Omega \rangle = \int \frac{\mathrm{d}k}{2\pi} H_{k,k}^{(1)} \frac{|K(k)|^2}{1 + |K(k)|^2} \,.
\tag{175}
$$

For the current 1-pt correlation function, the first two non-trivial orders are $\lambda$ and $\lambda^2$, whereas for the 2-pt function (needed in next section) they are $\lambda^0$, and $\lambda$. For both quantities, we shall need the expectation value of two and three fermion bilinears.

For two bilinears $H_{k,p}^{(1)} a_k^\dagger a_p$ and $H_{k,p}^{(2)} a_k^\dagger a_p$ corresponding to some operators $H^{(1)}$ and $H^{(2)}$, we have that,

$$
\langle \Omega | H^{(1)} H^{(2)} | \Omega \rangle = \int \frac{\mathrm{d}k_1}{2\pi} \frac{\mathrm{d}k_2}{2\pi} \frac{\mathrm{d}k_3}{2\pi} \frac{\mathrm{d}k_4}{2\pi} H_{k_1,k_2}^{(1)} H_{k_3,k_4}^{(2)} \langle \Omega | a_{k_1}^\dagger a_{k_2} a_{k_3}^\dagger a_{k_4} | \Omega \rangle
\tag{176}
$$

and performing the contractions (174) using Wick's theorem, we have that the connected part (neglecting the contraction between $k_1 - k_2$ and $k_3 - k_4$) reads

$$
\begin{aligned}
&\langle \Omega | H^{(1)} H^{(2)} | \Omega \rangle^c = \\
&= \int \frac{\mathrm{d}k}{2\pi} \frac{\mathrm{d}p}{2\pi} \frac{|K(k)|^2}{1 + |K(k)|^2} \frac{1}{1 + |K(p)|^2} H_{k,p}^{(1)} H_{p,k}^{(2)} + \frac{K^*(k)}{1 + |K(k)|^2} \frac{K(p)}{1 + |K(p)|^2} H_{k,p}^{(1)} H_{-k,-p}^{(2)} \,.
\end{aligned}
\tag{177}
$$

For three bilinears $H^{(1)}_{k,p} a^\dagger_k a_p$, $H^{(2)}_{k,p} a^\dagger_k a_p$ and $H^{(3)}_{k,p} a^\dagger_k a_p$, we already focus on the connected part $\langle \Omega | H^{(1)} H^{(2)} H^{(3)} | \Omega \rangle^c$. In order to extract this component, we cannot have contractions which link $a/a^\dagger$ within the same operator $H^{(i)}$, $i = 1, 2, 3$, as these would correspond to factorised expressions, like $\langle H^{(1)} \rangle \langle H^{(2)} \rangle \langle H^{(3)} \rangle$, $\langle H^{(1)} \rangle \langle H^{(2)} H^{(3)} \rangle$, etc. We have 8 sets of contractions using (174) and Wick's theorem, which give rise to the following expression for the connected component:

$$\langle \Omega | H^{(1)} H^{(2)} H^{(3)} | \Omega \rangle^c =$$

$$= \int \frac{dk_1}{2\pi} \frac{dk_2}{2\pi} \frac{dk_3}{2\pi} \left[ \frac{K^*(k_1)}{1 + |K(k_1)|^2} \frac{1}{1 + |K(k_2)|^2} \frac{K(-k_3)}{1 + |K(k_3)|^2} H^{(1)}_{k_1,k_2} H^{(2)}_{-k_1,k_3} H^{(3)}_{k_2,-k_3} \right.$$

$$- \frac{K^*(k_1)}{1 + |K(k_1)|^2} \frac{1}{1 + |K(k_2)|^2} \frac{K(-k_3)}{1 + |K(k_3)|^2} H^{(1)}_{k_1,k_2} H^{(2)}_{k_2,k_3} H^{(3)}_{-k_1,-k_3}$$

$$- \frac{|K(k_1)|^2}{1 + |K(k_1)|^2} \frac{1}{1 + |K(k_2)|^2} \frac{|K(k_3)|^2}{1 + |K(k_3)|^2} H^{(1)}_{k_1,k_2} H^{(2)}_{k_3,k_1} H^{(3)}_{k_2,k_3}$$

$$+ \frac{K^*(k_1)}{1 + |K(k_1)|^2} \frac{K(-k_2)}{1 + |K(k_2)|^2} \frac{|K(k_3)|^2}{1 + |K(k_3)|^2} H^{(1)}_{k_1,k_2} H^{(2)}_{k_3,-k_2} H^{(3)}_{-k_1,k_3} \quad (178)$$

$$- \frac{K^*(k_1)}{1 + |K(k_1)|^2} \frac{K(-k_2)}{1 + |K(k_2)|^2} \frac{1}{1 + |K(k_3)|^2} H^{(1)}_{k_1,k_2} H^{(2)}_{-k_1,k_3} H^{(3)}_{k_3,-k_2}$$

$$+ \frac{|K(k_1)|^2}{1 + |K(k_1)|^2} \frac{K(-k_2)}{1 + |K(k_2)|^2} \frac{K^*(k_3)}{1 + |K(k_3)|^2} H^{(1)}_{k_1,k_2} H^{(2)}_{k_3,k_1} H^{(3)}_{-k_3,-k_2}$$

$$+ \frac{|K(k_1)|^2}{1 + |K(k_1)|^2} \frac{1}{1 + |K(k_2)|^2} \frac{1}{1 + |K(k_3)|^2} H^{(1)}_{k_1,k_2} H^{(2)}_{k_2,k_3} H^{(3)}_{k_3,k_1}$$

$$\left. - \frac{|K(k_1)|^2}{1 + |K(k_1)|^2} \frac{K(-k_2)}{1 + |K(k_2)|^2} \frac{K^*(k_3)}{1 + |K(k_3)|^2} H^{(1)}_{k_1,k_2} H^{(2)}_{k_3,-k_2} H^{(3)}_{-k_3,k_1} \right].$$

Finally, for what follows, it is useful to consider some additional distribution identities, namely

$$\int \frac{dk'}{2\pi i} e^{ik'p't} P \frac{1}{k'} (k')^n = \frac{\partial^n_{p'}}{(it)^n} i \int \frac{dk'}{2\pi} e^{ik'p't} P \frac{1}{k'} = \begin{cases} \frac{1}{2} \mathrm{sgn}(tp') & \text{if } n = 0 \\ \mathrm{sgn}(t)(it)^{-n} \delta^{(n-1)}(p') & \text{if } n > 0, \end{cases}$$

$$(179)$$

where we used that $\partial_{p'} 1/2\, \mathrm{sgn}(tp') = t\delta(tp') = t/|t|\delta(p') = \mathrm{sgn}(t)\delta(p')$.

## 5.1 Current 1-pt function and its expansion and approach to the NESS

Below we compute quantities like $\langle \Omega | j(0,t) \tilde{Q} | \Omega \rangle^c$ and $\langle \Omega | j(0,t) \tilde{Q}^2 | \Omega \rangle^c$, which correspond to the 1st and 2nd order expansions of $\langle \Omega | j(0,t) | \tilde{\Omega} \rangle^c_{2\lambda}$ and $_\lambda \langle \tilde{\Omega} | j(0,t) | \tilde{\Omega} \rangle^c_\lambda$ (cf. (166)-(167)). To this aim, we will use (177) and (178), with $H^{(i)}_{k,p}$ specialized to the corresponding expression for either $j$ or $\tilde{Q}$.

### 5.1.1 First order in $\lambda$ of $\langle \Omega | j(0,t) | \tilde{\Omega}^* \rangle_{2\lambda}$

According to (166), this is equivalent to the expectation value $\langle \Omega | j(0,t) \tilde{Q} | \Omega \rangle^c$. Using (177), we have

$$\langle \Omega | j(0,t) \tilde{Q} | \Omega \rangle^c = \int \frac{dk}{2\pi} \frac{dp}{2\pi} \frac{|K(k)|^2}{1 + |K(k)|^2} \frac{1}{1 + |K(p)|^2} \left( e^{i(E_k - E_p)t}(k+p) \right) \left( \pi\delta(p-k) - i\frac{P}{p-k} \right)$$

$$+ \frac{K^*(k)}{1 + |K(k)|^2} \frac{K(p)}{1 + |K(p)|^2} \left( e^{i(E_k - E_p)t}(k+p) \right) \left( \pi\delta(-k+p) - i\frac{P}{-k+p} \right).$$

$$(180)$$

Before the eventual evaluation of (180) let us briefly comment on a technical aspect. Notice that we used the last line of (170), (i.e., a Dirac-$\delta$ and a principal value contributions) in (180), which is more transparent to deal with and whose application is justified by the following considerations. Namely, one could equivalently use the representation $\lim_{\epsilon \to 0} -i/((k-p)-i\epsilon)$, which corresponds to integrating the charge density up to a finite $X$ which is sent to infinity as $\epsilon$ is sent to zero. In this procedure to arrive at the last line of (170), one has to apply contour manipulations and infinitesimal shifts of integration variables by $\pm i\epsilon$. This step, however, makes the integration wrt. $p$ divergent due to the oscillating factor $\exp(itp^2)$. The integral can be regularised by multiplying the integrand of (180) by $\exp(-\delta(k^2+p^2))$ with some small and positive $\delta > t\epsilon$. This regularisation is in fact quite natural as it can be interpreted as a cut-off for the maximal time we can take. In addition, the procedure of sending first $\epsilon$ to zero and then $\delta$ to zero, which we have implicitly carried out to arrive at (180), is in accordance with the physically motivated order of limits, that is when first the subsystem size $X$ is sent to $\infty$ and afterwards the time variable $T$. Note also that regularising the integrand this way makes the integral also convergent for arbitrary currents of local conserved charges which would involve polynomials of $k$ and $p$.

Turning back to analysing (180), we can see that non-vanishing contributions in (180) only come from the principal value integrals which we now evaluate explicitly as the quadratic dispersion relation $E_k = k^2$ facilitates an easy and transparent analysis of the above expression. We rewrite the integrals as

$$
\begin{aligned}
\langle \Omega | j(0,t)\tilde{Q}|\Omega\rangle^c &= \int \frac{dk}{2\pi}\frac{dp}{2\pi} e^{i(E_k - E_p)t} i\mathrm{P}\frac{1}{k-p}\frac{|K(k)|^2 + K^*(k)K(p)}{(1+|K(k)|^2)(1+|K(p)|^2)} \\
&= \frac{1}{2}\int \frac{dk'}{2\pi}\frac{dp'}{2\pi} e^{ik'p't} p' i\mathrm{P}\frac{1}{k'}\frac{|K(\frac{k'+p'}{2})|^2 + K^*(\frac{k'+p'}{2})K(\frac{p'-k'}{2})}{(1+|K(\frac{k'+p'}{2})|^2)(1+|K(\frac{p'-k'}{2})|^2)}
\end{aligned}
\tag{181}
$$

where we introduced new variables $k' = k - p, p' = k + p$, with Jacobian $1/2$ and $E_k - E_p = k'p'$. The next step is to perform the integral wrt. $k'$, assuming $K(\frac{k'\pm p'}{2})$ functions can be expanded into Taylor series wrt. $k'$. When $t \to \infty$, we end up with

$$
\langle \Omega | j(0,\infty)\tilde{Q}|\Omega\rangle^c = -2\int \frac{dk}{2\pi}|k|\frac{|K(k)|^2}{(1+|K(k)|^2)^2},
\tag{182}
$$

due to the properties of $|K(k)|^2$. It is now easy to check that this expression is equals (289), that is, the 1st-order in $\lambda$ of the GGE expectation value $\langle j\rangle_{\rho_{\mathrm{bpp}}}$ after the partitioning protocol, using $v(k) = 2k$.

Regarding the time evolution, we have to consider sub-leading terms in (181), which can be addressed using (179) (with the order $n$ term giving rise to a $t^{-n}$ contribution). It is easy to see that the $n = 1$ term (179) gives zero since $K(p = 0) = 0$ and is an odd function. Furthermore, one can also show that the $n = 2$ term is zero as well, that is, the first non-trivial term is the $n = 3$ contribution coming from Eq. (181), given by

$$
\langle \Omega | j(0,t)\tilde{Q}|\Omega\rangle^c =
$$
$$
\partial_\lambda \langle j\rangle_{\rho_{\mathrm{bpp}}}|_{\lambda=0} - \frac{t^{-3}}{i^3 48\pi}\left(\frac{3}{2}K^{*'}(0)K^{(3)}(0) + \frac{1}{2}K^{*(3)}(0)K'(0) - 6K^{*'}(0)^2 K'(0)^2\right) + \mathcal{O}(t^{-4}),
\tag{183}
$$

that is, the convergence of the current expectation value to its GGE value is $\mathcal{O}(t^{-3})$ at the first order in $\lambda$.

### 5.1.2   First order in $\lambda$ of $_\lambda\langle\tilde{\Omega}|j(0,t)|\tilde{\Omega}\rangle_\lambda$

When considering the symmetric expectation value $_\lambda\langle\tilde{\Omega}|j(0,t)|\tilde{\Omega}\rangle_\lambda^c$ we also have to deal with $\langle\Omega|\tilde{Q}\,j(0,t)|\Omega\rangle^c$. This contribution yields asymptotically

$$\langle\Omega|\tilde{Q}\,j(0,t)|\Omega\rangle^c = \left(\langle\Omega|j(0,t)\tilde{Q}|\Omega\rangle^c\right)^* =$$
$$= -2\int\frac{dk}{2\pi}|k|\frac{|K(k)|^2}{(1+|K(k)|^2)^2} + \frac{t^{-3}}{\mathrm{i}^3 48\pi}\left(\frac{1}{2}K^{*'}(0)K^{(3)}(0) + \frac{3}{2}K^{*(3)}(0)K'(0) - 6K^{*'}(0)^2 K'(0)^2\right),$$
(184)

up to $\mathcal{O}(t^{-4})$ corrections, hence

$$\partial_\lambda\frac{\langle\Omega|e^{\lambda/2\tilde{Q}}j(0,t)e^{\lambda/2\tilde{Q}}|\Omega\rangle}{\langle\Omega|e^{\lambda\tilde{Q}}|\Omega\rangle}|_{\lambda=0} =$$
$$= \partial_\lambda\langle j\rangle_{\rho_{\mathrm{bpp}}}|_{\lambda=0} - \frac{t^{-3}}{\mathrm{i}^3 96\pi}\left(K^{*'}(0)K^{(3)}(0) - K^{*(3)}(0)K'(0)\right) + \mathcal{O}(t^{-4}).$$
(185)

It is worthwhile to note that as long as $K(k) = e^{\mathrm{i}\delta}K_0(k)$ with a real function $K_0$ and a real number $\delta$, the $t^{-3}$ type time-dependence is absent in the symmetric expression, and one can check that the $t^{-4}$ term is the leading and non-vanishing component. For generic $K(k) = K_1(k) + \mathrm{i}K_2(k)$, the $t^{-3}$ time-dependence is the leading one.

### 5.1.3   Second order in $\lambda$ of $\langle\Omega|j(0,t)|\tilde{\Omega}^*\rangle_{2\lambda}$

This quantity at second order in $\lambda$ is equivalent to $\langle\Omega|j(0,t)\tilde{Q}^2|\Omega\rangle^c$ (cf. (166)). Specifying (178) to the case of interest, we arrive at

$$\langle\Omega|j(0,t)\tilde{Q}^2|\Omega\rangle^c = \int\frac{dk_1}{2\pi}\frac{dk_2}{2\pi}\frac{dk_3}{2\pi}\frac{1}{1+|K(k_1)|^2}\frac{1}{1+|K(k_2)|^2}\frac{1}{1+|K(k_3)|^2}\times f_{k_1,k_2}(t)\times$$
$$\left[K^*(k_1)K(-k_3)\left(\pi\delta(k_1+k_3) + \mathrm{iP}\frac{1}{k_1+k_3}\right)\left(\pi\delta(k_2+k_3) - \mathrm{iP}\frac{1}{k_2+k_3}\right)\right.$$
$$-K^*(k_1)K(-k_3)\left(\pi\delta(k_2-k_3) - \mathrm{iP}\frac{1}{k_2-k_3}\right)\left(\pi\delta(k_1-k_3) + \mathrm{iP}\frac{1}{k_1-k_3}\right)$$
$$-|K(k_1)|^2|K(k_3)|^2\left(\pi\delta(k_3-k_1) - \mathrm{iP}\frac{1}{k_3-k_1}\right)\left(\pi\delta(k_2-k_3) - \mathrm{iP}\frac{1}{k_2-k_3}\right)$$
$$+K^*(k_1)K(-k_2)|K(k_3)|^2\left(\pi\delta(k_2+k_3) - \mathrm{iP}\frac{1}{k_2+k_3}\right)\left(\pi\delta(k_1+k_3) + \mathrm{iP}\frac{1}{k_1+k_3}\right)$$
(186)
$$-K^*(k_1)K(-k_2)\left(\pi\delta(k_1+k_3) + \mathrm{iP}\frac{1}{k_1+k_3}\right)\left(\pi\delta(k_3+k_2) - \mathrm{iP}\frac{1}{k_3+k_2}\right)$$
$$+|K(k_1)|^2 K(-k_2)K^*(k_3)\left(\pi\delta(k_3-k_1) - \mathrm{iP}\frac{1}{k_3-k_1}\right)\left(\pi\delta(k_3-k_2) + \mathrm{iP}\frac{1}{k_3-k_2}\right)$$
$$+|K(k_1)|^2\left(\pi\delta(k_2-k_3) - \mathrm{iP}\frac{1}{k_2-k_3}\right)\left(\pi\delta(k_3-k_1) - \mathrm{iP}\frac{1}{k_3-k_1}\right)$$
$$\left.-|K(k_1)|^2 K(-k_2)K^*(k_3)\left(\pi\delta(k_2+k_3) - \mathrm{iP}\frac{1}{k_2+k_3}\right)\left(\pi\delta(k_3+k_1) + \mathrm{iP}\frac{1}{k_3+k_1}\right)\right],$$

where we kept the shorthand $f_{k_1,k_2}(t) = \exp[\mathrm{i}t(E_{k_1} - E_{k_2})](E_{k_1} - E_{k_2})/(k_1 - k_2)$ as in (169). When arriving at (186) with Dirac-$\delta$ and principal value terms accounting for $\tilde{Q}$, we implicitly implemented the procedure we have discussed in 5.1.1. It is easy to see that contributions with two Dirac-$\delta$-s or two principal value integrals are zero. The first type because $f_{k_1,k_2}(t)$ reduces

to $f_{k_1,k_1} = v(k)$ (cf. (70)) and $v(k)$ is an odd function; and the second type because of $k_i \to -k_i$ ($i = 1, 2, 3$) transformation. That is, we only have to deal with cross-terms.

There is, however, one subtlety, which is very important to stress although it does not change the aforementioned conclusions neither the evaluation of the relevant terms in the topical case. Namely, in (186) the order of integration matters, and although it is implicit, the order is such that first the $k_1$ or $k_2$ integration is performed. This remark will only be important later on in Appendix D.2.2.

Turning back to the evaluation of (186) and without specifying the details of the computations, we present the resulting expressions for each line of (186) obtained by evaluating the cross-terms (i.e., a Dirac-$\delta$ term multiplied by a principal value), where the asymptotic time-dependence is obtained following a similar logic to the previous subsection. Denoting the lines of (186) by L1, L2, etc., we have for L1 in particular that

$$
\begin{aligned}
L1 = {} & \frac{\mathrm{i}}{2} \int \frac{\mathrm{d}k_1}{2\pi} \frac{\mathrm{d}k_2}{2\pi} \frac{|K(k_1)|^2}{(1+|K(k_1)|^2)^2} \frac{1}{1+|K(k_2)|^2} e^{\mathrm{i}t(E_{k_1}-E_{k_2})} (k_1+k_2) \mathrm{P}\frac{1}{k_1-k_2} \\
& + \frac{\mathrm{i}}{2} \int \frac{\mathrm{d}k_1}{2\pi} \frac{\mathrm{d}k_2}{2\pi} \frac{K^*(k_1)}{1+|K(k_1)|^2} \frac{K(k_2)}{(1+|K(k_2)|^2)^2} e^{\mathrm{i}t(E_{k_1}-E_{k_2})} (k_1+k_2) \mathrm{P}\frac{1}{k_1-k_2} .
\end{aligned}
\tag{187}
$$

One can check that $L1 = L2$ and $L6 = L8$ and, focusing on the leading temporal behaviour, we can extract two distinct contributions to which each line contributes equally. Namely,

$$
L1 + L2 + L5 + L7 = -2 \int \frac{\mathrm{d}p}{2\pi} 2|p| \frac{|K(p)|^2}{(1+|K(p)|^2)^3} + \mathcal{O}(t^{-3})
\tag{188}
$$

where adding up L1 and L2, and L5 and L7 certain $\mathcal{O}(t^{-2})$ terms cancel each other. From the other lines we have, similarly,

$$
L3 + L4 + L6 + L8 = 2 \int \frac{\mathrm{d}p}{2\pi} 2|p| \frac{|K(p)|^4}{(1+|K(p)|^2)^3} + \mathcal{O}(t^{-3}).
\tag{189}
$$

Altogether, we have that for

$$
\langle \Omega | j(0,t) \tilde{Q}^2 | \Omega \rangle^c = -2 \int \frac{\mathrm{d}p}{2\pi} 2|p| \frac{|K(p)|^2}{(1+|K(p)|^2)^3} + 2 \int \frac{\mathrm{d}p}{2\pi} 2|p| \frac{|K(p)|^4}{(1+|K(p)|^2)^3} + \mathcal{O}(t^{-3}).
\tag{190}
$$

We note that the stationary value agrees with the GGE prediction Eq. (290), that is

$$
\partial_\lambda^2 \langle j \rangle_{\rho_{\mathrm{bpp}}} |_{\lambda=0} = \langle \Omega | j(0,\infty) \tilde{Q}^2 | \Omega \rangle^c ,
\tag{191}
$$

using $v(k) = 2k$. It is straightforward to compute the numerical coefficients for the $\mathcal{O}(t^{-3})$ term and to check that the 8 lines do not cancel each other, hence

$$
\begin{aligned}
& \langle \Omega | j(0,t) \tilde{Q}^2 | \Omega \rangle^c = \\
& = \partial_\lambda^2 \langle j \rangle_{\rho_{\mathrm{bpp}}} |_{\lambda=0} - \frac{t^{-3}}{\mathrm{i}^3 96\pi} \left( 3K^{*'}(0)K^{(3)}(0) + K^{*(3)}(0)K'(0) - 24K^{*'}(0)^2 K'(0)^2 \right) + \mathcal{O}(t^{-4}).
\end{aligned}
\tag{192}
$$

### 5.1.4 Second order in $\lambda$ of $_\lambda\langle \tilde{\Omega} | j(0,t) | \tilde{\Omega} \rangle_\lambda$

In order to evaluate the 2nd order expansion of the symmetric quantity $_\lambda\langle \tilde{\Omega} | j(0,t) | \tilde{\Omega} \rangle_\lambda^c$, we need to compute two additional correlation functions besides $\langle \Omega | j(0,t) \tilde{Q}^2 | \Omega \rangle^c$, namely $\langle \Omega | \tilde{Q}^2 j(0,t) | \Omega \rangle^c$ and $\langle \Omega | \tilde{Q} j(0,t) \tilde{Q} | \Omega \rangle^c$. For $\langle \Omega | \tilde{Q}^2 j(0,t) | \Omega \rangle^c$ we find that

$$
\begin{aligned}
& \langle \Omega | \tilde{Q}^2 j(0,t) | \Omega \rangle^c = \left( \langle \Omega | j(0,t) \tilde{Q}^2 | \Omega \rangle^c \right)^* = \\
& \partial_\lambda^2 \langle j \rangle_{\rho_{\mathrm{bpp}}} |_{\lambda=0} + \frac{t^{-3}}{\mathrm{i}^3 96\pi} \left( K^{*'}(0)K^{(3)}(0) + 3K^{*(3)}(0)K'(0) - 24K^{*'}(0)^2 K'(0)^2 \right) + \mathcal{O}(t^{-4}),
\end{aligned}
\tag{193}
$$

and for $\langle\Omega|\tilde{Q}\,j(0,t)\tilde{Q}|\Omega\rangle^c$, repeating the computation of the previous subsection, we have that

$$\langle\Omega|\tilde{Q}\,j(0,t)\tilde{Q}|\Omega\rangle^c = \partial_\lambda^2\langle j\rangle_{\rho_{\rm bpp}}|_{\lambda=0} - \frac{t^{-3}}{{\rm i}^3 96\pi}\left(K^{*\prime}(0)K^{(3)}(0) - K^{*(3)}(0)K^\prime(0)\right) + \mathcal{O}(t^{-4}). \quad (194)$$

Summing up every bit, the second order expansion of $_\lambda\langle\tilde{\Omega}|j(0,t)|\tilde{\Omega}\rangle_\lambda^c$ gives

$$\partial_\lambda^2\frac{\langle\Omega|e^{\lambda/2\tilde{Q}}j(0,t)e^{\lambda/2\tilde{Q}}|\Omega\rangle}{\langle\Omega|e^{\lambda\tilde{Q}}|\Omega\rangle}|_{\lambda=0} = \quad (195)$$
$$= \partial_\lambda^2\langle j\rangle_{\rho_{\rm bpp}}|_{\lambda=0} - \frac{t^{-3}}{{\rm i}^3 96\pi}\left(K^{*\prime}(0)K^{(3)}(0) - K^{*(3)}(0)K^\prime(0)\right) + \mathcal{O}(t^{-4}).$$

Similarly to the first order expansion of the current, it can again be note that as long as $K(k) = e^{{\rm i}\delta}K_0(k)$ with a real function $f_0$ and a real number $\delta$, the $t^{-3}$ type time-dependence is absent in the symmetric expression, and one can check that the $t^{-4}$ term is the leading and non-vanishing component. For generic $K(k) = K_1(k) + {\rm i}K_2(k)$, the $t^{-3}$ time-dependence is the leading one.

# 6 The current 2-pt function after biased free fermion quenches

The main goal of this section is to demonstrate that the temporal cluster property or condition iii) holds for integrable free fermion quenches. Additionally, as we shall see, we can also explicitly highlight the validity of (44), that is, the convergence of autocorrelation type multi-point functions to the NESS, in an appropriate sense.

In order to carry out microscopic computations (similarly to the previous one), we perform an expansion in the exponential $e^{\lambda\tilde{Q}}$ or in $|\tilde{\Omega}\rangle_\lambda$ wrt. the counting field and proceed order by order to compute $\langle\Omega|j(0,t)j(0,t^\prime)|\tilde{\Omega}^*\rangle_{2\lambda}^c$ and $_\lambda\langle\tilde{\Omega}|j(0,t)j(0,t^\prime)|\tilde{\Omega}\rangle_\lambda^c$, that is, for the deformed integrable initial states.

Similarly to the expansion of the current expectation value the expansion may not converge for certain times, in particular when $t = t^\prime = 0$ and also for $t = t^\prime$. In this case, however, the reason is not the discontinuous state at $t = 0$. The only terms that diverge at these time instances, at least up to the order we carried out our calculations, are the ones that are also present in the expansion of the 2-pt function evaluated at the NESS. In fact the current-current autocorrelation function in the NESS (using the time translation invariance of the NESS) itself diverges when $t = t^\prime$.

Omitting such time instances from our analysis, we concentrate on the asymptotic time dependence of this 2-pt function at zeroth and first order in $\lambda$ and our goal is to show that the required strong enough temporal cluster property iii) is satisfied. First we focus on the non-symmetric expectation value $\langle\Omega|j(0,t)j(0,t^\prime)|\tilde{\Omega}^*\rangle_{2\lambda}^c$ and afterwards discuss the symmetric case $_\lambda\langle\tilde{\Omega}|j(0,t)j(0,t^\prime)|\tilde{\Omega}\rangle_\lambda^c$. Similarly to the case of 1-pt functions, we can use the fact that when the initial states are expanded wrt. $\lambda$, calculating the fully connected correlation functions, now also including $\tilde{Q}$, takes into account the normalisation of the deformed states. To compute quantities like $\langle\Omega|j(0,t)j(0,t^\prime)|\Omega\rangle^c$ and $\langle\Omega|j(0,t)j(0,t^\prime)\tilde{Q}|\Omega\rangle^c$ we use (177) and (178). In these expressions $H_{k,p}^{(1)}$ and $H_{k,p}^{(2)}$ are replaced by $f_{k,p}(t)$ and $f_{k,p}(t^\prime)$, and $H_{k,p}^{(3)}$ by $\left(\pi\delta(p-k) - {\rm i}{\rm P}\frac{1}{p-k}\right)$.

## 6.1 Zeroth order in $\lambda$ of $_\lambda\langle\tilde{\Omega}|j(0,t)j(0,t^\prime)|\tilde{\Omega}\rangle_\lambda^c$

At the zeroth order this is equivalent to compute the 2-point function $\langle\Omega|j(0,t)j(0,0)|\Omega\rangle = \langle\Omega|j(0,t)j(0,0)|\Omega\rangle^c$ as the expectation value of $j$ is zero in the homogeneous state. Substituting into (177) it is immediate

that

$$\langle\Omega|j(0,t)j(0,t')|\Omega\rangle^c = \int \frac{dk}{2\pi}\frac{dp}{2\pi} \frac{|K(k)|^2}{1+|K(k)|^2} \frac{1}{1+|K(p)|^2} e^{i(E_k-E_p)(t-t')}(k+p)^2$$
$$- \int \frac{dk}{2\pi}\frac{dp}{2\pi} \frac{K^*(k)}{1+|K(k)|^2} \frac{K(p)}{1+|K(p)|^2} e^{i(E_k-E_p)(t+t')}(k+p)^2 . \tag{196}$$

From the above integrals via asymptotic analysis it is easy to obtain $\mathcal{O}(t_+^{-3})$ and $\mathcal{O}(t_-^{-3})$ behaviours (with $t_\pm = t \pm t'$), and as expected, the integral for $t_-$ is divergent when $t_- = 0$.

## 6.2 First order in $\lambda$ of $\langle\Omega|j(0,t)j(0,t')|\tilde{\Omega}^*\rangle^c_{2\lambda}$

This is equivalent to evaluating $\langle\Omega|j(0,t)j(0,t')\tilde{Q}|\Omega\rangle^c$. Using (178) as well as performing the thermodynamic limit, we can express it as

$$\langle\Omega|j(0,t)j(0,t')\tilde{Q}|\Omega\rangle^c = \int \frac{dk_1}{2\pi}\frac{dk_2}{2\pi}\frac{dk_3}{2\pi} \times$$

$$\times \Bigg[ \frac{K^*(k_1)}{1+|K(k_1)|^2}\frac{1}{1+|K(k_2)|^2}\frac{K(-k_3)}{1+|K(k_3)|^2} f_{k_1,k_2}(t)f_{-k_1,k_3}(t')\left(\pi\delta(k_2+k_3)-iP\frac{1}{k_2+k_3}\right)$$

$$- \frac{K^*(k_1)}{1+|K(k_1)|^2}\frac{1}{1+|K(k_2)|^2}\frac{K(-k_3)}{1+|K(k_3)|^2} f_{k_1,k_2}(t)f_{k_2,k_3}(t')\left(\pi\delta(k_1-k_3)+iP\frac{1}{k_1-k_3}\right)$$

$$- \frac{|K(k_1)|^2}{1+|K(k_1)|^2}\frac{1}{1+|K(k_2)|^2}\frac{|K(k_3)|^2}{1+|K(k_3)|^2} f_{k_1,k_2}(t)f_{k_3,k_1}(t')\left(\pi\delta(k_2-k_3)-iP\frac{1}{k_2-k_3}\right)$$

$$+ \frac{K^*(k_1)}{1+|K(k_1)|^2}\frac{K(-k_2)}{1+|K(k_2)|^2}\frac{|K(k_3)|^2}{1+|K(k_3)|^2} f_{k_1,k_2}(t)f_{k_3,-k_2}(t')\left(\pi\delta(k_1+k_3)+iP\frac{1}{k_1+k_3}\right)$$

$$- \frac{K^*(k_1)}{1+|K(k_1)|^2}\frac{K(-k_2)}{1+|K(k_2)|^2}\frac{1}{1+|K(k_3)|^2} f_{k_1,k_2}(t)f_{-k_1,k_3}(t')\left(\pi\delta(k_3+k_2)-iP\frac{1}{k_3+k_2}\right)$$

$$+ \frac{|K(k_1)|^2}{1+|K(k_1)|^2}\frac{K(-k_2)}{1+|K(k_2)|^2}\frac{K^*(k_3)}{1+|K(k_3)|^2} f_{k_1,k_2}(t)f_{k_3,k_1}(t')\left(\pi\delta(k_3-k_2)+iP\frac{1}{k_3-k_2}\right)$$

$$+ \frac{|K(k_1)|^2}{1+|K(k_1)|^2}\frac{1}{1+|K(k_2)|^2}\frac{1}{1+|K(k_3)|^2} f_{k_1,k_2}(t)f_{k_2,k_3}(t')\left(\pi\delta(k_3-k_1)-iP\frac{1}{k_3-k_1}\right)$$

$$- \frac{|K(k_1)|^2}{1+|K(k_1)|^2}\frac{K(-k_2)}{1+|K(k_2)|^2}\frac{K^*(k_3)}{1+|K(k_3)|^2} f_{k_1,k_2}(t)f_{k_3,-k_2}(t')\left(\pi\delta(k_3+k_1)+iP\frac{1}{k_3+k_1}\right)\Bigg]. \tag{197}$$

Below we proceed line by line of the expression above. In particular, we compute the $\delta$-contributions as it is easy to see via the $k_1 \to -k_1$, $k_2 \to -k_2$, $k_3 \to -k_3$ transformation that the contributions originating from the principal value integrals are identically zero.

Eq. (197) gives rise to two types of temporal contribution, functions of $t_\pm = t \pm t'$, respectively. It is not hard to see that $t_-$ type dependence originates from lines 2,3,6 and 7 of (197). Namely, from line 2 the term

$$L2 = \frac{1}{2}\int \frac{dk_1}{2\pi}\frac{dk_2}{2\pi} e^{i(E_{k_1}-E_{k_2})(t-t')}(k_1+k_2)^2 \frac{|K(k_1)|^2}{(1+|K(k_1)|^2)^2}\frac{1}{1+|K(k_2)|^2}, \tag{198}$$

from line 3

$$L3 = -\frac{1}{2}\int \frac{dk_1}{2\pi}\frac{dk_2}{2\pi} e^{i(E_{k_1}-E_{k_2})(t-t')}(k_1+k_2)^2 \frac{|K(k_1)|^2}{1+|K(k_1)|^2}\frac{|K(k_2)|^2}{(1+|K(k_2)|^2)^2}. \tag{199}$$

and from line 6 and 7

$$L6 = L3, \quad L7 = L2. \tag{200}$$

Collecting them together

$$
\begin{aligned}
\langle\Omega|j(0,t)j(0,t')\tilde{Q}|\Omega\rangle_{t_-}^c = & \int \frac{dk_1}{2\pi}\frac{dk_2}{2\pi} e^{i(E_{k_1}-E_{k_2})(t-t')}(k_1+k_2)^2 \frac{|K(k_1)|^2}{(1+|K(k_1)|^2)^2}\frac{1}{1+|K(k_2)|^2} \\
& - \int \frac{dk_1}{2\pi}\frac{dk_2}{2\pi} e^{i(E_{k_1}-E_{k_2})(t-t')}(k_1+k_2)^2 \frac{|K(k_1)|^2}{1+|K(k_1)|^2}\frac{|K(k_2)|^2}{(1+|K(k_2)|^2)^2},
\end{aligned}
\tag{201}
$$

where the subscript $t_-$ means that in the rhs only the $t_-$ contribution is considered.

It is also immediate to see that $\langle\Omega|j(0,t)j(0,t')\tilde{Q}|\Omega\rangle_{t_-}^c$ equals the corresponding 1st order GGE results, cf. Eq. (297) in Appendix 297, that is

$$
\partial_\lambda \langle j(0,t)j(0,t')\rangle_{\rho_{\text{bpp}}}|_{\lambda=0} = \langle\Omega|j(0,t)j(0,t')\tilde{Q}|\Omega\rangle_{t_-}^c.
\tag{202}
$$

We can also analyze the corresponding time dependence in $t_-$. Exploiting the quadratic dispersion relation and using the integration variables $k_1' = k_1 - k_2$, $k_2' = k_1 + k_2$ (accounting for a Jacobian $1/2$), we can differentiate the expressions containing the $K$-functions wrt. $k_1'$, with the $n$-th derivative yields a term

$$
\frac{1}{(i(t-t'))^n}\delta^{(n)}(k_2'(t-t')) = \frac{\text{sgn}(t-t')}{i^n(t-t')^{n+1}}\delta^{(n)}(k_2').
\tag{203}
$$

Hence we have from $L2$ (and $L7$) a term $\mathcal{O}(|t_-|^{-3})$ dependence; and from $L3$ (and $L6$) a term $\mathcal{O}(|t_-|^{-5})$.

Eq (197) also gives rise to $t_+$-type temporal correlations. Proceeding analogously, we find that these contributions originate from lines 1, 4, 5 and 8 of (197), and more explicitly they are

$$
L1 = -\frac{1}{2}\int \frac{dk_1}{2\pi}\frac{dk_2}{2\pi}\frac{K^*(k_1)}{1+|K(k_1)|^2}\frac{K(k_2)}{(1+|K(k_2)|^2)^2}e^{i(E_{k_1}-E_{k_2})(t+t')}(k_1+k_2)^2 = \mathcal{O}(t_+^{-3}),
\tag{204}
$$

$$
L4 = \frac{1}{2}\int \frac{dk_1}{2\pi}\frac{dk_2}{2\pi}\frac{K^*(k_1)|K(k_1)|^2}{(1+|K(k_1)|^2)^2}\frac{K(k_2)}{1+|K(k_2)|^2}e^{i(E_{k_1}-E_{k_2})(t+t')}(k_1+k_2)^2 = \mathcal{O}(t_+^{-4}),
\tag{205}
$$

and

$$
L5 = L1, \quad L8 = L4,
\tag{206}
$$

respectively. Hence

$$
\begin{aligned}
\langle\Omega|j(0,t)j(0,t')\tilde{Q}|\Omega\rangle_{t_+}^c = & \int \frac{dk_1}{2\pi}\frac{dk_2}{2\pi}\frac{K^*(k_1)|K(k_1)|^2}{(1+|K(k_1)|^2)^2}\frac{K(k_2)}{1+|K(k_2)|^2}e^{i(E_{k_1}-E_{k_2})(t+t')}(k_1+k_2)^2 \\
& - \int \frac{dk_1}{2\pi}\frac{dk_2}{2\pi}\frac{K^*(k_1)}{1+|K(k_1)|^2}\frac{K(k_2)}{(1+|K(k_2)|^2)^2}e^{i(E_{k_1}-E_{k_2})(t+t')}(k_1+k_2)^2 \\
= & \mathcal{O}(t_+^{-3}),
\end{aligned}
\tag{207}
$$

where the subscript $t_+$ means that in the rhs only the $t_+ = t + t'$ contribution is considered. These asymptotic expressions were obtained analogously to the case of the $t_-$-type correlations.

## 6.3   First order in $\lambda$ of $_\lambda\langle\tilde{\Omega}|j(0,t)j(0,t')|\tilde{\Omega}\rangle_\lambda^c$

When considering the symmetric expression $_\lambda\langle\tilde{\Omega}|j(0,t)j(0,t')|\tilde{\Omega}\rangle_\lambda^c$ at the first order in $\lambda$ we also need to deal with the term $\langle\Omega|\tilde{Q}j(0,t)j(0,t')|\Omega\rangle^c$. Exploiting the fact that

$$
\langle\Omega|\tilde{Q}\,j(0,t)j(0,t')|\Omega\rangle^c = \left(\langle\Omega|j(0,t')j(0,t)\tilde{Q}|\Omega\rangle^c\right)^*
\tag{208}
$$

we readily have that

$$\langle\Omega|\tilde{Q}j(0,t)j(0,t')|\Omega\rangle^c_{t_-} = \langle\Omega|j(0,t)j(0,t')\tilde{Q}|\Omega\rangle^c_{t_-} = \partial_\lambda\langle j(0,t)j(0,t')\rangle_{\rho_{\mathrm{bpp}}}|_{\lambda=0}\,, \quad (209)$$

and that

$$\langle\Omega|\tilde{Q}j(0,t)j(0,t')|\Omega\rangle^c_{t_+} = \int\frac{\mathrm{d}k_1}{2\pi}\frac{\mathrm{d}k_2}{2\pi}\frac{K^*(k_1)}{1+|K(k_1)|^2}\frac{|K(k_2)|^2K(k_2)}{(1+|K(k_2)|^2)^2}e^{\mathrm{i}(E_{k_1}-E_{k_2})(t+t')}(k_1+k_2)^2$$
$$-\int\frac{\mathrm{d}k_1}{2\pi}\frac{\mathrm{d}k_2}{2\pi}\frac{K^*(k_1)}{(1+|K(k_1)|^2)^2}\frac{K(k_2)}{1+|K(k_2)|^2}e^{\mathrm{i}(E_{k_1}-E_{k_2})(t+t')}(k_1+k_2)^2$$
$$=\mathcal{O}(t_+^{-3})\,. \tag{210}$$

The $\mathcal{O}(t_+^{-3})$ leading order contributions in $\langle\Omega|j(0,t)j(0,t')\tilde{Q}|\Omega\rangle^c_{t_+}$ and $\langle\Omega|\tilde{Q}j(0,t)j(0,t')|\Omega\rangle^c_{t_+}$ are identical.

## 6.4 Overall asymptotic time-dependence of the current 2-pt function

It is instructive to put together the pieces of the current 2-pt function and analyse its overall time dependence. Up to first order in $\lambda$, we have

$$_\lambda\langle\tilde{\Omega}|j(0,t)j(0,t')|\tilde{\Omega}\rangle^c_\lambda = \mathcal{G}_0^-(t_-) + \mathcal{G}_0^+(t_+) + \lambda\mathcal{G}_1^-(t_-) + \lambda\mathcal{G}_1^+(t_+) + \mathcal{O}(\lambda^2) =$$
$$= \int\frac{\mathrm{d}k}{2\pi}\frac{\mathrm{d}p}{2\pi}\frac{|K(k)|^2}{1+|K(k)|^2}\frac{1}{1+|K(p)|^2}e^{\mathrm{i}(E_k-E_p)(t-t')}(k_1+k_2)^2$$
$$-\int\frac{\mathrm{d}k}{2\pi}\frac{\mathrm{d}p}{2\pi}\frac{K^*(k)}{1+|K(k)|^2}\frac{K(p)}{1+|K(p)|^2}e^{\mathrm{i}(E_k-E_p)(t+t')}(k_1+k_2)^2$$
$$+\lambda\int\frac{\mathrm{d}k_1}{2\pi}\frac{\mathrm{d}k_2}{2\pi}\frac{|K(k_1)|^2(1-|K(k_1)|^2|K(k_2)|^2)}{(1+|K(k_1)|^2)^2(1+|K(k_2)|^2)^2}e^{\mathrm{i}(E_{k_1}-E_{k_2})(t-t')}(k_1+k_2)^2$$
$$+\lambda\int\frac{\mathrm{d}k_1}{2\pi}\frac{\mathrm{d}k_2}{2\pi}\frac{K^*(k_1)K(k_2)(1-|K(k_2)|^2|K(k_2)|^2)}{(1+|K(k_1)|^2)^2(1+|K(k_2)|^2)^2}e^{\mathrm{i}(E_{k_1}-E_{k_2})(t+t')}(k_1+k_2)^2 + \mathcal{O}(\lambda^2)\,, \tag{211}$$

where the functions $\mathcal{G}_n^\pm(t_\pm)$ correspond to the contributions in $t_\pm$ at order $n(=0,1)$ in $\lambda$. We recall that the asymptotic behaviour of $\mathcal{G}_{0,1}^+(t_+)$ is $\mathcal{O}(t_+^{-3})$, whereas of $\mathcal{G}_{0,1}^-(t_-)$ is $\mathcal{O}(|t_-|^{-3})$.

The first remark we wish to stress again is that, up to first order (included) in the counting field, the time translation invariant bit in $_\lambda\langle\tilde{\Omega}|j(0,t)j(0,t')|\tilde{\Omega}\rangle^c_\lambda$, i.e. $\mathcal{G}_0^-(t_-) + \lambda\mathcal{G}_1^-(t_-)$, equals the corresponding contribution in the NESS, described by the GGE $\rho_{\mathrm{bpp}}(\lambda)$. In other words, we can explicitly demonstrate that when $t_+\to\infty$, for all $t_-$,

$$\lim_{t_+\to\infty}{}_\lambda\langle\tilde{\Omega}|j(0,t)j(0,t')|\tilde{\Omega}\rangle^c_\lambda = \mathcal{G}_0(t_-) + \lambda\mathcal{G}_1(t_-) + \mathcal{O}(\lambda^2)$$
$$= (1+\lambda\,\partial_\lambda)\,\mathrm{Tr}\big[\rho_{\mathrm{bpp}}(\lambda)\,j(0,t_-)j(0,0)\big]^c\,|_{\lambda=0} + \mathcal{O}(\lambda^2)\,, \tag{212}$$

where $\rho_{\mathrm{bpp}}$ is normalised, and we exploited the time translation invariance of the NESS. This finding strongly confirms the reasoning used in section 2.3, where we argued for the convergence of *multi-point* functions in certain temporal domains in the hypercube spanned by $t_1, t_2, ...t_n > 0$.

Our second remark is that we can explicitly verify the fulfillment of iii) for $_\lambda\langle\tilde{\Omega}|j(0,t)j(0,t')|\tilde{\Omega}\rangle^c_\lambda$ at least up to the first non-trivial order in the counting field, using the asymptotic time dependence of $\mathcal{G}_{0,1}^\pm(t_\pm)$. In particular, what we have to show is that $t_-\,_\lambda\langle\tilde{\Omega}|j(0,t)j(0,t')|\tilde{\Omega}\rangle^c_\lambda \overset{t\to\infty}{\to} 0$ uniformly when $t, t' > 0$. First we recall that we were able to write the functions $\mathcal{G}_{0,1}^\pm(t_\pm)$ as

$$\left|\mathcal{G}_{0,1}^\pm(t_\pm) - \frac{j_{0,1}^\pm}{t_\pm^3}\right| \le \frac{|c_{0,1}^\pm|}{t_\pm^4}\,, \quad \text{for} \quad t_\pm > \tau_{0,1}^\pm\,, \tag{213}$$

for some positive $\tau_{0,1}^{\pm}$ and some positive constants $c_{0,1}^{\pm}$, and for $j_{0,1}^{\pm}$ being the leading terms in $t^{\pm}$ of the integrals associated to $\mathcal{G}_{0,1}^{\pm}(t_{\pm})$. We need to study the behavior of these functions when $t_{-}$ is sent to infinity. Clearly, as $\mathcal{G}_{0,1}^{-}$ is only a function of $t_{-}$, $t_{-}\mathcal{G}_{0,1}^{-}(t_{-}) \to 0$ which can be regarded as uniform convergence. Now, focusing on the functions $\mathcal{G}_{0,1}^{+}$ we can assume $t > t'$ without loss of generality, and write $t_{+} = (t - t') + 2t' = t_{-} + 2t'$. Therefore from (213), we have that

$$0 \leq \lim_{\substack{t_{-}\to\infty \\ t,t'>0}} \left| t_{-}\mathcal{G}_{0,1}^{+}(t_{+}) - t_{-}\frac{j_{0,1}^{+}}{(t_{-}+2t')^3} \right| \leq \lim_{\substack{t_{-}\to\infty \\ t,t'>0}} t_{-}\frac{|c_{0,1}^{+}|}{(t_{-}+2t')^4} \leq \lim_{\substack{t_{-}\to\infty \\ t,t'>0}} t_{-}\frac{|c_{0,1}^{+}|}{t_{-}^4} = 0 \qquad (214)$$

(where used that $t_{-}$ becomes larger than $\tau_{0,1}^{+}$ in the limit) and at the last inequality we can bound the expressions on lhs. uniformly, as $t' > 0$. That is, we have that,

$$0 \leq \left| \lim_{\substack{t_{-}\to\infty \\ t,t'>0}} t_{-}\mathcal{G}_{0,1}^{+}(t_{+}) \right| = \lim_{\substack{t_{-}\to\infty \\ t,t'>0}} t_{-}\frac{|j_{0,1}^{+}|}{(t_{-}+2t')^3} \leq \lim_{\substack{t_{-}\to\infty \\ t,t'>0}} t_{-}\frac{|j_{0,1}^{+}|}{t_{-}^3} = 0, \qquad (215)$$

where after the last inequality we can have a uniform bound for any $t' > 0$. Henceforth

$$0 \leq \lim_{\substack{t_{-}\to\infty \\ t,t'>0}} t_{-}\mathcal{G}_{0,1}^{+}(t_{+}) = 0, \qquad (216)$$

uniformly, and since the function is continuous, we can omit the absolute value.

Finally, we note that just by the asymptotic behaviour of the functions $\mathcal{G}_{0,1}^{\pm}(t_{\pm})$ it is easy to see their contribution to the second cumulant, that is when integrating $_{\lambda}\langle\tilde{\Omega}|j(0,t)j(0,t')|\tilde{\Omega}\rangle_{\lambda}^{c}$ wrt. to both $t$ and $t'$ from zero to $T$, cf. (37). In particular, we obtain $\mathcal{O}(T)$ contributions from $\mathcal{G}_{0,1}^{-}(t_{-})$ and $\mathcal{O}(1)$ contributions from $\mathcal{G}_{0,1}^{+}(t_{+})$. In other words, we can explicitly see that the NESS bits only contribute to the scaled cumulants as was demonstrated in Sections 2.3.

# 7 Conclusion and outlook

In this work we discussed a general framework and showed how it allows for the computation of the fluctuations or the full counting statistics (FCS) of conserved charges after quantum quenches in one-dimensional systems. Despite the non-trivial out-of-equilibrium problem, we demonstrated that the time-dependent FCS can be expressed in terms of the fluctuations within the initial state, and of the dynamical fluctuations of the current (associated with the conserved charge), still expressible via thermodynamic and hydrodynamic quantities. More specifically, we focused on the fluctuations of the integrated charge densities $\int_{0}^{X} \mathrm{d}x\, q(x,T)$ for large $X$ and "intermediate" times such that $\ell_{\text{micro}} \ll T \ll X$. That is, while $T$ is much smaller than $X$, it is much larger than all microscopic scales $\ell_{\text{micro}}$, so that the dynamics can be described by the emergent Euler hydrodynamics for ballistic propagation of conserved quantities. In this regime, assuming the validity of large-deviation principles with a linear behaviour, the scaled cumulant generating function shows a $T$-independent linear growth in $X$ with an $O(X^0)$ correction that scales as $T$ at the leading order. Our results show that the slope of the latter temporal behaviour can be obtained solely in terms of thermodynamic and hydrodynamic quantities. In particular, it is related to the fluctuation of the current in specific non-equilibrium stead-states, which can be evaluated using the Ballistic Fluctuation Theory and the Flow Equations (which can be further specified and solved explicitly for integrable systems cf. (74) and (75)).

The core idea of the hydrodynamic approach is to circumvent the problem of spatial long-range correlations present on $[0, X]$ at time $T$, which give the main difficulty for directly computing the

FCS of $\int_0^X dx\, q(x, T)$ after quenches. This is accomplished via suitable contour-manipulations inspired by [53], exploiting the continuity equation of Noether charges. As shown in this paper, characterising the fluctuations of $\int_0^X dx\, q(x, T)$ boils down to dealing with that of $\int_0^X dx\, q(x, 0)$ in the initial state $|\Psi\rangle$ and $\int_0^T dt\, j(t)$ after the paradigmatic partitioning protocol, i.e., joining two semi-infinite systems with different homogeneous states. The corresponding inhomogeneous state is a charge-biased, modified version of the original initial state $|\Psi\rangle$, proportional to $\exp(\lambda \int_0^\infty dx\, q(x, 0))|\Psi\rangle$ where $\lambda$ is the counting field entering the FCS generating function. This recognition corresponds to what is phrased in (8), and is referred to as a non-trivial hydrodynamic identity. As argued, under suitable circumstances, one can make a further step showing that this is equivalent to evaluating the current fluctuation in the non-equilibrium steady-state emerging after the aforementioned partitioning protocol. The advantage is that this non-equilibrium steady-state is not affected by spatial long-range correlations. Specifying conditions that guarantee that there are no new dynamical long-range correlations on time $[0, T]$, and specifying this non-equilibrium steady-state, are the first new results of our work. Our manipulations provide a simple explanation for how such an inhomogeneous problem encodes the physics of the time-evolved FCS after a homogeneous quench. If the steady-state is a maximal entropy state, as generally assumed, the lack of dynamical long-range correlations guarantees that the current fluctuations are encoded by thermodynamic and hydrodynamic quantities via BFT, and in particular by the hydrodynamic formula (9). The evaluation of (9) (that is characterising the non-trivial current fluctuations) is possible by much simpler and tractable means if the system is integrable, i.e., through the explicit form of the Flow Equations dictated by BFT.

The applicability of the fully hydrodynamic picture, formula (9), necessitates the fulfillment of three conditions, out of which only the first one is required for the hydrodynamic identity (8): the strong enough spatial clustering of the initial state $|\Psi\rangle$ (wrt. local operators); the relaxation to the NESS after the bipartite quench; and a strong enough temporal cluster property of the current operators in the bipartite state, $\langle\Psi| \exp(\lambda \int_0^\infty dx\, q(x, 0))j(t)j(t')\exp(\lambda \int_0^\infty dx\, q(x, 0))|\Psi\rangle^c$. Here $j$ is the current associated with $Q$, and we assume the temporal cluster property to hold for higher connected multi-point functions as well. Also local relaxation has to be phrased slightly more precisely, i.e., we assume that generic local operators, and also certain quasi-local operators relax to a maximal entropy ensemble.

It is worth clarifying the importance of these conditions and commenting on the different roles they play concerning various aspects. These range from conceptual ones, such as how they contribute to the derivation of the hydrodynamic picture or how well-established certain related manipulations are in the derivation, to practical considerations such as the difficulty of checking their fulfillment and the predictive power of the hydrodynamic framework they allow for.

- The strong cluster property of the initial states is an important physical requirement, practically it is often easy to check and in fact most natural initial state satisfy this condition. This property was crucial for some steps in our derivation, in particular, for the "Large-X: separation between left- and right-contributions" and the " Asymptotic Commutativity" property, cf. Sec. 2.3 and Eqs. (29) and (33), accordingly.

  In particular, based on this condition, the validity of the "Separation" step was rigorously demonstrated for generic systems with a finite Lieb-Robinson (LR) velocity. For infinite LR velocity, we analysed the case of non-relativistic free fermions in a perturbative manner (wrt. the counting field). The behaviour of the lowest order contributions is strongly indicative that this property holds at any order. An intuitive argument can also be put forward, namely, if the occupation of high-velocity modes are suppressed enough after the quench, then relevant physical properties of the system shall not be different from the case of finite LR velocity (cf. Appendix A to link short range correlations of the initial state and decay of the mode occupation). This intuition and the analysis of the free fermion case makes it plausible that

the "Separation" step is valid in any short-ranged initial state and for generic short-range Hamiltonians also admitting an infinite LR velocity.

The justification of the "Asymptotic Commutativity" property was carried out only for the more general case of the infinte LR velocity but relying on the free-fermion theory again assuming exponentially clustering but otherwise rather generic states. In this setting, we have performed an order-by-order treatment up to third order in the counting field. The structure of the terms and how the clustering correlation functions result in negligible corrections for the factorised expression again promotes a more general validity of this property. However, a broader analysis including interacting systems (with infinite LR velocity) would be clearly desirable.

Additionally, we would like to briefly comment on possible connections among the following: the analytic properties of multi-point functions in the initial state, the fast-enough suppression of high velocity modes ensuring the "Separation" step, and the fulfillment of a linear-in-$T$ large-deviation principle for the current fluctuations, which has remained an implicit assumption in our work. As shown by the analysis of 4.1.2, the finiteness of the scaled second cumulant in GGEs is linked to the sufficiently fast decay of the fermionic occupation function and is guaranteed by two-times differentiable 2-pt functions at least in free-fermion theories. Whether the smoothness of certain multi-point functions implies a large-deviation principle more generally, and if yes, in what set of models, are interesting questions, hopefully attracting more attention in the near future.

Finally, as we have already stressed, the cluster property together with the principle of Asymptotic Commutativity, which we believe to be the consequence of the former, establishes the hydrodynamic identity (8), which, however, has limited predictive power. For the fully hydrodynamic hydrodynamic theory encapsulated in (9), the following two requirements are also necessitated.

- The condition of local relaxation to a steady-state $\rho_{\text{NESS}}(\lambda)$ from $\exp(\lambda/2 \int_0^\infty \mathrm{d}x\, q(x, 0))|\Psi\rangle$, is also important. Whereas local relaxation to maximal entropy steady-states generally occurs in isolated quantum systems, it must be mentioned that the characterisation of such states can be extremely complicated. However, only the knowledge of these states can provide predictive power to the hydrodynamic framework (9) of this work, at least in its present form.

  It is worth noting that in some cases this task can be made easier via an intermediate step, involving the relaxation after the quench of the left and right part of the systems to their associated steady-states, $\rho$ and $\rho^{(\lambda)}$, first. Then the asymptotic state is $\rho_{\text{NESS}} = \rho_{\text{bpp}}$, where $\rho_{\text{bpp}}$ is the NESS emerging after joining $\rho$ and $\rho^{(\lambda)}$. In other words, the system when quenched from $\exp(\lambda \int_0^\infty \mathrm{d}x\, q(x, 0))|\Psi\rangle$ first converges to these two maximal entropy states on its left and right sides, and the NESS in the middle region emerges from these two ensembles further away in space. Importantly, we have explicitly verified this intuition in free fermion systems for a special class of quenches in terms of GGEs, but the findings of [21] confirms the validity of this idea for the Rule 54 model (i.e., an interacting integrable system) as well.

  However, the characterisation of either $\rho$ or $\rho^{(\lambda)}$ can remain very complicated. In free or interacting integrable models, where the steady-states are generally GGEs, their determination is known for integrable initial states, such as the squeezed-coherent initial states in the free fermioin case. For such states, integrable or free dynamics also guarantees the local relaxation from $e^{\lambda/2Q}|\Psi\rangle$ to $\rho_{\text{GGE}}^{(\lambda)} \propto \rho_{\text{GGE}} e^{2\lambda Q}$ where $\rho_{\text{GGE}}$ is the GGE emerging from the unbiased initial state $|\Psi\rangle$. This is a non-trivial results originating from the specific structure of the initial state, which we demonstrated in this work by explicit microscopic computations and via the Quench Action method for generic integrable systems in [24].

If the system and/or the initial state is non-integrable, as already mentioned, the computation of the steady-state is generally difficult. There is, however, an interesting case, namely initial states for which $e^{\lambda Q/2}|\Psi\rangle$ relaxes to $\rho e^{\lambda Q}$, if $|\Psi\rangle$ relaxes to $\rho$. In this case, the stationary property of the steady-state and the non-equilibrium fluctuation relations imply that the hydrodynamic picture (9) predicts a $o(T)$ temporal bit for the non-equilibrium FCS, i.e., a weaker than linear behaviour (cf. [24]). Note that in this case the hydrodynamic predictions may not be valid since an $\mathcal{O}(T)$ behaviour in the FCS, was assumed to hold in our derivation. However, the naive predictions imply either the temporal invariance of the FCS or a sublinear growth in the cumulant generating function, which would be important to test by different means and in other relevant class of initial states.

- The last condition about temporal clustering is a non-trivial requirement. In fact, it has to be pointed out that checking it is highly non-trivial even in the case free fermion and integrable quenches, and it is an interesting question, whether this condition can be replaced by any other which might be easier to test. Another noteworthy comment is that (inhomogeneous) quenches generally give rise to long-range correlations and clarifying under what circumstances they may be strong enough to violate the temporal cluster property is non-trivial as well. This temporal cluster property is an important ingredient in the current formulation of our hydrodynamic picture to compute non-equilibrium FCS, as only in the absence of strong temporal correlations can this quantity be computed in terms of solely thermodynamic and hydrodynamic properties of the system via BFT.

  However, there are a few points to mention at this stage. An intriguing finding of the companion paper [24] of this work is that the temporal cluster property is in fact violated in the quenched Lieb-Liniger model, (that is, in an interacting integrable model). Nonetheless, the same hydrodynamic picture (i.e., the same formula (9)) specifying the particle number fluctuations still gives exact results up to the 5th scaled cumulant. In other words, long-range correlations and the violation of the temporal cluster property still allow for the usage of the fully hydrodynamic picture (9) in a constrained yet rather non-trivial and powerful way. The other comment we wish to make is that at least in integrable systems, a rather new theoretical framework, the Ballistic Macroscopic Fluctuation Theory (BMFT) [30, 31] could in principle take into account the effect of temporal correlations directly in quantities such as the dynamical free energy [24] (accounting for the time dependent fluctuation of the FCS). This method relies on the thermodynamic and hydrodynamic properties of the system as well as on the knowledge of initial correlations. However, we leave the challenging task of integrating BMFT into our current hydrodynamic picture for future work.

In our work an important role was played by non-relativistic free fermions for two reasons. On the one hand, we could rigorously show certain steps in the derivation of the hydrodynamics picture for the case of free fermions. Such steps are the "Separation" step with infinite LR velocity, or the "Asymptotic Commutativity" property for which free fermions offered a transparent analysis, despite the guiding principles underlying these properties are expected to be general. On the other hand, we also studied integrable or solvable quantum quenches in free fermion models to study the fulfillment of the required conditions in physically relevant situations, and to eventually cross-check the hydrodynamic predictions. For the fulfillment of the physical conditions, we performed explicit microscopic computations and obtained closed-formed expressions or carried out suitable series expansion wrt. the counting field. Since the FCS after such quenches has been computed in Refs. [20] and [21] by different means, independent cross-checks could be carried out easily to test the hydrodynamic picture.

Finally, we would like to stress that the range of applicability of the hydrodynamic framework (9) includes further interesting situations on the top of the ones we explicitly discussed. In particular, the extensivity of the initial fluctuations (i.e., that the scaled cumulant generating function in the

initial state scales linearly with $X$) can be relaxed, at least to cases when certain charges have no fluctuations in the initial state at all. An example can be the spin variable and states defined on the lattice with consecutive spin-up spin-down structure, such as the Néel state. In such case of zero initial fluctuations, the contour manipulations do not alter the initial state. Moreover, the NESS clearly remains a homogeneous maximal entropy state $\rho$ of the original homogeneous quench problem (with dynamical partition function given by $\lim_{T\to\infty} T^{-1} \sum_\pm \ln \mathrm{Tr} \exp(\pm\lambda \int_0^T \mathrm{d}t\, j(0,t))\rho)$. In fact, in this case, for interacting integrable lattice models and integrable initial states, the fact that the modified quench problem gives rise to a state that looks, at hydrodynamic scale, homogeneous, and consequently the absence of the "hydrodynamical waft effect" [24, 58], strongly suggests that this condition is actually satisfied. That is, in such cases, the hydrodynamic picture predicts that the dynamical fluctuations of the specific conserved charge are encoded by the current fluctuation in the steady-state after the original quench problem. The dynamical fluctuations can hence be expressed by the (integrable version of the) Flow Equations which can be recast in very similar fashion to (74) and (75) using the effective velocity and dressed quantities [21, 24] but without any inhomogeneity involved. An important consequence of our reasoning is that the results of [20], indeed generalise to generic integrable systems and such non-fluctuating initial states with pair-structure, which are necessarily integrable initial states [59].

Last but not least, besides the numerous aforementioned open problems, let us finish by mentioning some additional natural directions and improvements that our work offers. It would be particularly important to extend the hydrodynamic picture to generic $X/T$ ratios and obtain the full temporal behaviour of the FCS on the Euler-scale. Describing the time evolution of standard and symmetry-resolved Rényi entropies (associated with non-Noether conserved charges) in interacting cases is anticipated to be feasible within this approach and achieving this task would be of great importance as a possible derivation of the quasi-particle picture and its generalisations. Finally, elaborating on cases which violate the large deviation principle (such as the behaviour identified in connection with "squeezed ensemble" [40]) is of high interest as well.

# Acknowledgements

DXH is grateful to F. Hübner and C. von Keyserlingk for inspiring discussions. The work of BD and DXH has been supported by the Engineering and Physical Sciences Research Council (EPSRC) under grants number EP/W010194/1 and EP/Z534304/1. PR acknowledges support from Engineering and Physical Sciences Research Council (EPSRC) under the New Investigator Award scheme (Grant number EP/Y015363/1).

# A    Clustering of the free fermion integrable initial states

Whereas the strong enough clustering property i) is expected to hold quite generally, in particular, in initial states associated with ground states of gapped Hamiltonians, it is easy to show that the squeezed-coherent free fermion initial states satisfy i) which we now explicitly demonstrate below at least for the case of 2-pt functions.

First, let us consider densities of conserved charges. The density operators corresponding to the class of local and quasi-local operators are always fermion bilinears with a general structure

$$q(x) = \frac{1}{2\pi} \int \mathrm{d}k\,\mathrm{d}p\, q(k,p) a_k^\dagger a_p e^{i(k-p)x} . \tag{217}$$

Thanks to the Bogolyubov transformation (63) and (64) for the integrable quench, the operators in the connected 2-pt functions can be easily expressed via $\tilde{a}_k$ and $\tilde{a}_k^\dagger$ which annihilate the initial

state $|\Omega\rangle$, $\langle\Omega|$, respectively, and using Wick's theorem the correlation functions are easy to evaluate. This way we end up with

$$\langle\Omega|\psi^\dagger(x)\psi(y)|\Omega\rangle^c = \int \frac{dk}{2\pi} \frac{|K(k)|^2}{1+|K(k)|^2} e^{ik(x-y)}, \; \langle\Omega|\psi(x)\psi(y)|\Omega\rangle^c = \int \frac{dk}{2\pi} \frac{K(k)}{1+|K(k)|^2} e^{ik(x-y)},$$
(218)

with $\langle\Omega|\psi^\dagger(x)\psi^\dagger(y)|\Omega\rangle^c = (\langle\Omega|\psi(x)\psi(y)|\Omega\rangle^c)^*$ and hence with

$$\langle\Omega|q_i(x)q_j(y)|\Omega\rangle^c =$$
$$= \int \frac{dk}{2\pi}\frac{dp}{2\pi}\left\{\frac{|K(k)|^2}{1+|K(k)|^2}\frac{q_i(k,p)q_j(p,k)}{1+|K(p)|^2} + \frac{K^*(k)q_i(k,p)}{1+|K(k)|^2}\frac{K(p)q_j(-k,-p)}{1+|K(p)|^2}\right\} e^{i(k-p)(x-y)}.$$
(219)

It is easy to see a $\delta(x-y)$ contribution in (219) and exponential decaying terms both in (218) and (219).We start by clarifying that $K(k) = K_1(k) + iK_2(k)$ with two real functions, therefore $|K(pk)|^2$ is rather understood as $K_1^2(k)+K_2^2(k)$ defining a non-trivial complex function for complex arguments. Using the theorems of complex analysis, $|K(k)|^2 = K_1^2(k)+K_2^2(k)$ as a complex function cannot be a bounded function as it is assumed to be analytic and not a constant. Consequently, on the complex plain there exists $k^*$ values with $-1 = |K(k^*)|^2$ and according to the residue theorem (after subtracting the contribution giving rise to the Dirac-$\delta$ contribution) the decay of the correlation function is $e^{-2\text{Im}(k^*)|x-y|}$ with the smallest $\text{Im}(k^*)$ and with some possible power-law pre-factors.

Regarding $O$ as a more generic local operator of the form

$$O(x,0) = f(\psi(x), \partial_x\psi(x), ..., \psi^\dagger(x), \partial_x\psi^\dagger(x), ...),$$
(220)

where ... indicate higher order derivatives and $f$ is a generic function, we can use Wick's theorem, thanks to the properties of the initial state, and (218), which ensure that composite fermion operators also give rise to exponentially decaying correlations.

# B  Factorisation property 2 in the 3rd order wrt. $\lambda$

In this appendix we provide more details on demonstrating the validity of (137), that is,

$$\ln\frac{\langle\Psi|e^{\lambda J_0|_0^T + \lambda Q|_0^\infty(0)}|\Psi\rangle}{\langle\Psi|e^{\lambda Q|_0^\infty(0)}|\Psi\rangle} = \ln\frac{\langle\Psi|e^{\lambda J_0|_0^T}e^{\lambda Q|_0^\infty(0)}|\Psi\rangle}{\langle\Psi|e^{\lambda Q|_0^\infty(0)}|\Psi\rangle} + o(T),$$
(221)

in free fermion systems, but generic, exponentially clustering and even parity states. In particular, continuing with the series expansion started in Section 4.2 here we focus on the 3rd order, which means we have to show that for the corresponding terms in (143), namely

$$\lambda^3: \quad \langle\Psi|\tilde{J}[\tilde{J},\tilde{Q}]|\Psi\rangle = \langle\Psi|\tilde{Q}[\tilde{J},\tilde{Q}]|\Psi\rangle - \langle\Psi|\tilde{Q}|\Psi\rangle\langle\Psi|[\tilde{J},\tilde{Q}]|\Psi\rangle =$$
$$= \langle\Psi|\left[\tilde{Q},[\tilde{J},\tilde{Q}]\right]|\Psi\rangle = \langle\Psi|\left[\tilde{J},[\tilde{J},\tilde{Q}]\right]|\Psi\rangle = o(T)$$
(222)

holds in the aforementioned clustering states. For simplicity, we shall assume that fermion 4-pt functions are free of divergences and use the cluster property (153) in what follows.

Staring with the evaluation of the two nested commutators, and first focusing on $\left[\tilde{Q},[\tilde{J},\tilde{Q}]\right]$,

we can write that

$$\left[\tilde{Q},[\tilde{J},\tilde{Q}]\right]=\lim_{\epsilon\to0}\int_0^\infty dx\int_{-\infty}^0 dy\int_0^\infty dz\left\{G(y,z,T)_\epsilon[\psi^\dagger(x)\psi(x),\psi^\dagger(y)\psi(z)]\right.$$

$$\left.-G(z,y,T)_\epsilon[\psi^\dagger(x)\psi(x),\psi^\dagger(z)\psi(y)]\right\}$$

$$=\lim_{\epsilon\to0}\int_0^\infty dx\int_{-\infty}^0 dy\int_0^\infty dz\left\{G(y,z,T)_\epsilon\left(\delta(x-y)-\delta(z-x)\right)\psi^\dagger(y)\psi(z)\right. \quad(223)$$

$$\left.-G(z,y,T)_\epsilon\left(\delta(z-x)-\delta(x-y)\right)\psi^\dagger(z)\psi(y)\right\}$$

$$=-\lim_{\epsilon\to0}\int_0^\infty dx\int_{-\infty}^0 dy\left\{G(y,x,T)_\epsilon\psi^\dagger(y)\psi(x)+G(x,y,T)_\epsilon\psi^\dagger(x)\psi(y)\right\}$$

Comparing the above formula with the one obtained for the single commutator in (147) we can immediately see that

$$\langle\Psi|\left[\tilde{Q},[\tilde{J},\tilde{Q}]\right]|\Psi\rangle=0, \tag{224}$$

since

$$\langle\Psi|\left[\tilde{Q},[\tilde{J},\tilde{Q}]\right]|\Psi\rangle=$$

$$-\lim_{\epsilon\to0}\int_0^\infty dx\int_{-\infty}^0 dy\,G(y,x,T)_\epsilon C_{+-}(y-x)-\lim_{\epsilon\to0}\int_0^\infty dx\int_{-\infty}^0 dy\,G(x,y,T)_\epsilon C_{+-}(x-y)$$

$$-\lim_{\epsilon\to0}\int_0^\infty dx\int_{-\infty}^0 dy\,G(y,x,T)_\epsilon C_{+-}(y-x)-\lim_{\epsilon\to0}\int_{-\infty}^0 dx\int_0^\infty dy\,G(-x,-y,T)_\epsilon C_{+-}(y-x)$$

$$-\lim_{\epsilon\to0}\int_0^\infty dx\int_{-\infty}^0 dy\,G(y,x,T)_\epsilon C_{+-}(y-x)+\lim_{\epsilon\to0}\int_{-\infty}^0 dy\int_0^\infty dx\,G(y,x,T)_\epsilon C_{+-}(x-y)$$

$$=0$$

$$(225)$$

as $G(-x,-y,T)_\epsilon=-G(x,y,T)_\epsilon$ and $C_{+-}$ is an even function.

Turning to the other nested commutator $\left[\tilde{J},[\tilde{J},\tilde{Q}]\right]$, we can write that

$$\left[\tilde{J},[\tilde{J},\tilde{Q}]\right]=\lim_{\epsilon\to0}\int dy\int dz\int_\infty^0 dy'\int_0^\infty dz'\,G(y,z,T)_\epsilon\left\{G(y',z',T)_\epsilon[\psi^\dagger(y)\psi(z),\psi^\dagger(y')\psi(z')]\right.$$

$$\left.-G(z',y',T)_\epsilon[\psi^\dagger(y)\psi(z),\psi^\dagger(z')\psi(y')]\right\}. \tag{226}$$

Working out the commutators using (146) and applying suitable changes of integration variables, we end up with the following expression,

$$\left[\tilde{J},[\tilde{J},\tilde{Q}]\right]=\lim_{\epsilon\to0}\int dx\int_{-\infty}^0 dy\int_0^\infty dz\left\{G(x,y,T)_\epsilon G(y,z,T)_\epsilon\psi^\dagger(x)\psi(z)-G(z,x,T)_\epsilon G(y,z,T)_\epsilon\times\right.$$

$$\left.\times\psi^\dagger(y)\psi(x)-G(x,z,T)_\epsilon G(z,y,T)_\epsilon\psi^\dagger(x)\psi(y)+G(y,x,T)_\epsilon G(z,y,T)_\epsilon\psi^\dagger(z)\psi(x)\right\}. \tag{227}$$

To proceed with taking the expectation value of the above expression we only need to consider the first and the third terms in (227) (as the second and the fourth can be treated in complete analogy)

and which we denote as $\mathcal{A}$. This quantity can be written as

$$
\begin{aligned}
\mathcal{A} &= \int dx \int_{-\infty}^{0} dy \int_{0}^{\infty} dz \left\{ G(z+x,y,T)_{\epsilon} G(y,z,T)_{\epsilon} - G(y+x,z,T)_{\epsilon} G(z,y,T)_{\epsilon} \right\} C_{+-}(x) \\
&= \int dx \int_{-\infty}^{0} dy \int_{0}^{\infty} dz \, I(y,z,x,T,\epsilon),
\end{aligned}
\tag{228}
$$

due to the range of integration and to the fact that the function $I$ integrand satisfies (as $C_{+-}(x)$ is an even function)

$$
I(y,z,x,T,\epsilon) = -I(z,y,x,T,\epsilon) = I(-y,-z,-x,T,\epsilon),
\tag{229}
$$

from which it follows that

$$
\begin{aligned}
\int dx \int_{-\infty}^{0} dy \int_{0}^{\infty} dz \, I(y,z,x,T,\epsilon) &= -\int dx \int_{-\infty}^{0} dy \int_{0}^{\infty} dz \, I(z,y,x,T,\epsilon) \\
&= -\int dx \int_{-\infty}^{0} dy \int_{0}^{\infty} dz \, I(-z,-y,-x,T,\epsilon) = -\int dx \int_{-\infty}^{0} dz \int_{0}^{\infty} dy \, I(z,y,x,T,\epsilon) = \\
&= -\int dx \int_{-\infty}^{0} dy \int_{0}^{\infty} dz \, I(y,z,x,T,\epsilon),
\end{aligned}
\tag{230}
$$

and hence the integral of $I$ is zero. That is, we have that

$$
\langle \Psi | \left[ \tilde{J}, [\tilde{J}, \tilde{Q}] \right] | \Psi \rangle = 0,
\tag{231}
$$

if the state $|\Psi\rangle$ is translation invariant and parity even.

Our last task is to elaborate on the mixed expressions consisting of a single commutator and an operator, $\langle \Psi | \tilde{Q} [\tilde{J}, \tilde{Q}] | \Psi \rangle$ and $\langle \Psi | \tilde{Q} [\tilde{J}, \tilde{Q}] | \Psi \rangle$. For the former term we can write

$$
\begin{aligned}
\langle \Psi | \tilde{Q} [\tilde{J}, \tilde{Q}] | \Psi \rangle &= \lim_{\epsilon \to 0} \int_{0}^{\infty} dx \int_{-\infty}^{0} dy \int_{0}^{\infty} dz \left\{ G(y,z,T)_{\epsilon} \langle \Psi | \psi^{\dagger}(x)\psi(x)\psi^{\dagger}(y)\psi(z) | \Psi \rangle \right. \\
&\qquad\qquad\qquad\qquad\qquad \left. -G(z,y,T)_{\epsilon} \langle \Psi | \psi^{\dagger}(x)\psi(x)\psi^{\dagger}(z)\psi(y) | \Psi \rangle \right\} \\
&= \lim_{\epsilon \to 0} \int_{0}^{\infty} dx \int_{-\infty}^{0} dy \int_{0}^{\infty} dz \left\{ G(y,z,T)_{\epsilon} C_{+-+-}(x,x,y,z) - G(z,y,T)_{\epsilon} C_{+-+-}(x,x,z,y) \right\},
\end{aligned}
\tag{232}
$$

where $C_{+-+-}$ denotes the corresponding 4-pt correlation function of fermion creation and fermion annihilation operators. Using the fact that $\langle \Psi | \psi(x) | \Psi \rangle = 0$, this function can be rewritten in terms of connected correlation functions as

$$
\begin{aligned}
C_{+-+-}(x,x,y,z) &= C^{c}_{+-+-}(x,x,y,z) + C_{+-}(0) C_{+-}(y-z) \\
&\quad - C_{++}(x-z) C_{--}(x-y) + C_{+-}(x-z) C_{-+}(x-y),
\end{aligned}
\tag{233}
$$

where we exploited translational invariance of the state and $C_{+-}(0) = \langle q(0) \rangle$. We can immediately note that the expression $C_{+-}(0) C_{+-}(y-z)$ type expression plugged in (232) cancels with $\langle \Psi | \tilde{Q} | \Psi \rangle \langle \Psi | [\tilde{J}, \tilde{Q}] | \Psi \rangle$. We rewrite (232) via the remaining expressions including the connected

correlation functions as

$$\langle\Psi|\tilde{Q}[\tilde{J},\tilde{Q}]|\Psi\rangle = C_4^Q + C_{2,a}^Q + C_{2,b}^Q$$

$$C_4^Q = \lim_{\epsilon\to 0}\int_0^\infty dx\int_{-\infty}^0 dy\int_0^\infty dz\,\{G(y,z,T)_\epsilon C_{+-+-}^c(x,x,y,z) - G(z,y,T)_\epsilon C_{+-+-}^c(x,x,z,y)\}$$

$$C_{2,a}^Q = -\lim_{\epsilon\to 0}\int_0^\infty dx\int_{-\infty}^0 dy\int_0^\infty dz\,\{G(y,z,T)_\epsilon C_{++}(x-z)C_{++}^*(x-y)$$

$$-G(z,y,T)_\epsilon C_{++}(x-y)C_{++}^*(x-z)\}$$

$$C_{2,b}^Q = -\lim_{\epsilon\to 0}\int_0^\infty dx\int_{-\infty}^0 dy\int_0^\infty dz\,\{G(y,z,T)_\epsilon C_{+-}(x-z)C_{+-}(x-y)$$

$$-G(z,y,T)_\epsilon C_{+-}(x-y)C_{+-}(x-z)\} - \lim_{\epsilon\to 0}\int_{-\infty}^0 dy\int_0^\infty dz\,G(z,y,T)_\epsilon C_{+-}(z-y),$$

$$(234)$$

where we used that all the 2-pt functions are translation invariant, and in addition, $C_{++}(x) = C_{--}^*(x)$, and $C_{-+}(x) = \delta(x) - C_{+-}(x)$. We now analyse first the terms $C_2^Q$ associated with products of 2-pt functions before turning to the connected 4-pt functions.

Starting with $C_{2,a}^Q$ we can repeat the steps of (150) and expand $G_1(y,z,T)$ ($\propto (y+z)/T + ...$)

$$C_{2,a}^Q = -\int_0^\infty dx\int_{-\infty}^0 dy\int_0^\infty dz\,\{\left((y+z)T^{-1} + \text{higher powers}\times T^{-n}\right)C_{++}(x-z)C_{++}^*(x-y)$$

$$-\left((z+y)T^{-1} + \text{higher powers}\times T^{-n}\right)C_{++}(x-y)C_{++}^*(x-z)\}$$

$$+\frac{i}{2\pi}\int_0^\infty dx\int_{-\infty}^0 dy\int_0^\infty dz\,\frac{C_{++}(x-z)C_{++}^*(x-y) + C_{++}(x-y)C_{++}^*(x-z)}{y-z}$$

$$= \mathcal{O}(1) + o(T^{-1}) = o(T),$$

$$(235)$$

where the integrals are finite, and higher powers refer to higher powers of the spatial coordinates $y$ and $z$, and $T^{-n}$ stands for increasing negative powers of $T$ as the order of the expansion grows. Analogously for we have that

$$C_{2,b}^Q = -\int_0^\infty dx\int_{-\infty}^0 dy\int_0^\infty dz\,\{\left((y+z)T^{-1} + \text{higher powers}\times T^{-n}\right)C_{+-}(x-z)C_{+-}(x-y)$$

$$-\left((z+y)T^{-1} + \text{higher powers}\times T^{-n}\right)C_{+-}(x-y)C_{+-}(x-z)\}$$

$$-\int_{-\infty}^0 dy\int_0^\infty dz\,\left((z+y)T^{-1} + \text{higher powers}\times T^{-n}\right)C_{+-}(z-y)$$

$$-\frac{i}{2\pi}\int_0^\infty dx\int_{-\infty}^0 dy\int_0^\infty dz\,\frac{C_{+-}(x-z)C_{+-}(x-y) + C_{+-}(x-y)C_{+-}(x-z)}{y-z}$$

$$-\frac{i}{2\pi}\int_{-\infty}^0 dy\int_0^\infty dz\,\frac{C_{+-}(z-y)}{z-y}$$

$$= \mathcal{O}(1) + \mathcal{O}(T^{-1}) = o(T).$$

$$(236)$$

For the bit $C_4^Q$ including the connected 4-pt function $C_{+-+-}^c$ we have

$$C_4^Q = \lim_{\epsilon \to 0} \int_0^\infty dx \int_{-\infty}^0 dy \int_0^\infty dz \left\{ G(y,z,T)_\epsilon C_{+-+-}^c(x,x,y,z) - G(z,y,T)_\epsilon C_{+-+-}^c(x,x,z,y) \right\},$$

(237)

and recognise, that due to the integration range either the $y$ or the $z$ argument is always far away from the other coordinates. Accordingly, we can proceed the usual way, namely by expanding the $T$-dependent oscillatory functions around the origin wrt. their spatial variables. By (144), we we have that

$$\begin{aligned}
C_4^Q &= \int_0^\infty dx \int_{-\infty}^0 dy \int_0^\infty dz \left\{ \left( (y+z)T^{-1} + ... \right) C_{+-+-}^c(x,x,y,z) \right. \\
&\qquad\qquad\qquad\qquad \left. - \left( (z+y)T^{-1} + ... \right) C_{+-+-}^c(x,x,z,y) \right\} \\
&\quad - \frac{i}{2\pi} \int_0^\infty dx \int_{-\infty}^0 dy \int_0^\infty dz \frac{C_{+-+-}^c(x,x,y,z) + C_{+-+-}^c(x,x,z,y)}{y-z} \\
&= \mathcal{O}(T^{-1}) + \mathcal{O}(1) = o(T),
\end{aligned}$$

(238)

where we used that for small arguments $G_1(y,z,T) \approx (y+z)/T$ and that the integrals are finite due to the exponential cluster property (153) and to the range of integration.

Finally, let us now elaborate on the last term $\langle \Psi | \tilde{J}[\tilde{J}, \tilde{Q}] | \Psi \rangle$, (as it is immediate to see that $C_{2,c}^J = \langle \Psi | \tilde{J} | \Psi \rangle \langle \Psi | [\tilde{J}, \tilde{Q}] | \Psi \rangle = 0$), which we write as

$$\begin{aligned}
\langle \Psi | \tilde{J}[\tilde{J}, \tilde{Q}] | \Psi \rangle &= \lim_{\epsilon \to 0} \int dy \int dz \int_{-\infty}^0 dy' \int_0^\infty dz' G(y,z,T)_\epsilon \left\{ G(y',z',T)_\epsilon \times \right. \\
&\quad \langle \Psi | \psi^\dagger(y)\psi(z)\psi^\dagger(y')\psi(z') | \Psi \rangle - G(z',y',T)_\epsilon \langle \Psi | \psi^\dagger(y)\psi(z)\psi^\dagger(z')\psi(y') | \Psi \rangle \Big\} \\
&= \lim_{\epsilon \to 0} \int dy \int dz \int_{-\infty}^0 dy' \int_0^\infty dz' G(y,z,T)_\epsilon \left\{ G(y',z',T)_\epsilon C_{+-+-}(y,z,y',z') \right. \\
&\qquad\qquad\qquad\qquad\qquad \left. - G(z',y',T)_\epsilon C_{+-+-}(y,z,z',y') \right\}.
\end{aligned}$$

(239)

Assuming that $\langle \Psi | \psi^\dagger(y) | \Psi \rangle = 0$, we can easily express the 4-pt correlation function in terms of the connected ones:

$$\begin{aligned}
\langle \Psi | \tilde{J}[\tilde{J}, \tilde{Q}] | \Psi \rangle &= C_4^J + C_{2,a}^J + C_{2,b}^J + C_{2,c}^J \\
C_4^J &= \lim_{\epsilon \to 0} \int dy \int dz \int_{-\infty}^0 dy' \int_0^\infty dz' G(y,z,T)_\epsilon \left[ G(y',z',T)_\epsilon C_{+-+-}^c(y,z,y',z') \right. \\
&\qquad\qquad\qquad\qquad\qquad \left. - G(z',y',T)_\epsilon C_{+-+-}^c(y,z,z',y') \right] \\
C_{2,a}^J &= -\lim_{\epsilon \to 0} \int dy \int dz \int_{-\infty}^0 dy' \int_0^\infty dz' G(y,z,T)_\epsilon \left[ G(y',z',T)_\epsilon C_{++}^c(y,y')C_{--}^c(z,z') \right. \\
&\qquad\qquad\qquad\qquad\qquad \left. - G(z',y',T)_\epsilon C_{++}^c(y,z')C_{--}^c(z,y') \right] \\
C_{2,b}^J &= \lim_{\epsilon \to 0} \int dy \int dz \int_{-\infty}^0 dy' \int_0^\infty dz' G(y,z,T)_\epsilon \left[ G(y',z',T)_\epsilon C_{+-}^c(y,z')C_{-+}^c(z,y') \right. \\
&\qquad\qquad\qquad\qquad\qquad \left. - G(z',y',T)_\epsilon C_{+-}^c(y,y')C_{-+}^c(z,z') \right] \\
C_{2,c}^J &= \lim_{\epsilon \to 0} \int dy \int dz \int_{-\infty}^0 dy' \int_0^\infty dz' G(y,z,T)_\epsilon \left[ G(y',z',T)_\epsilon C_{+-}^c(y,z)C_{+-}^c(y',z') \right. \\
&\qquad\qquad\qquad\qquad\qquad \left. - G(z',y',T)_\epsilon C_{+-}^c(y,z)C_{+-}^c(z',y') \right].
\end{aligned}$$

(240)

For $\mathcal{C}_{2,a}^J$, after suitable change of integration variables, we can write that

$$
\mathcal{C}_{2,a}^J = -\int \mathrm{d}y \int \mathrm{d}z \int_{-\infty}^{0} \mathrm{d}y' \int_{0}^{\infty} \mathrm{d}z' G(y'+y, z'+z, T)_\epsilon G(y', z', T)_\epsilon C_{++}(y) C_{++}^*(z)
$$
$$
- G(z'+y, y'+z, T)_\epsilon G(z', y', T)_\epsilon C_{++}(y) C_{++}^*(z) \tag{241}
$$
$$
= \int \mathrm{d}y \int \mathrm{d}z \int_{-\infty}^{0} \mathrm{d}y' \int_{0}^{\infty} \mathrm{d}z' \, I'(y', z', y, z, T, \epsilon) = 0,
$$

due to the specific integration range and to the fact that

$$
I'(y', z', y, z, T, \epsilon) = -I'(z', y', y, z, T, \epsilon) = I'(-y', -z', -y, -z, T, \epsilon), \tag{242}
$$

which is a consequence of the properties of the $G$ functions as well as the fact that the fermionic 2-pt functions are even ones in even parity states (cf. (230)). We can proceed in an analogous way for $\mathcal{C}_{2,b}^J$ and we make use of the fact that $C_{-+}(x) = \delta(x) - C_{+-}(x)$. This way, again performing changes of integration variables we have that

$$
\mathcal{C}_{2,b}^J = -\int \mathrm{d}y \int \mathrm{d}z \int_{-\infty}^{0} \mathrm{d}y' \int_{0}^{\infty} \mathrm{d}z' G(z'+y, y'+z, T)_\epsilon G(y', z', T)_\epsilon C_{+-}(y) C_{+-}(z)
$$
$$
- G(y'+y, z'+z, T)_\epsilon G(z', y', T)_\epsilon C_{+-}(y) C_{+-}(z)
$$
$$
+ \int \mathrm{d}y \int_{-\infty}^{0} \mathrm{d}y' \int_{0}^{\infty} \mathrm{d}z' \, G(y, y', T)_\epsilon G(y', z', T)_\epsilon C_{+-}(y - z') \tag{243}
$$
$$
- G(y, z', T)_\epsilon G(z', y', T)_\epsilon C_{+-}(y - y')
$$
$$
= \int \mathrm{d}y \int \mathrm{d}z \int_{-\infty}^{0} \mathrm{d}y' \int_{0}^{\infty} \mathrm{d}z' \, I''(y', z', y, z, T, \epsilon) + \mathscr{A} = 0,
$$

where $\mathscr{A} = 0$ is defined in (228) and where the 4-fold integral is zero as the integrand has the properties

$$
I''(y', z', y, z, T, \epsilon) = -I''(z', y', y, z, T, \epsilon) = I''(-y', -z', -y, -z, T, \epsilon). \tag{244}
$$

That is, we have that

$$
\mathcal{C}_{2,a}^J = 0, \, \mathcal{C}_{2,b}^J = 0, \, \mathcal{C}_{2,c}^J = 0. \tag{245}
$$

Now the only remaining bit the connected 4-pt function

$$
\mathcal{C}_4^J = \lim_{\epsilon \to 0} \int \mathrm{d}y \int \mathrm{d}z \int_{-\infty}^{0} \mathrm{d}y' \int_{0}^{\infty} \mathrm{d}z' G(y, z, T)_\epsilon \times
$$
$$
\times \left( G(y', z', T)_\epsilon C_{+-+-}^c(y, z, y', z') - G(z', y', T)_\epsilon C_{+-+-}^c(y, z, z', y') \right) \tag{246}
$$

Here we can proceed as previously and note that due to the restricted integration range for the variables $y'$ and $z'$ the connected correlation function is not exponentially small only when all the variables are close to the origin. Assuming the lack of divergencies, we again expand the $T$-dependent oscillatory functions around the origin. Because $G_\epsilon$ can be split to a $T$ and an $\epsilon$-dependent term, and we have two products of $G_\epsilon$ we shall investigate the corresponding contributions $\mathcal{C}_4^J(T, T)$, $\mathcal{C}_4^J(T, \epsilon)$, $\mathcal{C}_4^J(\epsilon, T)$, and $\mathcal{C}_4^J(\epsilon, \epsilon)$ separately. In particular, approximating $G$ for small arguments, we can write

$$
\mathcal{C}_4^J(T, T) = \int \mathrm{d}y \int \mathrm{d}z \int_{-\infty}^{0} \mathrm{d}y' \int_{0}^{\infty} \mathrm{d}z' \left((y+z)T^{-1} + ...\right) \left\{ \left((y'+z')T^{-1} + ...\right) C_{+-+-}^c(y, z, y', z') \right.
$$
$$
\left. - \left((y'+z')T^{-1} + ...\right) C_{+-+-}^c(y, z, z', y') \right\}
$$
$$
= \mathcal{O}(T^{-2}) = o(T), \tag{247}
$$

where we assumed the 4-pt function is regular and the finiteness of the integral follows from (153). We can now repeat a very similar analysis for the cross terms coming from $G_\epsilon G_\epsilon$ and first write, after expanding $G_1(y, z, T)$,

$$
\begin{aligned}
\mathcal{C}_4^J(T, \epsilon) &= \lim_{\epsilon \to 0} \int \mathrm{d}y \int \mathrm{d}z \int_{-\infty}^0 \mathrm{d}y' \int_0^\infty \mathrm{d}z' \left((y+z)T^{-1} + \ldots\right) \times \\
&\qquad\qquad \times \left(G_1(y', z', \epsilon) C_{+-+-}^c(y, z, y', z') - G_1(z', y', \epsilon) C_{+-+-}^c(y, z, z', y')\right) \\
&= \lim_{\epsilon \to 0} \int \mathrm{d}y \int \mathrm{d}z \int_{-\infty}^0 \mathrm{d}y' \int_0^\infty \mathrm{d}z' \left((y+z)T^{-1} + \ldots\right) \times \\
&\qquad\qquad\qquad\qquad \times \left(\mathrm{i}\frac{1}{2\pi} \frac{C_{+-+-}^c(y, z, y', z') + C_{+-+-}^c(y, z, z', y')}{y' - z'}\right) \\
&= \mathcal{O}(T^{-1}) = o(T),
\end{aligned}
$$

(248)

since the Dirac-$\delta$ terms from (145) do not contribute due to the range of integration and similarly, the principal value operation can be omitted. For the other cross terms coming from $G_\epsilon G_\epsilon$ we write

$$
\begin{aligned}
\mathcal{C}_4^J(\epsilon, T) &= \lim_{\epsilon \to 0} \int \mathrm{d}y \int \mathrm{d}z \int_{-\infty}^0 \mathrm{d}y' \int_0^\infty \mathrm{d}z' G_1(y, z, \epsilon) \times \\
&\quad \times \left\{\left((y'+z')T^{-1} + \ldots\right) C_{+-+-}^c(y, z, y', z') - \left((y'+z')T^{-1} + \ldots\right) C_{+-+-}^c(y, z, z', y')\right\} \\
&= \int \mathrm{d}y \int \mathrm{d}z \int_{-\infty}^0 \mathrm{d}y' \int_0^\infty \mathrm{d}z' \left[-\frac{1}{2}\delta(y-z)\,\mathrm{sgn}(y) + \mathrm{i}\frac{1}{2\pi}\mathrm{P}\frac{1}{y-z}\right] \times \\
&\quad \times \left\{\left((y'+z')T^{-1} + \ldots\right) C_{+-+-}^c(y, z, y', z') - \left((y'+z')T^{-1} + \ldots\right) C_{+-+-}^c(y, z, z', y')\right\} \\
&= \mathcal{O}(T^{-1}) = o(T),
\end{aligned}
$$

(249)

where the integrals are finite due to (153). Finally, we invoke that

$$
\begin{aligned}
\mathcal{C}_4^J(\epsilon, \epsilon) &= \lim_{\epsilon \to 0} \int \mathrm{d}y \int \mathrm{d}z \int_{-\infty}^0 \mathrm{d}y' \int_0^\infty \mathrm{d}z' G_1(y, z, \epsilon) \times \\
&\qquad\qquad \times \left(G_1(y', z', \epsilon) C_{+-+-}^c(y, z, y', z') - G_1(z', y', \epsilon) C_{+-+-}^c(y, z, z', y')\right) \\
&= \int \mathrm{d}y \int \mathrm{d}z \int_{-\infty}^0 \mathrm{d}y' \int_0^\infty \mathrm{d}z' \left[-\frac{1}{2}\delta(y-z)\,\mathrm{sgn}(y) + \mathrm{i}\frac{1}{2\pi}\mathrm{P}\frac{1}{y-z}\right] \times \\
&\qquad\qquad \times \left(\mathrm{i}\frac{1}{2\pi} \frac{C_{+-+-}^c(y, z, y', z') + C_{+-+-}^c(y, z, z', y')}{y' - z'}\right)
\end{aligned}
$$

(250)

is finite as well.

That is, we have argued that also $\langle \Psi | \tilde{J} [\tilde{J}, \tilde{Q}] | \Psi \rangle = o(T)$, and in summary, that at the 3rd order each term

$$
\langle \Psi | \tilde{J} [\tilde{J}, \tilde{Q}] | \Psi \rangle = \mathcal{O}(1) + \mathcal{O}(T^{-1}), \quad \langle \Psi | \tilde{Q} [\tilde{J}, \tilde{Q}] | \Psi \rangle - \langle \Psi | \tilde{Q} | \Psi \rangle \langle \Psi | [\tilde{J}, \tilde{Q}] | \Psi \rangle = \mathcal{O}(1) + \mathcal{O}(T^{-1}),
$$
$$
\langle \Psi | [\tilde{Q}, [\tilde{J}, \tilde{Q}]] | \Psi \rangle = 0, \quad \langle \Psi | [\tilde{J}, [\tilde{J}, \tilde{Q}]] | \Psi \rangle = 0.
$$

(251)

## C  Free fermion GGEs

In this appendix we characterise certain GGEs for the free fermion problem. In particular, we first consider the GGEs corresponding to the integrable quenches, characterised by the state $|\Omega\rangle$ via a $K$-function. Then we show the emergence of the modified GGE $\rho_{\text{GGE}}e^{2\lambda Q}$ after the modified, homogeneous quench $e^{\lambda/2Q}|\Omega\rangle$. Finally we compute the certain 1- and 2-pt functions in the NESS emerging after the bipartite quench with initial left and right density matrices $\rho_{\text{GGE}}$ and $\rho_{\text{GGE}}^{2\lambda}$ where $\rho_{\text{GGE}}$ is the GGE after the homogeneous integrable quench defined by a $K$-function.

### C.1  The GGE after integrable quenches

It is well-known [46,60] that GGEs in a free fermion system can be written as

$$\rho_{\text{GGE}} = \frac{1}{Z} \exp\left(-\int dp\, \beta(p) a_p^\dagger a_p\right). \tag{252}$$

The normalisation can easily be expressed by using that $\text{Tr}(a_p^\dagger a_p)^n = 1$ and $\text{Tr}\, 1 = 2$ and assuming discrete fermionic oscillator modes monetarily. In particular,

$$\text{Tr}\left[\exp\left(-\beta(p') a_{p'}^\dagger a_{p'}\right)\right] = 1 + e^{-\beta(p')} \tag{253}$$

from which, in very large volume

$$\ln \text{Tr}\left[\exp\left(-\int dp'\beta(p') a_{p'}^\dagger a_{p'}\right)\right] = L \int \frac{dp'}{2\pi} \ln\left(1 + e^{-\beta(p')}\right) \tag{254}$$

and hence

$$Z = \exp\left[L \int \frac{dp'}{2\pi} \ln\left(1 + e^{\beta(p')}\right)\right]. \tag{255}$$

In the GGE emerging after and integrable quench, $\beta(p)$ can be fixed by prescribing that

$$L \int \frac{dp}{2\pi} \frac{|K(p)|^2}{1 + |K(p)|^2} = \frac{1}{Z}\text{Tr}\left[\int dp\, a_p^\dagger a_p \exp\left(-\int dp'\beta(p') a_{p'}^\dagger a_{p'}\right)\right] \tag{256}$$

from which

$$\begin{aligned}
\frac{L}{2\pi} \frac{|K(p)|^2}{1 + |K(p)|^2} &= \frac{1}{Z}\text{Tr}\left[a_p^\dagger a_p \exp\left(-\int dp'\beta(p') a_{p'}^\dagger a_{p'}\right)\right] \\
&= -\frac{1}{Z}\text{Tr}\left[\frac{\delta}{\delta\beta(p)} \exp\left(-\int dp'\beta(p') a_{p'}^\dagger a_{p'}\right)\right] \\
&= -\frac{1}{Z}\frac{\delta Z}{\delta\beta(p)} = -\frac{\delta}{\delta\beta(p)}\ln Z\,,
\end{aligned} \tag{257}$$

and hence $e^{-\beta(p)} = |K(p)|^2$ or $\beta(p) = -\ln(|K(p)|^2)$. Whereas the above explanation is completely rigorous, we whall demonstrate it in the next subsection via microscopic computations as well. Therefore the GGE after the quench reads as

$$\rho_{\text{GGE}} = \frac{1}{Z} \exp\left(\int dp\, \ln\left(|K(p)|^2\right) a_p^\dagger a_p\right) \tag{258}$$

with

$$Z = \exp\left[L \int \frac{dp}{2\pi} \ln\left(1 + |K(p)|^2\right)\right]. \tag{259}$$

Now we can continue by computing

$$\text{Tr}[O(t,x)\rho_{\text{GGE}}] =: \langle O \rangle_{\rho_{\text{GGE}}}, \tag{260}$$

where $O$ is a generic local operator which can be cast as a bilinear of the creation and annihilation operators. In any free fermion GGE written in terms of number operators we have the following relations

$$\langle a_p^\dagger a_{p'}^\dagger \rangle_{\rho_{\text{GGE}}} = 0, \qquad \langle a_p a_{p'} \rangle_{\rho_{\text{GGE}}} = 0, \qquad \langle a_p^\dagger a_{p'} \rangle_{\rho_{\text{GGE}}} = \delta(p-p')\frac{|K(p)|^2}{1+|K(p)|^2}, \tag{261}$$

together with the applicability of Wick's theorem. Recalling the form of the operator $O(t,x)$ in infinite volume

$$\begin{aligned}
O(t,x) = \int &\frac{\mathrm{d}p}{\sqrt{2\pi}}\frac{\mathrm{d}p'}{\sqrt{2\pi}} h_O(p,p') a_p^\dagger a_{p'} e^{i(p-p')x} e^{-it(E_p-E_{p'})} \\
&+ \tilde{h}_O^*(p,p') a_p^\dagger a_{p'}^\dagger e^{-i(p+p')x} e^{it(E_p+E_{p'})} + \tilde{h}_O(p,p') a_p a_{p'} e^{i(p+p')x} e^{-it(E_p+E_{p'})},
\end{aligned} \tag{262}$$

it is clear that

$$\text{Tr}\left[O^{++}(t,x)\rho_{\text{GGE}}^{2\lambda_i}\right] = \left[O^{--}(t,x)\rho_{\text{GGE}}^{2\lambda_i}\right] = 0, \tag{263}$$

and from $O^{+-}$ only the diagonal parts have non-vanishing expectation values. That is,

$$\text{Tr}[O(t,x)\rho_{\text{GGE}}] = \int \frac{\mathrm{d}p}{2\pi} h_O(p)\frac{|K(p)|^2}{1+|K(p)|^2}. \tag{264}$$

## C.2  Microscopic demonstration of the emergence of the biased GGE after integrable quenches

Below we show that

$$\lim_{t\to\infty} \frac{\langle\Omega|e^{\lambda/2Q}O(0,t)e^{\lambda/2Q}|\Omega\rangle}{\langle\Omega|e^{\lambda Q}|\Omega\rangle} = \lim_{t\to\infty} \frac{\langle\Omega|O(0,t)e^{\lambda Q}|\Omega\rangle}{\langle\Omega|e^{\lambda Q}|\Omega\rangle} = \frac{1}{Z^{2\lambda}}\text{Tr}\left[O(0,0)e^{2\lambda Q}\rho_{\text{GGE}}\right] \tag{265}$$

for generic local operators $O$ after integrable quenches. While below we do that via microscopic computations in the free fermion model, we mention that in [24] the authors provided a general derivation valid for interacting integrable models using the Quench Action method. The GGE density matrix is defined by the integrable quench $|\Omega\rangle$ whose explicit form is well-known but for a self-contained presentation, we shall recall this result and its derivation at the end of this section.

We first consider the case of the biased *homogeneous* initial state

$$|\bar{\Omega}\rangle_\lambda = e^{\lambda/2Q}|\Omega\rangle, \qquad |\bar{\Omega}\rangle_{2\lambda} = e^{\lambda Q}|\Omega\rangle, \tag{266}$$

and the non-symmetric expectation value

$$\frac{\langle\Omega|O(t,x)e^{\lambda Q}|\Omega\rangle}{\langle\Omega|e^{\lambda Q}|\Omega\rangle} = \frac{\langle\Omega|O(t,x)|\bar{\Omega}\rangle_{2\lambda}}{_\lambda\langle\bar{\Omega}|\bar{\Omega}\rangle_\lambda} = \langle\Omega|O(t,x)e^{\lambda Q}|\Omega\rangle^c = \langle\Omega|O(t,x)|\bar{\Omega}\rangle_{2\lambda}^c \tag{267}$$

but the analysis is straightforward to repeat for the symmetric cases $_\lambda\langle\bar{\Omega}|(.)|\bar{\Omega}\rangle_\lambda$ as well and the only difference regards the time evolution of operators that do not commute with $Q$.

We start by rewriting the biased initial state in finite volume as can be done directly and in an easy way. The key observation is that

$$e^{\lambda Q}|\Omega\rangle_L = N_L^{-1/2}\prod_{k>0} e^{\lambda q(k)a_k^\dagger a_k} e^{\lambda q(k)a_{-k}^\dagger a_{-k}}\left(1 + K(k)a_k^\dagger a_{-k}^\dagger\right)|0\rangle \tag{268}$$

as $q(k) = q(-k)$. The above expression is is easy to evaluate, since

$$[a^\dagger a, a^\dagger] = a^\dagger \to [(a^\dagger a)^n, a^\dagger] = a^\dagger, \text{ for } n \ge 1, \to [e^{\beta h a^\dagger a}, a^\dagger] = (e^{\beta h} - 1)a^\dagger, \quad (269)$$

therefore

$$
\begin{aligned}
e^{\lambda Q}|\Omega\rangle_L &= N_L^{-1/2} \prod_{k>0} e^{\lambda q(k) a_k^\dagger a_k} e^{\lambda q(k) a_{-k}^\dagger a_{-k}} \left(1 + K(k) a_k^\dagger a_{-k}^\dagger\right) |0\rangle \\
&= N_L^{-1/2} \prod_{k>0} \left(1 + K(k) e^{2\lambda q(k)} a_k^\dagger a_{-k}^\dagger\right) |0\rangle,
\end{aligned}
\quad (270)
$$

or in other words, the $K$-function $K(k)$ merely modifies to $K(k)e^{2\lambda q(k)}$. Considering for simplicity a quadratic, but generic local operator $O(x, t)$ and writing it as

$$
\begin{aligned}
O(t, x) =& \frac{1}{L} \sum_{p, p'} h_O(p, p') a_p^\dagger a_{p'} e^{i(p-p')x} e^{-it(E_p - E_{p'})} \\
&+ \tilde{h}_O^*(p, p') a_p^\dagger a_{p'}^\dagger e^{-ii(p+p')x} e^{it(E_p + E_{p'})} + \tilde{h}_O(p, p') a_p a_{p'} e^{i(p+p')x} e^{-it(E_p + E_{p'})} \\
=& O^{+-} + O^{++} + O^{--},
\end{aligned}
\quad (271)
$$

we have to deal with 3 types of terms. From its $a^\dagger a$ part only the $p = p'$ elements are non-vanishing consequently no space or time-dependence is originates from the diagonal term, as expected. The corresponding expectation value is easy to obtain. Focusing on a single bilinear $a_p^\dagger a_p$ with a fixed $p$, we have

$$\frac{\langle\Omega|a_p^\dagger a_p e^{\lambda Q}|\Omega\rangle}{\langle\Omega|e^{\lambda Q}|\Omega\rangle} = \frac{\langle 0| \prod_{k'>0} \left(1 + K^*(k') a_{-k'} a_{k'}\right) a_p^\dagger a_p \prod_{k>0} \left(1 + K(k) e^{2\lambda q(k)} a_k^\dagger a_{-k}^\dagger\right) |0\rangle}{\prod_{k>0} \left(1 + |K(k)|^2 e^{2\lambda q(k)}\right)}. \quad (272)$$

Form the products, only $k' = |p|$ and $k = |p|$ have to be taken into account. For positive and negative $p$, we have

$$\frac{\langle\Omega|a_p^\dagger a_p e^{\lambda Q}|\Omega\rangle}{\langle\Omega|e^{\lambda Q}|\Omega\rangle} = \frac{|K(p)|^2 e^{2\lambda q(k)}}{1 + |K(p)|^2 e^{2\lambda q(k)}} \langle 0|a_{\pm p} a_{\pm p}^\dagger|0\rangle \quad (273)$$

respectively, where $|K(-p)| = |K(p)|$ and $q(p) = q(-p)$ were used. Therefore, summing over all $p$ values, we have

$$\frac{\langle\Omega|O^{+-}(t, x)e^{\lambda Q}|\Omega\rangle}{\langle\Omega|e^{\lambda Q}|\Omega\rangle} = \frac{1}{L} \sum_p h_O(p) \frac{|K(p)|^2 e^{2\lambda q(p)}}{1 + |K(p)|^2 e^{2\lambda q(p)}} = \int_{-\infty}^{\infty} \frac{dp}{2\pi} h_O(p) \frac{|K(p)|^2 e^{2\lambda q(p)}}{1 + |K(p)|^2 e^{2\lambda q(p)}}, \quad (274)$$

in the thermodynamic limit. We can easily evaluate the $a^\dagger a^\dagger$ and the $aa$ type terms as well. In these cases only $p = -p'$ terms from the expansion of $O^{++}(x, t)$ and $O^{--}(x, t)$ give non-vanishing contributions resulting in the overall expression. In particular,

$$\frac{\langle 0| \left(1 + K^*(p) a_{-p} a_p\right) a_p^\dagger a_{-p}^\dagger \left(1 + K(p) e^{2\lambda q(p)} a_p^\dagger a_{-p}^\dagger\right) |0\rangle}{\left(1 + |K(p)|^2 e^{2\lambda q(p)}\right)} \quad (275)$$

contributes to

$$\frac{K^*(p) \tilde{h}_O^*(p, -p)}{\left(1 + |K(p)|^2 e^{2\lambda q(p)}\right)}, \quad (276)$$

and $a_p^\dagger a_{-p}^\dagger$ amounts to the same factor due to $q(p) = q(-p)$; whereas

$$\frac{\langle 0| \left(1 + K^*(p) a_{-p} a_p\right) a_p a_{-p} \left(1 + K(p) e^{2\lambda q(p)} a_p^\dagger a_{-p}^\dagger\right) |0\rangle}{\left(1 + |K(p)|^2 e^{2\lambda q(p)}\right)} \quad (277)$$

yields

$$\frac{K(p)e^{2\lambda q(p)}\tilde{h}_O(p,-p)}{\left(1+|K(p)|^2 e^{2\lambda q(p)}\right)}.\tag{278}$$

Therefore, again performing the thermodynamic limit as well, we end up with the final expression

$$\frac{\langle \Omega|O(t,x)|\bar{\Omega}\rangle_{2\lambda}}{{}_{\lambda}\langle\bar{\Omega}|\bar{\Omega}\rangle_{\lambda}} = \int_{-\infty}^{\infty}\frac{\mathrm{d}p}{2\pi}h_O(p)\frac{|K(p)|^2 e^{2\lambda q(p)}}{1+|K(p)|^2 e^{2\lambda q(p)}} \\ + \int_{-\infty}^{\infty}\frac{\mathrm{d}p}{2\pi}\frac{\tilde{h}_O^*(p)K^*(p)e^{2\mathrm{i}TE_p}+\tilde{h}_O(p)K(p)e^{2\lambda q(p)}e^{-2\mathrm{i}TE_p}}{1+|K(p)|^2 e^{2\lambda q(p)}}.\tag{279}$$

and it is easy to see that for the symmetric initial state only the time dependent part gets modified to

$$\frac{{}_{\lambda}\langle\bar{\Omega}|O(t,x)|\bar{\Omega}\rangle_{\lambda}}{{}_{\lambda}\langle\bar{\Omega}|\bar{\Omega}\rangle_{\lambda}} = \int_{-\infty}^{\infty}\frac{\mathrm{d}p}{2\pi}h_O(p)\frac{|K(p)|^2 e^{2\lambda q(p)}}{1+|K(p)|^2 e^{2\lambda q(p)}} \\ + \int_{-\infty}^{\infty}\frac{\mathrm{d}p}{2\pi}\frac{\tilde{h}_O^*(p)K^*(p)e^{2\mathrm{i}TE_p}e^{\lambda q(p)}+\tilde{h}_O(p)K(p)e^{\lambda q(p)}e^{-2\mathrm{i}TE_p}}{1+|K(p)|^2 e^{2\lambda q(p)}}.\tag{280}$$

If now we send $t$ to infinity too, we obtain

$$\lim_{t\to\infty}\frac{\langle\Omega|e^{\lambda/2Q}O(t,x)e^{\lambda/2Q}|\Omega\rangle}{\langle\Omega|e^{\lambda Q}|\Omega\rangle}=\lim_{t\to\infty}\frac{\langle\Omega|O(t,x)e^{\lambda Q}|\Omega\rangle}{\langle\Omega|e^{\lambda Q}|\Omega\rangle}=\int_{-\infty}^{\infty}\frac{\mathrm{d}p}{2\pi}h_O(p)\frac{|K(p)|^2 e^{2\lambda q(p)}}{1+|K(p)|^2 e^{2\lambda q(p)}},\tag{281}$$

which equals the expectation value wrt. $\rho_{\mathrm{GGE}}e^{2\lambda Q}$ as shown in Appendix C.1 upon substituting $|K(k)|^2$ by $|K(k)|^2 e^{2\lambda q(k)}$ in the corresponding expressions.

## C.3 Bipartite quenches and $\lambda$-expansion for free fermion GGEs

We first consider expectation values of the 1- and 2-point functions in the NESS emerging after the partitioning protocol at ray $\lim_{x,t\to\infty}x/t=\xi=0$. The aforementioned NESS is a maximum entropy state and can be written as a GGE. The initial condition for the partitioning protocol can be written as

$$\rho_{\mathrm{in}}\propto(\rho_{\mathrm{GGE}})^L\otimes\left(\rho_{\mathrm{GGE}}^{2\lambda}\right)^R\tag{282}$$

where the R and L superscripts mean that the density matrices are constrained on right and left semi-infinite halves of the system and we assume them to be normalised as well. The NESS can in fact be described by a GGE $\rho_{\mathrm{bpp}}(\lambda)$ [61]

$$\rho_{\mathrm{bpp}}(\lambda)=\frac{1}{Z^{\mathrm{bpp}}}\exp\left(\int_{-\infty}^{0}\mathrm{d}p\,\ln\left(|K(p)|^2 e^{2\lambda q(p)}\right)a_p^{\dagger}a_p+\int_{0}^{\infty}\mathrm{d}p\,\ln\left(|K(p)|^2\right)a_p^{\dagger}a_p\right),\tag{283}$$

and by the filling function filling function

$$\theta_{\mathrm{bpp}}(k)=\Theta_{\mathrm{H}}(v(k))\theta[|K|^2](k)+\Theta_{\mathrm{H}}(-v(k))\theta[|K|^2 e^{2\lambda q}](k),\tag{284}$$

same as in (69) and by the fermionic contraction rules

$$\langle a_p^{\dagger}a_{p'}^{\dagger}\rangle_{\rho_{\mathrm{bpp}}}=0,\qquad\qquad\langle a_p a_{p'}\rangle_{\rho_{\mathrm{bpp}}}=0,$$
$$\langle a_p^{\dagger}a_{p'}\rangle_{\rho_{\mathrm{bpp}}}=\delta(p-p')G(p)$$
$$=\delta(p-p')\Theta_{\mathrm{H}}(-p)\frac{|K(p)|^2 e^{2\lambda q(p)}}{1+|K(p)|^2 e^{2\lambda q(p)}}+\delta(p-p')\Theta_{\mathrm{H}}(p)\frac{|K(p)|^2}{1+|K(p)|^2}\tag{285}$$
$$\langle a_p a_{p'}^{\dagger}\rangle_{\rho_{\mathrm{bpp}}}=\delta(p-p')\tilde{G}(p)$$
$$=\delta(p-p')\Theta_{\mathrm{H}}(-p)\frac{1}{1+|K(p)|^2 e^{2\lambda q(p)}}+\delta(p-p')\Theta_{\mathrm{H}}(p)\frac{1}{1+|K(p)|^2},$$

together with the rules of Wick's theorem, since the above GGE is Gaussian. We used the shorthand notation

$$\text{Tr}\big[O(0,0)\rho_{\text{bpp}}(\lambda)\big] =: \langle O \rangle_{\rho_{\text{bpp}}} . \tag{286}$$

We evaluate the 1- and 2-point functions in this GGE and expand them up to 2nd and 1st order in $\lambda$, respectively. The charge and current densities can be generally written as

$$q_i(x,0) = \frac{1}{2\pi} \int \mathrm{d}k\mathrm{d}p \, e^{-\mathrm{i}(k-p)x} \, q_i(k,p)a_k^\dagger a_p \,, \quad j_i(0,t) = \frac{1}{2\pi} \int \mathrm{d}k\mathrm{d}p \, e^{\mathrm{i}(E_k-E_p)t}(k+p) \, q_i(k,p)a_k^\dagger a_p \,, \tag{287}$$

for local quantities.

### C.3.1   Current 1-pt function in the GGE $\rho_{\text{bpp}}$

It is easy to evaluate the expectation value of the current in the non-equilibrium steady-state after the partitioning protocol described by above GGE $\langle j \rangle_{\rho_{\text{bpp}}}$ and also its value at the first and second orders wrt. $\lambda$. Note that it is sufficient to focus only on the particular current associated with the charge whose FCS we are interested in, since we are eventually interested in autocorrelation type multi-point function of this current in GGE-s and after the biased quench. The entire expectation value is given by

$$\langle j \rangle_{\rho_{\text{bpp}}} = \int_0^\infty \frac{\mathrm{d}k}{2\pi} v(k)q(k) \frac{|K(k)|^2}{1+|K(k)|^2} + \int_{-\infty}^0 \frac{\mathrm{d}k}{2\pi} v(k)q(k) \frac{|K(k)|^2 e^{2\lambda q(k)}}{1+|K(k)|^2 e^{2\lambda q(k)}} \tag{288}$$

and hence

$$\partial_\lambda \langle j \rangle_{\rho_{\text{bpp}}}|_{\lambda=0} = 2 \int_{-\infty}^0 \frac{\mathrm{d}k}{2\pi} v(k)q(k) \frac{|K(k)|^2}{(1+|K(k)|^2)^2} = - \int_{-\infty}^\infty \frac{\mathrm{d}k}{2\pi} |v(k)|q(k) \frac{|K(k)|^2}{(1+|K(k)|^2)^2} . \tag{289}$$

For the second order, we have

$$\begin{aligned}\partial_\lambda^2 \langle j \rangle_{\rho_{\text{bpp}}}|_{\lambda=0} &= 4 \int_{-\infty}^0 \frac{\mathrm{d}k}{2\pi} v(k)q^2(k) \frac{|K(k)|^2(1-|K(k)|^2)}{(1+|K(k)|^2)^3} \\ &= -2 \int_{-\infty}^\infty \frac{\mathrm{d}k}{2\pi} |v(k)|q^2(k) \frac{|K(k)|^2(1-|K(k)|^2)}{(1+|K(k)|^2)^3} .\end{aligned} \tag{290}$$

### C.3.2   Charge 1-pt functions in the GGE $\rho_{\text{bpp}}$

It is easy to evaluate the expectation value of the charge in the non-equilibrium steady-state after the bipartitioning protocol described by above GGE $\langle q_\kappa \rangle_{\rho_{\text{bpp}}}$ and also its value at the first and second orders wrt. $\lambda$. Note, that to argue for the fast enough emergence of the above NESS, we need to consider all local charges, hence we will use the subscript $\kappa$ to distinguish them, but it can refer to a quasi-local charge with a continuous index as well. It is also important to distinguish the parity even and odd charges, therefore we shall you the corresponding superscript e or o as well and we keep in mind that the charge, biasing the GGE is an even one. The expectation value for both even and odd charge densities is given by

$$\langle q_\kappa^{e/o} \rangle_{\rho_{\text{bpp}}} = \int_0^\infty \frac{\mathrm{d}k}{2\pi} q_\kappa^{e/o}(k) \frac{|K(k)|^2}{1+|K(k)|^2} + \int_{-\infty}^0 \frac{\mathrm{d}k}{2\pi} q_\kappa^{e/o}(k) \frac{|K(k)|^2 e^{2\lambda q(k)}}{1+|K(k)|^2 e^{2\lambda q(k)}} \tag{291}$$

and hence

$$\langle q_\kappa^e \rangle_{\rho_{\text{bpp}}}|_{\lambda=0} = \int_{-\infty}^\infty \frac{\mathrm{d}k}{2\pi} q_\kappa^e(k) \frac{|K(k)|^2}{(1+|K(k)|^2)^2} \tag{292}$$

$$\langle q_\kappa^o \rangle_{\rho_{\text{bpp}}}|_{\lambda=0} = 0 \,,$$

since $q(k)$, and $q_\kappa^e(k)$ are even and $q_\kappa(k)$ are odd functions; and

$$\partial_\lambda \langle q_\kappa^e \rangle_{\rho_{\text{bpp}}}|_{\lambda=0} = 2 \int_{-\infty}^0 \frac{dk}{2\pi} q_\kappa^e(k)q(k) \frac{|K(k)|^2}{(1+|K(k)|^2)^2} = \int_{-\infty}^\infty \frac{dk}{2\pi} q_\kappa^e(k)q(k) \frac{|K(k)|^2}{(1+|K(k)|^2)^2}$$

$$\partial_\lambda \langle q_\kappa^o \rangle_{\rho_{\text{bpp}}}|_{\lambda=0} = -\int_{-\infty}^\infty \frac{dk}{2\pi} \text{sgn}(k) q_\kappa^o(k)q(k) \frac{|K(k)|^2}{(1+|K(k)|^2)^2} \, .$$

(293)

For the second order, we have

$$\partial_\lambda^2 \langle q_\kappa^e \rangle_{\rho_{\text{bpp}}}|_{\lambda=0} = 4 \int_{-\infty}^0 \frac{dk}{2\pi} q_\kappa^e(k)q^2(k) \frac{|K(k)|^2(1-|K(k)|^2)}{(1+|K(k)|^2)^3}$$

$$= 2 \int_{-\infty}^\infty \frac{dk}{2\pi} q_\kappa^e(k)q^2(k) \frac{|K(k)|^2(1-|K(k)|^2)}{(1+|K(k)|^2)^3}$$

(294)

$$\partial_\lambda^2 \langle q_\kappa^o \rangle_{\rho_{\text{bpp}}}|_{\lambda=0} = -2 \int_{-\infty}^\infty \frac{dk}{2\pi} \text{sgn}(k) q_\kappa^o(k)q^2(k) \frac{|K(k)|^2(1-|K(k)|^2)}{(1+|K(k)|^2)^3} \, .$$

### C.3.3   Current 2-pt function in the GGE $\rho_{\text{bpp}}$

Regarding the 2-point functions, for our analysis it is again sufficient to focus only on the particular current associated with the charge whose FCS we are interested in. For this two-point function, we simply have that

$$\langle j(0,t)j(0,t') \rangle_{\rho_{\text{bpp}}}^c =$$

$$\frac{1}{(2\pi)^2} \int dk\,dp\,dk'\,dp'\, e^{i(E_k-E_p)t} q(k,p)(k+p) e^{i(E_{k'}-E_{p'})t'} q(k',p')(k'+p') \langle a_k^\dagger a_{p'} \rangle_{\rho_{\text{bpp}}} \langle a_p a_{k'}^\dagger \rangle_{\rho_{\text{bpp}}}$$

$$= \int \frac{dk}{2\pi} \frac{dp}{2\pi} e^{i(E_k-E_p)(t-t')} q^2(k,p)(k+p)^2 G(k)\tilde{G}(p) \, .$$

(295)

To obtain its value at the first order wrt. $\lambda$, we have

$$\partial_\lambda \langle j(0,t)j(0,t') \rangle_{\rho_{\text{bpp}}}^c |_{\lambda=0} =$$

$$= \int \frac{dk}{2\pi} \frac{dp}{2\pi} e^{i(E_k-E_p)(t-t')} q^2(k,p)(k+p)^2 \left( G'(k)\tilde{G}(p)|_{\lambda=0} + G(k)\tilde{G}'(p)|_{\lambda=0} \right)$$

$$= 2 \int_{-\infty}^0 \frac{dk}{2\pi} \int_{-\infty}^\infty \frac{dp}{2\pi} e^{i(E_k-E_p)(t-t')} q^2(k,p)(k+p)^2 q(k) \frac{|K(k)|^2}{(1+|K(k)|^2)^2} \frac{1}{1+|K(p)|^2}$$

(296)

$$- 2 \int_{-\infty}^\infty \frac{dk}{2\pi} \int_{-\infty}^0 \frac{dp}{2\pi} e^{i(E_k-E_p)(t-t')} q^2(k,p)(k+p)^2 q(p) \frac{|K(k)|^2}{1+|K(k)|^2} \frac{|K(p)|^2}{(1+|K(p)|^2)^2} \, ,$$

and eventually

$$\partial_\lambda \langle j(0,t)j(0,t') \rangle_{\rho_{\text{bpp}}}^c |_{\lambda=0} =$$

$$= \int_{-\infty}^\infty \frac{dk}{2\pi} \int_{-\infty}^\infty \frac{dp}{2\pi} e^{i(E_k-E_p)(t-t')} q^2(k,p)(k+p)^2 q(k) \frac{|K(k)|^2}{(1+|K(k)|^2)^2} \frac{1}{1+|K(p)|^2}$$

(297)

$$- \int_{-\infty}^\infty \frac{dk}{2\pi} \int_{-\infty}^\infty \frac{dp}{2\pi} e^{i(E_k-E_p)(t-t')} q^2(k,p)(k+p)^2 q(p) \frac{|K(k)|^2}{1+|K(k)|^2} \frac{|K(p)|^2}{(1+|K(p)|^2)^2} \, ,$$

where we recall that $q(k) = q(k,k)$.

# D  Relaxation of conserved densities to the NESS for free fermions

In this appendix we eventually demonstrate that quenching from $|\tilde{\Omega}\rangle_\lambda$ in free fermion systems, conserved densities converge to a NESS $\rho_{\text{NESS}}(\lambda)$ and the corresponding density matrix also equals $\rho_{\text{bpp}}(\lambda)$, where $\rho_{\text{bpp}}(\lambda)$ is the NESS of a slightly different partitioning protocol defined in (22).

## D.1  1-pt functions of parity odd conserved densities and their approach to the NESS

Now we consider the relaxation of parity odd local conserved densities. The main recognition is, that the treatment of these densities is completely analogous to the charge of a parity even density. This is because they have the same parity properties and consequently the corresponding function $H_{k,p}^{(1)}$ has the same symmetry properties, given that $q_i^o(k,p)$ is an odd function (upon changing sign of both variables, i.e., $q_i^o(-k,-p) = -q_i^o(k,p)$). This means, that we can readily write down the results of the series expansion for any local odd currents, using the results of the previous subsection by specifying $H_{k,p}^{(1)}$ as $e^{i(E_k-E_p)t}q_i^o(k,p)$. As clear from the notation, throughout this appendix we will label the parity odd local charges as $q_i^o$ where $i$ is a non negative odd integer number (53).

### D.1.1  First order in $\langle\Omega|q_i^o(0,t)|\tilde{\Omega}^*\rangle_{2\lambda}$ wrt. $\lambda$: the current expectation value $\langle\Omega|q_i^o(0,t)\tilde{Q}|\Omega\rangle^c$

Regarding $\langle\Omega|q_i^o(0,t)\tilde{Q}|\Omega\rangle^c$, we can immediately write that

$$\langle\Omega|q_i^o(0,t)\tilde{Q}|\Omega\rangle^c = \frac{1}{2}\int\frac{dk'}{2\pi}\frac{dp'}{2\pi}e^{ik'p't}q_i^o\left(\frac{p'+k'}{2},\frac{p'-k'}{2}\right)i\frac{P}{k'}\frac{|K(\frac{k'+p'}{2})|^2 + K^*(\frac{k'+p'}{2})K(\frac{p'-k'}{2})}{(1+|K(\frac{k'+p'}{2})|^2)(1+|K(\frac{p'-k'}{2})|^2)}$$
(298)

from which the stationary value is

$$\langle\Omega|q_i^o(0,\infty)\tilde{Q}|\Omega\rangle^c = -\int\frac{dk}{2\pi}\text{sgn}(k)q_i^o(k)\frac{|K(k)|^2}{(1+|K(k)|^2)^2}.$$
(299)

It is now easy to check that this expression equals (289), that is, the expansion of the GGE expectation value $\langle q_i^o\rangle_{\rho_{\text{bpp}}}$ up to first order after the partitioning protocol, since $v(k) = 2k$.

Regarding the time evolution, for local odd parity charges, we have

$$\langle\Omega|q_i^o(0,t)\tilde{Q}|\Omega\rangle^c = \partial_\lambda\langle q_i^o\rangle_{\rho_{\text{bpp}}}|_{\lambda=0} + \mathcal{O}(t^{-3}),$$
(300)

but the actual exponent is actually larger than 3 and depends on $i$ in a monotonous way (but not strictly monotonous); these details are not important for our purposes to demonstrate the convergence to the NESS.

### D.1.2  First order from the symmetric expression $_\lambda\langle\tilde{\Omega}|q_i^o(0,t)|\tilde{\Omega}\rangle_\lambda$

When considering $_\lambda\langle\tilde{\Omega}|q_i^o(0,t)|\tilde{\Omega}\rangle_\lambda$ we also have to deal with $\langle\Omega|\tilde{Q}q_i^o(0,t)|\Omega\rangle^c$ at the first order, similarly to the case of the current 1-pt function. Omitting the completely analogous calculations we readily have

$$\partial_\lambda\frac{\langle\Omega|e^{\lambda/2\tilde{Q}}q_i^o(0,t)e^{\lambda/2\tilde{Q}}|\Omega\rangle}{\langle\Omega|e^{\lambda\tilde{Q}}|\Omega\rangle}|_{\lambda=0} = \partial_\lambda\langle q_i^o\rangle_{\rho_{\text{bpp}}}|_{\lambda=0} + \mathcal{O}(t^{-3}),$$
(301)

with an actually again an i-dependent, larger then 3 exponent. Contrary to the case $i = 1$, the leading order asymptotic time dependence may differ for larger $i$-s when the symmetry and the asymmetric expectation values are regarded; but this dependence is at least $t^{-3}$ in any case.

### D.1.3 Second order in $\langle\Omega|q_i^o(0,t)|\tilde{\Omega}^*\rangle_{2\lambda}$ wrt. $\lambda$: expectation value of parity odd densities $\langle\Omega|q_i^o(0,t)\tilde{Q}^2|\Omega\rangle^c$

Utilising Eq (178) and eventually (186) we can repeat the analysis and the treatment of the second order expression of the current 1-pt function. Omitting straightforward steps, we arrive at

$$\langle\Omega|q_i^o(0,t)\tilde{Q}^2|\Omega\rangle^c = \partial_\lambda^2\langle q_{2n+1}^o\rangle_{\rho_{\text{bpp}}}|_{\lambda=0} + \mathcal{O}(t^{-3}). \tag{302}$$

Similarly to the first order case, the eventual asymptotic time dependence is stronger than $t^{-3}$ where the exponent again monotonously depends on $i$. This dependence is not strictly monotonous, but the asymptotic time dependence from the second order is no stronger than that of the first order, as we checked for $i = 1, 3, 5$.

### D.1.4 Second order from the symmetric expression $_\lambda\langle\tilde{\Omega}|q_i^o(0,t)|\tilde{\Omega}\rangle_\lambda$

In order to evaluate the 2nd order expansion of $_\lambda\langle\tilde{\Omega}|q_i^o(0,t)|\tilde{\Omega}\rangle_\lambda$, we need to compute two additional correlation functions besides $\langle\Omega|q_i^o(0,t)\tilde{Q}^2|\Omega\rangle^c$, namely $\langle\Omega|\tilde{Q}^2q_i^o(0,t)|\Omega\rangle^c$ and $\langle\Omega|\tilde{Q}\,q_i^o(0,t)\tilde{Q}|\Omega\rangle^c$. These can be done analogously to the case of the current. The end result is also analogous: the second order expansion gives

$$\partial_\lambda^2\frac{\langle\Omega|e^{\lambda/2\tilde{Q}}q_{2n+1}^o(0,t)e^{\lambda/2\tilde{Q}}|\Omega\rangle}{\langle\Omega|e^{\lambda\tilde{Q}}|\Omega\rangle}\Big|_{\lambda=0} = \partial_\lambda^2\langle q_i^o\rangle_{\rho_{\text{bpp}}}|_{\lambda=0} + \mathcal{O}(t^{-3}). \tag{303}$$

Similarly, the eventual decay is stronger than $t^{-3}$ for $i > 1$ but it is not stronger than the corresponding behaviour from the first order as checked for $i = 1, 3, 5$. The exponent may be different (weaker) in the symmetric case than for the asymmetric expectation value.

## D.2 1-pt functions of parity even conserved densities and their approach to the NESS

### D.2.1 Zeroth and first order in $_\lambda\langle\tilde{\Omega}|q_i^e(0,t)|\tilde{\Omega}\rangle_\lambda$ wrt. $\lambda$

When the zeroth order is considered, it is immediate that

$$\langle\Omega|q_i^e(0,t)|\Omega\rangle^c = \int\frac{dk}{2\pi}\frac{|K(k)|^2}{(1+|K(k)|^2)}q_i^e(k), \tag{304}$$

without time dependence, since the initial state is translational invariant and $Q_i$ is conserved. Clearly, (304) matches with (292), that is

$$\frac{\langle\Omega|q_i^e(0,t)e^{\lambda Q}|\Omega\rangle}{\langle\Omega|e^{\lambda Q}|\Omega\rangle}\Big|_{\lambda=0} = \frac{\langle\Omega|e^{\lambda/2Q}q_i^e(0,t)e^{\lambda/2Q}|\Omega\rangle}{\langle\Omega|e^{\lambda Q}|\Omega\rangle}\Big|_{\lambda=0} = \langle q_i^e\rangle_{\rho_{\text{bpp}}}|_{\lambda=0}. \tag{305}$$

Regarding $\langle\Omega|q_i^e(0,t)\tilde{Q}|\Omega\rangle^c$, according to (177), that we have

$$\langle\Omega|q_i^e(0,t)\tilde{Q}|\Omega\rangle^c = \int\frac{dk}{2\pi}\frac{dp}{2\pi}\frac{|K(k)|^2}{1+|K(k)|^2}\frac{1}{1+|K(p)|^2}e^{i(E_k-E_p)t}q_i^e(k,p)\left(\pi\delta(p-k)-iP\frac{1}{p-k}\right)$$
$$+\frac{K^*(k)}{1+|K(k)|^2}\frac{K(p)}{1+|K(p)|^2}e^{i(E_k-E_p)t}q_i^e(k,p)\left(\pi\delta(-k+p)-iP\frac{1}{-k+p}\right). \tag{306}$$

Non-vanishing contributions only comes from the Dirac-$\delta$-s: the contributions from the principal value integrals vanish, which can be seen by performing the transformations $k \to -k$ and $p \to -p$. We can therefore write the integrals as

$$\langle\Omega|q_i^e(0,t)\tilde{Q}|\Omega\rangle^c = \int\frac{dk}{2\pi}\frac{|K(k)|^2}{(1+|K(k)|^2)^2}q_i^e(k) \tag{307}$$

It is also obvious that $\langle\Omega|q_i(0,t)\tilde{Q}|\Omega\rangle^c = \langle\Omega|\tilde{Q}q_i^e(0,t)|\Omega\rangle^c$ and it is immediate to check that this expression equals (293) when $q(k)=1$ (i.e., when the FCS of the particle number is regarded), that is, the expansion of the GGE expectation value $\langle q_i^e\rangle_{\rho_{\text{bpp}}}$ up to first order after the partitioning protocol and regarding the time evolution, no time-dependent term is obtained in the first order in $\lambda$:

$$\partial_\lambda \frac{\langle\Omega|q_i^e(0,t)e^{\lambda Q}|\Omega\rangle}{\langle\Omega|e^{\lambda Q}|\Omega\rangle}\Big|_{\lambda=0} = \partial_\lambda \frac{\langle\Omega|e^{\lambda/2Q}q_i^e(0,t)e^{\lambda/2Q}|\Omega\rangle}{\langle\Omega|e^{\lambda Q}|\Omega\rangle}\Big|_{\lambda=0} = \partial_\lambda \langle q_i^e\rangle_{\rho_{\text{bpp}}}|_{\lambda=0}. \tag{308}$$

### D.2.2 Second order in $_\lambda\langle\tilde{\Omega}|q_i^e(0,t)|\tilde{\Omega}^*\rangle_{2\lambda}$ wrt. $\lambda$: $\langle\Omega|q_i^e(0,t)\tilde{Q}^2|\Omega\rangle^c$

The analysis is the second order expression is slightly more complicated. Using (178), we arrive at

$$\langle\Omega|q_i^e(0,t)\tilde{Q}^2|\Omega\rangle^c = \int \frac{dk_1}{2\pi}\frac{dk_2}{2\pi}\frac{dk_3}{2\pi} \frac{1}{1+|K(k_1)|^2}\frac{1}{1+|K(k_2)|^2}\frac{1}{1+|K(k_3)|^2} e^{i(E_{k_1}-E_{k_2})t}q_i^e(k_1,k_2)$$

$$\left[K^*(k_1)K(-k_3)\left(\pi\delta(k_1+k_3)+\text{iP}\frac{1}{k_1+k_3}\right)\left(\pi\delta(k_2+k_3)-\text{iP}\frac{1}{k_2+k_3}\right)\right.$$

$$-K^*(k_1)K(-k_3)\left(\pi\delta(k_2-k_3)-\text{iP}\frac{1}{k_2-k_3}\right)\left(\pi\delta(k_1-k_3)+\text{iP}\frac{1}{k_1-k_3}\right)$$

$$-|K(k_1)|^2|K(k_3)|^2\left(\pi\delta(k_3-k_1)-\text{iP}\frac{1}{k_3-k_1}\right)\left(\pi\delta(k_2-k_3)-iP\frac{1}{k_2-k_3}\right)$$

$$+K^*(k_1)K(-k_2)|K(k_3)|^2\left(\pi\delta(k_2+k_3)-\text{iP}\frac{1}{k_2+k_3}\right)\left(\pi\delta(k_1+k_3)+\text{iP}\frac{1}{k_1+k_3}\right)$$

$$-K^*(k_1)K(-k_2)\left(\pi\delta(k_1+k_3)+\text{iP}\frac{1}{k_1+k_3}\right)\left(\pi\delta(k_3+k_2)-\text{iP}\frac{1}{k_3+k_2}\right)$$

$$+|K(k_1)|^2K(-k_2)K^*(k_3)\left(\pi\delta(k_3-k_1)-\text{iP}\frac{1}{k_3-k_1}\right)\left(\pi\delta(k_3-k_2)+\text{iP}\frac{1}{k_3-k_2}\right)$$

$$+|K(k_1)|^2\left(\pi\delta(k_2-k_3)-iP\frac{1}{k_2-k_3}\right)\left(\pi\delta(k_3-k_1)-\text{iP}\frac{1}{k_3-k_1}\right)$$

$$\left.-|K(k_1)|^2K(-k_2)K^*(k_3)\left(\pi\delta(k_2+k_3)-\text{iP}\frac{1}{k_2+k_3}\right)\left(\pi\delta(k_3+k_1)+\text{iP}\frac{1}{k_3+k_1}\right)\right]. \tag{309}$$

Since $q_i^e(k_2)$ is an even function, we find that, performing the change of variables $k_1 \to -k_1$, $k_2 \to -k_2$ and $k_3 \to -k_3$, that by symmetry, the non-vanishing contributions come from multiplying two Dirac-$\delta$-s, or two principal value terms. This means, that the treatment of (309) is slightly different from that of the parity odd current. It is easy to see that the Dirac-$\delta$-s contribute to time-independent expressions, which are immediate to evaluate

$$\langle\Omega|q_i^e(0,\infty)\tilde{Q}^2|\Omega\rangle^c = \int \frac{dk}{2\pi}\frac{|K(k)|^2-|K(k)|^4}{(1+|K(k)|^2)^3}q_i^e(k) = \frac{1}{2}\partial_\lambda^2\langle q_i^e\rangle_{\rho_{\text{bpp}}}|_{\lambda=0} \tag{310}$$

However, the double principal value terms also contribute to such time independent term. In particular, we have recall that the order of integration matters in such case, and in (309) the first integration is performed wrt. $k_1$ (or $k_2$) as we claimed in Section 5.1.3 and as we shall now demonstrate it explicitly.

Concentrating on the order of integration, the easiest way to see this as follows. One has to realise that when the charge operators are multiplied, the correct way of writing their contribution is

$$\lim_{\epsilon\to 0}\int dk_1 dk_2 dk_3 \frac{-\text{i}}{k_3-k_1-\text{i}\epsilon}\frac{\text{i}}{k_3-k_2+\text{i}\epsilon}, \tag{311}$$

and sending $\epsilon$ to zero (corresponding to first integrating the charge density up to a fine region and then extending the upper boundary of this region to infinity). The term (311) is therefore present in all in 3-fold integrals of (186) and in this rewriting the order of integration does not matter. However, writing

$$\lim_{\epsilon \to 0} \frac{-i}{k_3 - k_1 - i\epsilon} \frac{i}{k_3 - k_2 + i\epsilon} = \left( \pi\delta(k_3 - k_1) - i\frac{P}{k_3 - k_1} \right)\left( \pi\delta(k_3 - k_2) + i\frac{P}{k_3 - k_2} \right), \quad (312)$$

that is, using the (170) for a product like (311) is only allowed if first the $k_1$ or $k_2$ is performed. This is due to the fact that $k_3$ appears in two terms, whereas $k_1$ and $k_2$ in only once. Unlike $k_1$ and $k_2$, $k_3$ approaches two poles close to the real line; moreover, it is not obvious how to carry out the contour manipulations (discussed in 5.1.1) if first the $k_3$ is intended to be performed and $k_1 = k_2$. On the contrary, if first the $k_1$ and/or $k_2$ integration is performed no ambiguity appears. In summary, we eventually have that, in the distribution sense,

$$\lim_{\epsilon \to 0} \int dk_1 \, dk_2 \, dk_3 \, \frac{-i}{k_3 \pm k_1 - i\epsilon_1} \frac{i}{k_3 \pm k_2 + i\epsilon_2}$$

$$= \int dk_1 \, dk_2 \, dk_3 \left( \pi\delta(k_3 \pm k_1) - i\frac{P}{k_3 \pm k_1} \right)\left( \pi\delta(k_3 \pm k_2) + i\frac{P}{k_3 \pm k_1} \right) \quad (313)$$

$$:= \int dk_3 \int dk_2 \int dk_1 \left( \pi\delta(k_3 \pm k_1) - i\frac{P}{k_3 \pm k_1} \right)\left( \pi\delta(k_3 \pm k_2) + i\frac{P}{k_3 \pm k_1} \right)$$

where from the third line the order of integration goes from the right to left in $\int dk_i$.

However, to evaluate (313) within (309), it is useful to perform first the integration wrt. $k_3$. Changing the order integration for the principal value integrals can be carried out thanks to the Poincaré-Bertrand theorem, which claims that

$$\int dk_3 \int dk_2 \int dk_1 \frac{P}{k_3 \pm k_1} \frac{P}{k_3 \pm k_2} - \pi^2 \delta(k_3 \pm k_1)\delta(k_3 \pm k_2) = \int dk_1 \int dk_2 \int dk_3 \frac{P}{k_3 \pm k_1} \frac{P}{k_3 \pm k_2},$$
$$(314)$$

in which the integrations wrt. $k_1$ and $k_2$ can be exchanged. The way how the "double principal value" integration is understood in the rhs. of (314) is as follows: the integration wrt. $k_3$ can be performed for any fixed $k_1 \neq k_2$ with $|k_1 - k_2| > 2\epsilon > 0$ and afterwards $\epsilon$ (regulating the principal value integration) is sent to zero. After this step, the $k_1 \to k_2$ limit can be applied. We shall explicitly demonstrate the evaluation of such term in Appendix D.2.2. Using (314), we can eventually rewrite (313) as

$$\int dk_3 \int dk_2 \int dk_1 \left( \pi\delta(k_3 \pm k_1) - i\frac{P}{k_3 \pm k_1} \right)\left( \pi\delta(k_3 \pm k_2) + i\frac{P}{k_3 \pm k_1} \right)$$

$$= \int dk_1 \int dk_2 \int dk_3 \left( \pi\delta(k_3 \pm k_1) - i\frac{P}{k_3 \pm k_1} \right)\left( \pi\delta(k_3 \pm k_2) + i\frac{P}{k_3 \pm k_1} \right) \quad (315)$$

$$+ \pi^2 \delta(k_3 \pm k_1)\delta(k_3 \pm k_2).$$

Applying the aforementioned consideration to the problem of evaluating (309) by first carrying out the $k_3$-integral (as the oscillating functions are independent of $k_3$) we can proceed by focusing on

only the integrands from all the double principal value terms as

$$
\int \frac{dk_1}{2\pi}\frac{dk_2}{2\pi}\frac{dk_3}{2\pi}\, P(k_1,k_2,k_3,t) := \int \frac{dk_3}{2\pi}\int \frac{dk_2}{2\pi}\int \frac{dk_1}{2\pi}\, P(k_1,k_2,k_3,t)
$$

$$
:= \int \frac{dk_3}{2\pi}\int \frac{dk_2}{2\pi}\int \frac{dk_1}{2\pi}\,\frac{1}{1+|K(k_1)|^2}\frac{1}{1+|K(k_2)|^2}\frac{1}{1+|K(k_3)|^2}e^{i(E_{k_1}-E_{k_2})t}q_i^e(k_1,k_2)
$$

$$
\left[-K^*(k_1)K(k_3)\left(\mathrm{P}\frac{1}{k_3+k_1}\right)\left(\mathrm{P}\frac{1}{k_3+k_2}\right)+K^*(k_1)K(k_3)\left(\mathrm{P}\frac{1}{k_3-k_2}\right)\left(\mathrm{P}\frac{1}{k_3-k_1}\right)\right.
$$

$$
-|K(k_1)|^2|K(k_3)|^2\left(\mathrm{P}\frac{1}{k_3-k_1}\right)\left(\mathrm{P}\frac{1}{k_3-k_2}\right)-K^*(k_1)K(k_2)|K(k_3)|^2\left(\mathrm{P}\frac{1}{k_3+k_2}\right)\left(\mathrm{P}\frac{1}{k_3+k_1}\right)
$$

$$
+K^*(k_1)K(k_2)\left(\mathrm{P}\frac{1}{k_1+k_3}\right)\left(\mathrm{P}\frac{1}{k_3+k_2}\right)-|K(k_1)|^2K(k_2)K^*(k_3)\left(\mathrm{P}\frac{1}{k_3-k_1}\right)\left(\mathrm{P}\frac{1}{k_3-k_2}\right)
$$

$$
+|K(k_1)|^2\left(\mathrm{P}\frac{1}{k_3-k_2}\right)\left(\mathrm{P}\frac{1}{k_3-k_1}\right)+\left.|K(k_1)|^2K(k_2)K^*(k_3)\left(\mathrm{P}\frac{1}{k_3+k_2}\right)\left(\mathrm{P}\frac{1}{k_3+k_1}\right)\right].
$$

$$(316)$$

Hence we find, according to (314), that

$$
\int \frac{dk_3}{2\pi}\int \frac{dk_2}{2\pi}\int \frac{dk_1}{2\pi}\, P(k_1,k_2,k_3,t) = \int \frac{dk_1}{2\pi}\int \frac{dk_2}{2\pi}\int \frac{dk_3}{2\pi}\, P(k_1,k_2,k_3,t)
$$

$$
+\int \frac{dk_1}{2\pi}\int \frac{dk_2}{2\pi}\int \frac{dk_3}{2\pi}\,\frac{1}{1+|K(k_1)|^2}\frac{1}{1+|K(k_2)|^2}\frac{1}{1+|K(k_3)|^2}e^{i(E_{k_1}-E_{k_2})t}q_i^e(k_1,k_2)
$$

$$
[-K^*(k_1)K(k_3)\pi\delta(k_3+k_1)\pi\delta(k_3+k_2)+K^*(k_1)K(k_3)\pi\delta(k_3-k_1)\pi\delta(k_3-k_2)
$$

$$
-|K(k_1)|^2|K(k_3)|^2\pi\delta(k_3-k_1)\pi\delta(k_3-k_2)-K^*(k_1)K(k_2)|K(k_3)|^2\pi\delta(k_3+k_1)\pi\delta(k_3+k_2)
$$

$$
+K^*(k_1)K(k_2)\pi\delta(k_3+k_1)\pi\delta(k_3+k_2)-|K(k_1)|^2K(k_2)K^*(k_3)\pi\delta(k_3-k_1)\pi\delta(k_3-k_2)
$$

$$
+|K(k_1)|^2\pi\delta(k_3-k_1)\pi\delta(k_3-k_2)+|K(k_1)|^2K(k_2)K^*(k_3)\pi\delta(k_3+k_1)\pi\delta(k_3+k_2)].
$$

$$(317)$$

It is now easy to see that the additional $\delta$ contributions emerge in (309) which yield

$$
\frac{1}{4}\int \frac{dk}{2\pi}\,\frac{q_i^e(k)}{(1+|K(k)|^2)^3}\left[4|K(k_1)|^2-4|K(k)|^4\right]=\frac{1}{2}\partial_\lambda^2\langle q_i^e\rangle_{\rho_{\mathrm{bpp}}}|_{\lambda=0}\,,
\tag{318}
$$

that is, we have that

$$
\langle\Omega|q_i^e(0,t)\tilde{Q}^2|\Omega\rangle^c = \partial_\lambda^2\langle q_i^e\rangle_{\rho_{\mathrm{bpp}}}|_{\lambda=0}+\int \frac{dk_1}{2\pi}\int \frac{dk_2}{2\pi}\int \frac{dk_3}{2\pi}\, P(k_1,k_2,k_3,t)\,,
\tag{319}
$$

where $P(k_1,k_2,k_3,t)$ is defined in (316).

Now we would like to comment on the time dependence, that is, the convergence to the NESS as well. It is easy to see that the double principal value integrals in (319) are regular, at least for $t > 0$. It is also immediate that for large $t$, the asymptotic time-dependence can be approximated as $t^{-3/2}$, which can be seen by replacing the integration variables $k_i \to k_i\sqrt{t}$ and exploiting that the dominant contribution to the integral comes from regions close to the origin, either due to the suppression of the $K$-functions, or to the oscillatory exponential function. However, it is less obvious whether non-time-dependent bits may be present. If we still assume that $|K(k)|^2 = K_1(k)^2 + K_2(k)^2$ as a complex function is entire and analytic and also tends to zero at $|k_3| \to \infty$ on the upper half complex plane, we can be slightly more precise. In this case, we can perform the integration wrt. $k_3$ in (319) and use that

$$
\int \frac{dk_3}{2\pi}f(k_3)\frac{\mathrm{P}}{k_3\mp k_1}\frac{\mathrm{P}}{k_3\mp k_2} = i\left(\operatorname*{Res}_{k_3=k_3^*}f(k_3)\right)\frac{1}{k_3^*\mp k_1}\frac{1}{k_3^*\mp k_2}\pm\frac{i}{2}\frac{f(\pm k_1)-f(\pm k_2)}{k_1-k_2}\,,
\tag{320}
$$

where we applied the residue theorem for $f(k_3)$ assuming that it is complex analytic and has one single pole at $k_3^*$ (on the upper half complex plane $\mathrm{Im}\, k_3^* > 0$) for simplicity, and we also used the "half-residue" theorem to evaluate the pole contributions at $\pm k_1$ and $\pm k_2$ from the principal value terms. Applying (320) to each term in $P(k_1, k_2, k_3, t)$ is again easy to see by the $k_1 \to -k_1$, $k_2 \to -k_2$ transformation in the integrands, that all $(f(k_1) - f(k_2))/(k_1 - k_2)$ type contributions from (320) vanish. Hence, we end up with the residue contribution of $1/(1 + |K(k_3)|^2)$ and the expression

$$
\begin{aligned}
&\int \frac{\mathrm{d}k_1}{2\pi} \int \frac{\mathrm{d}k_2}{2\pi} \int \frac{\mathrm{d}k_3}{2\pi}\, P(k_1, k_2, k_3, t) = \\
&= \int \frac{\mathrm{d}k_2}{2\pi} \frac{\mathrm{d}k_1}{2\pi} \frac{1}{1 + |K(k_1)|^2} \frac{1}{1 + |K(k_2)|^2} e^{\mathrm{i}(E_{k_1} - E_{k_2})t} q_i^{\mathrm{e}}(k_1, k_2) \mathrm{i}\left( \operatorname*{Res}_{k_3 = k_3^*} \frac{1}{1 + |K(k_3)|^2} \right) \\
&\quad \left[ -K^*(k_1)K(k_3^*)\frac{1}{k_3^* + k_1}\frac{1}{k_3^* + k_2} + K^*(k_1)K(k_3^*)\frac{1}{k_3^* - k_1}\frac{1}{k_3^* - k_2} \right. \\
&\quad - |K(k_1)|^2 |K(k_3^*)|^2 \frac{1}{k_3^* - k_1}\frac{1}{k_3^* - k_2} - K^*(k_1)K(k_2)|K(k_3^*)|^2 \frac{1}{k_3^* + k_1}\frac{1}{k_3^* + k_2} \\
&\quad + K^*(k_1)K(k_2)\frac{1}{k_3^* + k_1}\frac{1}{k_3^* + k_2} - |K(k_1)|^2 K(k_2)K^*(k_3^*)\frac{1}{k_3^* - k_1}\frac{1}{k_3^* - k_2} \\
&\quad \left. + |K(k_1)|^2 \frac{1}{k_3^* - k_1}\frac{1}{k_3^* - k_2} + |K(k_1)|^2 K(k_2)K^*(k_3^*)\frac{1}{k_3^* + k_1}\frac{1}{k_3^* + k_2} \right].
\end{aligned}
\tag{321}
$$

Because each term in (321) is multiplied by at least one functions $(K(k_1))$ which is at least linear at small $k_1$, the asymptotic time dependence, using the rescaling argument $k_i \to k_i \sqrt{t}$ is at least $t^{-3/2}$, but $q_i(k_1, k_2) \neq 1$ eventually makes this decay even stronger. To summarise, we can claim that

$$
\langle \Omega | q_i^{\mathrm{e}}(0, t)\tilde{Q}^2 |\Omega\rangle^c = \partial_\lambda^2 \langle q_i^{\mathrm{e}} \rangle_{\rho_{\mathrm{bpp}}}|_{\lambda=0} + \mathcal{O}(t^{-3/2})
\tag{322}
$$

asymptotically for any $i$, that is, for any parity even local conserved density operator. In other words

$$
\partial_\lambda^2 \frac{\langle \Omega | q_i^{\mathrm{e}}(0, t)e^{\lambda \tilde{Q}}|\Omega\rangle}{\langle \Omega | e^{\lambda \tilde{Q}}|\Omega\rangle}\Big|_{\lambda=0} = \partial_\lambda^2 \langle q_i^{\mathrm{e}} \rangle_{\rho_{\mathrm{bpp}}}|_{\lambda=0} + \mathcal{O}(t^{-3/2}).
\tag{323}
$$

### D.2.3 Second order in $_\lambda\langle \tilde{\Omega} | q_i^{\mathrm{e}}(0, t)|\tilde{\Omega}\rangle_\lambda$ wrt. $\lambda$; the symmetric expectation value

To obtain this quantity, we also need to deal with $\langle \Omega | \tilde{Q}\, q_i^{\mathrm{e}}(0, t)\tilde{Q}|\Omega\rangle^c$, noting that

$$
\langle \Omega | \tilde{Q}^2\, q_i^{\mathrm{e}}(0, t)|\Omega\rangle^c = \left( \langle \Omega | q_i^{\mathrm{e}}(0, t)\tilde{Q}^2 |\Omega\rangle^c \right)^*.
\tag{324}
$$

Using (178), we arrive at

$$\langle\Omega|\tilde{Q}\,q_i^e(0,t)\tilde{Q}|\Omega\rangle^c =$$

$$= \int \frac{dk_1}{2\pi}\frac{dk_2}{2\pi}\frac{dk_3}{2\pi}\frac{1}{1+|K(k_1)|^2}\frac{1}{1+|K(k_2)|^2}\frac{1}{1+|K(k_3)|^2}\left(\pi\delta(k_1-k_2)-\mathrm{i}P\frac{1}{k_1-k_2}\right)$$

$$\left[K^*(k_1)K(-k_3)e^{\mathrm{i}(E_{k_1}-E_{k_3})t}q_i^e(-k_1,k_3)\left(\pi\delta(k_2+k_3)-\mathrm{i}P\frac{1}{k_2+k_3}\right)\right.$$

$$-K^*(k_1)K(-k_3)e^{\mathrm{i}(E_{k_2}-E_{k_3})t}q_i^e(k_2,k_3)\left(\pi\delta(k_1-k_3)+\mathrm{i}P\frac{1}{k_1-k_3}\right)$$

$$-|K(k_1)|^2|K(k_3)|^2e^{\mathrm{i}(E_{k_3}-E_{k_1})t}q_i^e(k_3,k_1)\left(\pi\delta(k_2-k_3)-\mathrm{i}P\frac{1}{k_2-k_3}\right)$$

$$+K^*(k_1)K(-k_2)|K(k_3)|^2e^{\mathrm{i}(E_{k_3}-E_{k_2})t}q_i^e(k_3,-k_2)\left(\pi\delta(k_1+k_3)+\mathrm{i}P\frac{1}{k_1+k_3}\right)$$

$$-K^*(k_1)K(-k_2)e^{\mathrm{i}(E_{k_1}-E_{k_3})t}q_i^e(-k_1,k_3)\left(\pi\delta(k_3+k_2)-\mathrm{i}P\frac{1}{k_3+k_2}\right)$$

$$+|K(k_1)|^2K(-k_2)K^*(k_3)e^{\mathrm{i}(E_{k_3}-E_{k_1})t}q_i^e(k_3,k_1)\left(\pi\delta(k_3-k_2)+\mathrm{i}P\frac{1}{k_3-k_2}\right)$$

$$+|K(k_1)|^2e^{\mathrm{i}(E_{k_2}-E_{k_3})t}q_i^e(k_2,k_3)\left(\pi\delta(k_3-k_1)-\mathrm{i}P\frac{1}{k_3-k_1}\right)$$

$$\left.-|K(k_1)|^2K(-k_2)K^*(k_3)e^{\mathrm{i}(E_{k_3}-E_{k_2})t}q_i^e(k_3,-k_2)\left(\pi\delta(k_3+k_1)+\mathrm{i}P\frac{1}{k_3+k_1}\right)\right].$$

$$(325)$$

Since $q_i^e(k,p)$ is an even function, we find that, performing the change of variables $k_1 \to -k_1$, $k_2 \to -k_2$ and $k_3 \to -k_3$, that by symmetry, the non-vanishing contributions come from multiplying two Dirac-$\delta$-s, or two principal value terms. This means, that the analysis of (325) is similar to that of (309). The Dirac-$\delta$-s contribute to time-independent expressions, which are immediate to evaluate

$$\langle\Omega|\tilde{Q}\,q_i^e(0,\infty)\tilde{Q}|\Omega\rangle^c = \int \frac{dk}{2\pi}\frac{|K(k)|^2-|K(k)|^4}{(1+|K(k)|^2)^3}q_i^e(k) = \frac{1}{2}\partial_\lambda^2\langle q_i^e\rangle_{\rho_{\mathrm{bpp}}}|_{\lambda=0} \qquad (326)$$

However, in complete analogy with the previous term, $\langle\Omega|q_i^e(0,t)\tilde{Q}^2|\Omega\rangle^c$, the double principal value terms also contribute to such time independent term. In particular, we recall that in (326) the first integration is performed wrt. $k_1$ or $k_3$. However, as the oscillating function is independent on $k_2$, it is worthwhile change the orders of integration and first carry out the $k_2$-integral. Denoting the integrands from all the double principal value terms as

$$\int \frac{dk_1}{2\pi}\frac{dk_2}{2\pi}\frac{dk_3}{2\pi}\,P(k_1,k_2,k_3,t) := \int \frac{dk_1}{2\pi}\frac{dk_2}{2\pi}\frac{dk_3}{2\pi}\frac{1}{1+|K(k_1)|^2}\frac{1}{1+|K(k_2)|^2}\frac{1}{1+|K(k_3)|^2}\times$$

$$\left\{\left[-K^*(k_1)K(-k_3)+K^*(k_1)K(-k_2)\right]e^{\mathrm{i}(E_{k_1}-E_{k_3})t}q_i^e(-k_1,k_3)P\frac{1}{k_1-k_2}P\frac{1}{k_2+k_3}\right.$$

$$+\left[-K^*(k_1)K(-k_3)+|K(k_1)|^2\right]e^{\mathrm{i}(E_{k_2}-E_{k_3})t}q_i^e(k_2,k_3)P\frac{1}{k_1-k_2}P\frac{1}{k_1-k_3}$$

$$+\left[|K(k_1)|^2|K(k_3)|^2-|K(k_1)|^2K(-k_2)K^*(k_3)\right]e^{\mathrm{i}(E_{k_3}-E_{k_1})t}q_i^e(k_3,k_1)P\frac{1}{k_1-k_2}P\frac{1}{k_2-k_3}$$

$$\left.+\left[K^*(k_1)K(-k_2)|K(k_3)|^2-|K(k_1)|^2K(-k_2)K^*(k_3)\right]e^{\mathrm{i}(E_{k_3}-E_{k_2})t}q_i^e(k_3,-k_2)P\frac{1}{k_1-k_2}P\frac{1}{k_1+k_3}\right\},$$

$$(327)$$

we can rewrite the expression by relabeling integration variables as

$$\int \frac{dk_1}{2\pi} \frac{dk_2}{2\pi} \frac{dk_3}{2\pi} P(k_1, k_2, k_3, t) := \int \frac{dk_2}{2\pi} \int \frac{dk_1}{2\pi} \int \frac{dk_3}{2\pi} P(k_1, k_2, k_3, t)$$

$$= \int \frac{dk_2}{2\pi} \frac{dk_1}{2\pi} \frac{dk_3}{2\pi} \frac{1}{1+|K(k_1)|^2} \frac{1}{1+|K(k_2)|^2} \frac{1}{1+|K(k_3)|^2} \times e^{i(E_{k_1}-E_{k_3})t} q_i^e(k_1, k_3) \times$$

$$\left[ -K^*(k_1)K(-k_3)\left(\frac{P}{k_2+k_1}\right)\left(\frac{P}{k_2+k_3}\right) - K^*(k_2)K(-k_3)\left(\frac{P}{k_2-k_1}\right)\left(\frac{P}{k_2-k_3}\right) \right.$$

$$-|K(k_1)|^2|K(k_3)|^2\left(\frac{P}{k_2-k_3}\right)\left(\frac{P}{k_2-k_1}\right) + K^*(k_2)K(k_3)|K(k_1)|^2\left(\frac{P}{k_2+k_3}\right)\left(\frac{P}{k_2+k_1}\right)$$

$$+K^*(k_1)K(-k_2)\left(\frac{P}{k_2+k_1}\right)\left(\frac{P}{k_2+k_3}\right) + |K(k_3)|^2K(-k_2)K^*(k_1)\left(\frac{P}{k_2-k_3}\right)\left(\frac{P}{k_2-k_1}\right)$$

$$\left. +|K(k_2)|^2\left(\frac{P}{k_2-k_1}\right)\left(\frac{P}{k_2-k_3}\right) - |K(k_2)|^2K(k_3)K^*(k_1)\left(\frac{P}{k_2+k_3}\right)\left(\frac{P}{k_2+k_1}\right) \right].$$

(328)

Now, according to (314), we find that

$$\int \frac{dk_2}{2\pi} \frac{dk_1}{2\pi} \frac{dk_3}{2\pi} P(k_1, k_2, k_3, t) := \int \frac{dk_1}{2\pi} \int \frac{dk_3}{2\pi} \int \frac{dk_2}{2\pi} P(k_1, k_2, k_3, t)$$
$$+ \int \frac{dk}{2\pi} \frac{|K(k)|^2 - |K(k)|^4}{(1+|K(k)|^2)^3} q_i^e(k).$$

(329)

The principal value terms can be evaluated assuming the analyticity of $|K(k)|^2$. Using (320) and accounting for the residue contribution of $1/(1+|K(k_2)|^2)$ we can write that

$$\int \frac{dk_1}{2\pi} \int \frac{dk_3}{2\pi} \int \frac{dk_2}{2\pi} P(k_1, k_2, k_3, t) =$$

$$= \int \frac{dk_1}{2\pi} \frac{dk_3}{2\pi} \frac{1}{1+|K(k_1)|^2} \frac{1}{1+|K(k_3)|^2} e^{i(E_{k_1}-E_{k_3})t} q_i^e(k_1, k_3) i \left( \operatorname*{Res}_{k_2=k_2^*} \frac{1}{1+|K(k_2)|^2} \right)$$

$$\left[ K^*(k_1)K(k_3)\frac{1}{k_2^*+k_1}\frac{1}{k_2^*+k_3} + K^*(k_2^*)K(k_3)\frac{1}{k_2^*-k_1}\frac{1}{k_2^*-k_3} \right.$$

$$-K^*(k_1)K(k_2^*)\frac{1}{k_2^*+k_1}\frac{1}{k_2^*+k_3} + |K(k_2^*)|^2\frac{1}{k_2^*-k_1}\frac{1}{k_2^*-k_3}$$

$$-|K(k_1)|^2|K(k_3)|^2\frac{1}{k_2^*-k_3}\frac{1}{k_2^*-k_1} + K^*(k_2^*)K(k_3)|K(k_1)|^2\frac{1}{k_2^*+k_3}\frac{1}{k_2^*+k_1}$$

$$\left. -|K(k_3)|^2K(k_2^*)K^*(k_1)\frac{1}{k_2^*-k_3}\frac{1}{k_2^*-k_1} - |K(k_2^*)|^2K(k_3)K^*(k_1)\frac{1}{k_2^*+k_3}\frac{1}{k_2^*+k_1} \right].$$

(330)

Repeating the argument in the previous subsection, we can say that each term in (330) is multiplied by at least one functions ($K(k_1)$) which is at least linear at small $k_1$, the asymptotic time dependence, using the rescaling argument $k_i \to k_i \sqrt{t}$, is at least $t^{-3/2}$, but $q_i^e(k_1, k_3) \neq 1$ eventually makes this decay even stronger. To summarise, we can again claim that

$$\langle \Omega | \tilde{Q} q_i^e(0, t) \tilde{Q} | \Omega \rangle^c = \partial_\lambda^2 \langle q_i^e \rangle_{\rho_{bpp}} |_{\lambda=0} + \mathcal{O}(t^{-3/2})$$

(331)

asymptotically for any $i$, that is, for any local parity even conserved densities.

That is, we have demonstrated that $\langle \Omega | \tilde{Q}^2 q_i^e(0, t) | \Omega \rangle^c$ and $\langle \Omega | \tilde{Q} q_i^e(0, t) \tilde{Q} | \Omega \rangle^c$ behave the same way, which implies that

$$\partial_\lambda^2 \frac{\langle \Omega | e^{\lambda/2 \tilde{Q}} q_i^e(0, t) e^{\lambda/2 \tilde{Q}} | \Omega \rangle}{\langle \Omega | e^{\lambda \tilde{Q}} | \Omega \rangle} |_{\lambda=0} = \partial_\lambda^2 \langle q_i^e \rangle_{\rho_{bpp}} |_{\lambda=0} + \mathcal{O}(t^{-3/2}).$$

(332)

### D.3 Quasi-local charge densities $q_\alpha^{\text{e/o}}(x, t)$

Concerning the behaviour of the quasi-local charges, it is useful to recall (55) and focus first on $q_\alpha(x, t)$. In the Fourier basis it can be written as

$$q_\alpha(x, t) = \frac{1}{2\pi} \int \mathrm{d}k \mathrm{d}p \, e^{\mathrm{i}(E_k - E_p)t} e^{-\mathrm{i}(k-p)x} q_\alpha(k, p) a_k^\dagger a_p, \tag{333}$$

where now $q_\alpha(k, p) = e^{-\mathrm{i}p\alpha} = \cos(\alpha p) - \mathrm{i}\sin(\alpha p)$, hence $q_\alpha(x, t)$ itself can be split into an even and an odd operator and we can readily use the results of the previous subsections. Similarly, for $q_\alpha^\dagger(x, t)$ we find that $q_\alpha^\dagger(k, p) = e^{\mathrm{i}k\alpha} = \cos(\alpha k) + \mathrm{i}\sin(\alpha k)$. Accordingly, it is immediate that

$$\partial_\lambda \frac{\langle \Omega | e^{\lambda/2 \tilde{Q}} q_\alpha(0, t) e^{\lambda/2 \tilde{Q}} | \Omega \rangle}{\langle \Omega | e^{\lambda \tilde{Q}} | \Omega \rangle} |_{\lambda=0} = \partial_\lambda \langle q_\alpha \rangle_{\rho_{\text{bpp}}} |_{\lambda=0} + a_1 t^{-3/2} - \mathrm{i}b_1 t^{-3}$$
$$\partial_\lambda \frac{\langle \Omega | q_\alpha(0, t) e^{\lambda \tilde{Q}} | \Omega \rangle}{\langle \Omega | e^{\lambda \tilde{Q}} | \Omega \rangle} |_{\lambda=0} = \partial_\lambda \langle q_\alpha \rangle_{\rho_{\text{bpp}}} |_{\lambda=0} + a_1' t^{-3/2} - \mathrm{i}b_1' t^{-3}, \tag{334}$$

and similarly for the second order

$$\partial_\lambda^2 \frac{\langle \Omega | e^{\lambda/2 \tilde{Q}} q_\alpha(0, t) e^{\lambda/2 \tilde{Q}} | \Omega \rangle}{\langle \Omega | e^{\lambda \tilde{Q}} | \Omega \rangle} |_{\lambda=0} = \partial_\lambda^2 \langle q_\alpha \rangle_{\rho_{\text{bpp}}} |_{\lambda=0} + a_2 t^{-3/2} - \mathrm{i}b_2 t^{-3}$$
$$\partial_\lambda^2 \frac{\langle \Omega | q_\alpha(0, t) e^{\lambda \tilde{Q}} | \Omega \rangle}{\langle \Omega | e^{\lambda \tilde{Q}} | \Omega \rangle} |_{\lambda=0} = \partial_\lambda^2 \langle q_\alpha \rangle_{\rho_{\text{bpp}}} |_{\lambda=0} + a_2' t^{-3/2} - \mathrm{i}b_2' t^{-3}, \tag{335}$$

Therefore, accounting for the $q_\alpha^\dagger$ part as well, we have that

$$\partial_\lambda \frac{\langle \Omega | e^{\lambda/2 \tilde{Q}} q_\alpha^{\text{p}}(0, t) e^{\lambda/2 \tilde{Q}} | \Omega \rangle}{\langle \Omega | e^{\lambda \tilde{Q}} | \Omega \rangle} |_{\lambda=0} = \partial_\lambda \langle q_\alpha^{\text{p}} \rangle_{\rho_{\text{bpp}}} |_{\lambda=0} + \mathcal{O}(t^{-\alpha_{\text{p}}})$$
$$\partial_\lambda \frac{\langle \Omega | q_\alpha^{\text{p}}(0, t) e^{\lambda \tilde{Q}} | \Omega \rangle}{\langle \Omega | e^{\lambda \tilde{Q}} | \Omega \rangle} |_{\lambda=0} = \partial_\lambda \langle q_\alpha^{\text{p}} \rangle_{\rho_{\text{bpp}}} |_{\lambda=0} + \mathcal{O}(t^{-\alpha_{\text{p}}}), \tag{336}$$

where p= e or o, refers to the parity even and odd self-adjoint densities of the quasi-local charges, and $\alpha_{\text{e}} = 3/2$, and $\alpha_{\text{o}} = 3$. For the second order, similarly, we can write that

$$\partial_\lambda^2 \frac{\langle \Omega | e^{\lambda/2 \tilde{Q}} q_\alpha^{\text{p}}(0, t) e^{\lambda/2 \tilde{Q}} | \Omega \rangle}{\langle \Omega | e^{\lambda \tilde{Q}} | \Omega \rangle} |_{\lambda=0} = \partial_\lambda^2 \langle q_\alpha^{\text{p}} \rangle_{\rho_{\text{bpp}}} |_{\lambda=0} + \mathcal{O}(t^{-\alpha_{\text{p}}})$$
$$\partial_\lambda^2 \frac{\langle \Omega | q_\alpha^{\text{p}}(0, t) e^{\lambda \tilde{Q}} | \Omega \rangle}{\langle \Omega | e^{\lambda \tilde{Q}} | \Omega \rangle} |_{\lambda=0} = \partial_\lambda^2 \langle q_\alpha^{\text{p}} \rangle_{\rho_{\text{bpp}}} |_{\lambda=0} + \mathcal{O}(t^{-\alpha_{\text{p}}}). \tag{337}$$

## E  Bogolyubov transformations

In this appendix we elaborate more on the microscopic treatment of the inhomogeneous quench $\langle \Omega | e^{\lambda/2 Q|_0^\infty(0)} O(x, t) e^{\lambda/2 Q|_0^\infty(0)} | \Omega \rangle^c$ via an appropriate inhomogeneous Bogolyubov transformation and show where the approach breaks down. To ease the notation, we shall focus on the particle number and denote $Q|_0^\infty(0)$ by $\tilde{Q}$.

Let us start by specifying the inhomogeneous microscopic initial state and the Bogolyubov transformation which makes it formally tractable. We have canonical fermion operators $a_k$ and $a_p^\dagger$ and their linear combinations $\bar{a}_k$ and $\bar{a}_p^\dagger$ which annihilates the homogeneous squeezed-coherent

initial state $|\Omega\rangle$ and its adjoint $\langle\Omega|$, respectively. This can be phrased as

$$\bar{a}_k|\Omega\rangle = \frac{1}{\sqrt{2\pi}}\int_{-\infty}^{\infty}\mathrm{d}x\,e^{-\mathrm{i}kx}\left[u_k\psi(x)-v_k\psi^{\dagger}(x)\right]|\Omega\rangle = 0,\text{ for }\forall p\,,$$

$$\langle\Omega|\bar{a}_k^{\dagger} = \frac{1}{\sqrt{2\pi}}\int_{-\infty}^{\infty}\mathrm{d}x\,e^{\mathrm{i}kx}\langle\Omega|\left[u_k\psi^{\dagger}(x)-v_k^*\psi(x)\right] = 0,\text{ for }\forall p\,. \tag{338}$$

The modified, inhomogeneous state is

$$|\tilde{\Omega}\rangle = e^{\frac{\lambda}{2}\tilde{Q}}|\Omega\rangle\,,\quad\text{and}\quad\langle\tilde{\Omega}| = \langle\Omega|e^{\frac{\lambda}{2}\tilde{Q}} = \left(e^{\frac{\lambda}{2}\tilde{Q}}|\Omega\rangle\right)^{\dagger} \tag{339}$$

and we can obtain another set of operators $\tilde{a}_k$ and $\tilde{a}_p'$, via the similarity transformation. We have $a_k,a_k^{\dagger}$ diagonalizing the 'initial Hamiltonian' (in terms of those, the homogeneous vacuum $|\Omega\rangle$ is a squeezed state). Then we have $\bar{a}_k,\bar{a}_k^{\dagger}$ which annihilate $|\Omega\rangle$. Then we have a third set of operators, $\tilde{a}_k,\tilde{a}_k'$ which annihilate the inhomogeneous vacuum $|\tilde{\Omega}\rangle$]

$$\tilde{a}_k = e^{\frac{\lambda}{2}\tilde{Q}}\bar{a}_k e^{-\frac{\lambda}{2}\tilde{Q}}\,,\quad\text{and}\quad\tilde{a}_k' = e^{-\frac{\lambda}{2}\tilde{Q}}\bar{a}_k^{\dagger}e^{\frac{\lambda}{2}\tilde{Q}}\,, \tag{340}$$

which satisfy

$$\tilde{a}_k|\tilde{\Omega}\rangle = e^{\frac{\lambda}{2}\tilde{Q}}\bar{a}_k e^{-\frac{\lambda}{2}\tilde{Q}}e^{\frac{\lambda}{2}\tilde{Q}}|\Omega\rangle = e^{\frac{\lambda}{2}\tilde{Q}}\bar{a}_k|\Omega\rangle = 0$$

$$\langle\tilde{\Omega}|\tilde{a}_k' = \langle\Omega|e^{\lambda/2\tilde{Q}}e^{-\lambda/2\tilde{Q}}\bar{a}_k^{\dagger}e^{\lambda/2\tilde{Q}} = \langle\Omega|\bar{a}_k^{\dagger}e^{\lambda/2\tilde{Q}} = 0\,. \tag{341}$$

Importantly, the operators $\tilde{a}_k$ and $\tilde{a}_p'$ are not canonical. Although $\tilde{a}_k' = \tilde{a}_k^{\dagger}$ and $\{\tilde{a}_p,\tilde{a}_k\} = 0$ and $\{\tilde{a}_p',\tilde{a}_k'\} = 0$, $\{\tilde{a}_p,\tilde{a}_k'\}\neq\delta(p-k)$, which is easy to see and we demonstrate shortly. However, via the algebra these operators define, they facilitate, in principle, a completely legitimate way of computing correlation functions and, additionally, they can be expressed in a particularly compact way. Using that

$$e^{\frac{\lambda}{2}\tilde{Q}}O_k e^{-\frac{\lambda}{2}\tilde{Q}} = O + \frac{\lambda}{2}[\tilde{Q},O] + \frac{1}{2!}\left(\frac{\lambda}{2}\right)^2[\tilde{Q},[\tilde{Q},O]] + \dots \tag{342}$$

and that

$$[\tilde{Q},\psi(x)] = -\Theta_{\mathrm{H}}(x)\psi(x),\quad[\tilde{Q},\psi^{\dagger}(x)] = \Theta_{\mathrm{H}}(x)\psi^{\dagger}(x)\,, \tag{343}$$

we end up with

$$\tilde{a}_k = \frac{1}{\sqrt{2\pi}}\int_{-\infty}^{\infty}\mathrm{d}x\,e^{-\mathrm{i}kx}\left[u_k(e^{-\lambda/2}\Theta_{\mathrm{H}}(x)+\Theta_{\mathrm{H}}(-x))\psi(x)-v_k(e^{\lambda/2}\Theta_{\mathrm{H}}(x)+\Theta_{\mathrm{H}}(-x))\psi^{\dagger}(x)\right]$$

$$\tilde{a}_k' = \tilde{a}_k^{\dagger} = \frac{1}{\sqrt{2\pi}}\int_{-\infty}^{\infty}\mathrm{d}x\,e^{\mathrm{i}kx}\left[u_k(e^{-\lambda/2}\Theta_{\mathrm{H}}(x)+\Theta_{\mathrm{H}}(-x))\psi^{\dagger}(x)-v_k^*(e^{\lambda/2}\Theta_{\mathrm{H}}(x)+\Theta_{\mathrm{H}}(-x))\psi(x)\right]. \tag{344}$$

We now demonstrate the transformation is not canonical. It is easy to see that

$$\{\tilde{a}_k,\tilde{a}_p\} = -\delta(k+p)(u_k v_{-k}+v_k u_{-k}) = 0\,, \tag{345}$$

due to the properties of $u$ and $v$, the former being even and the latter odd. For $\{\tilde{a}_k,\tilde{a}_p^{\dagger}\}$, we have that

$$\{\tilde{a}_k,\tilde{a}_p^{\dagger}\} = \frac{1}{2\pi}\int_{-\infty}^{\infty}\mathrm{d}x\,e^{-\mathrm{i}(k-p)x}u_k u_p(e^{-\lambda}\Theta_{\mathrm{H}}(x)+\Theta_{\mathrm{H}}(-x))+v_k v_p^*(e^{\lambda}\Theta_{\mathrm{H}}(x)+\Theta_{\mathrm{H}}(-x))$$

$$= \frac{1}{2\pi}\left(u_k u_p e^{-\lambda}+v_k v_p^* e^{\lambda}\right)\left(\pi\delta(k-p)-\mathrm{i}\mathrm{P}\frac{1}{k-p}\right) \tag{346}$$

$$+ \frac{1}{2\pi}\left(u_k u_p+v_k v_p^*\right)\left(\pi\delta(k-p)+\mathrm{i}\mathrm{P}\frac{1}{k-p}\right),$$

that is the transformation is not canonical unless $\lambda = 0$.

The essential question is whether we can invert the transformation in order to express the original operators $a_k$ and $a_p^\dagger$ in terms of $\tilde{a}_k$ and $\tilde{a}_p'$, whose algebra we know and which annihilate the inhomogeneous initial state $|\tilde{\Omega}\rangle$ and $\langle\tilde{\Omega}|$. Introducing the field operators $\tilde{\psi}(x)$ and $\tilde{\psi}'(x)$ corresponding to $\tilde{a}_k$ and $\tilde{a}_k'$ satisfying $\{\tilde{\psi}(x), \tilde{\psi}(x')\} = \{\tilde{\psi}'(x), \tilde{\psi}'(x')\} = 0$ and $\{\tilde{\psi}(x), \tilde{\psi}'(x')\} \neq \delta(x - x')$, and

$$U(x) = \frac{1}{2\pi}\int_{-\infty}^{\infty} dk\, e^{ikx} u_k, \quad \text{and} \quad V(x) = \frac{1}{2\pi}\int_{-\infty}^{\infty} dk\, e^{ikx} v_k, \tag{347}$$

we can use the real space representation of the transformation (346) (where $U(x - x')$ contains a $\delta(x - x')$ piece as well). This can be written as

$$\begin{bmatrix} \tilde{\psi}(x) \\ \tilde{\psi}'(x) \end{bmatrix} = \int_{-\infty}^{\infty} dx \begin{bmatrix} U(x-x')\Theta_2(x') & -V(x-x')\Theta_1(x') \\ -V^*(x-x')\Theta_1(x') & U(x-x')\Theta_2(x') \end{bmatrix} \begin{bmatrix} \psi(x') \\ \psi^\dagger(x') \end{bmatrix}, \tag{348}$$

with

$$\begin{aligned} \Theta_1(x) &= e^{-\lambda/2}\Theta_H(x) + \Theta_H(-x) \\ \Theta_2(x) &= e^{\lambda/2}\Theta_H(x) + \Theta_H(-x), \end{aligned} \tag{349}$$

which we further rewrite as

$$\begin{bmatrix} \tilde{\psi}(x) \\ \tilde{\psi}'(x) \end{bmatrix} = \int_{-\infty}^{\infty} dx \begin{bmatrix} U_2(x-x') & -V_1(x-x') \\ -V_1^*(x-x') & U_2(x-x') \end{bmatrix} \begin{bmatrix} \psi(x') \\ \psi^\dagger(x') \end{bmatrix} \tag{350}$$

using the notation, for simplicity,

$$U_2(x-x') = U(x-x')\Theta_2(x'), \quad \text{and} \quad V_1(x-x') = V(x-x')\Theta_1(x'). \tag{351}$$

Inverting the integral kernel in (348) is not obvious due to the presence and the particular alignment of the $\Theta_1$ and $\Theta_2$ functions, but can be carried out, in principle, using known formulas for inverting $2x2$ block matrices. These formulas dictate that the inverse transformation is given by the kernel

$$\begin{bmatrix} U_2^{-1} + U_2^{-1}*V_1*\left(U_2 - V_1^**U_2^{-1}*V_1\right)^{-1}*V_1^**U_2^{-1} & -U_2^{-1}*V_1*\left(U_2 - V_1^**U_2^{-1}*V_1\right)^{-1} \\ -\left(U_2 - V_1^**U_2^{-1}*V_1\right)^{-1}*V_1^**U_2^{-1} & \left(U_2 - V_1^**U_2^{-1}*V_1\right)^{-1} \end{bmatrix}, \tag{352}$$

where each star denotes a convolution. Whereas the inversion of $U$ or $U_2$ is easy via Fourier transform, namely

$$U_2^{-1}(x-x') = \Theta_1(x)\mathscr{F}t^{-1}[u_k^{-1}](x-x'), \tag{353}$$

(where $\mathscr{F}t^{-1}$ denotes the inverse Fourier transform and $V$ is not invertible, but its inverse is not needed), inverting $\left(U_2 - V_1^**U_2^{-1}*V_1\right)$ or, equivalently,

$$\left(1 - U_2^{-1}*V_1^**U_2^{-1}*V_1\right) := (1 - \mathscr{K}) \tag{354}$$

is non-trivial. In particular, this step is not known how to carry out in real space, and in Fourier space one faces the following difficulty. In real space, $\mathscr{K} = U_2^{-1}*V_1^**U_2^{-1}*V_1$ consists of both convolutions and multiplication of functions (due to the $\Theta_{1/2}$ factors, which do not cancel), hence after Fourier transform $\mathscr{K}$ can only be expressed again in terms of both multiplications and convolutions. For the inversion of $(1 - \mathscr{K})$ in Fourier space, $\mathscr{K}$ should be expressed solely in terms of multiplication of functions, which is not the case. Therefore, although $(1 - \mathscr{K})$ is invertible (unlike $V_2$ itself), $(1 - \mathscr{K})^{-1}$ can only be obtained by a systematic expansion.

In summary, despite the linear nature of the bipartite microscopic quench problem, we are not aware of any mathematical method which could provide closed-form predictions.

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
