# Peer review of "A Hydrodynamic Theory for Non-Equilibrium Full Counting Statistics in One-Dimensional Quantum Systems"

_SciPost Physics_

## Round 1 · Referee Report · Anonymous (Referee 1) · 2025-10-17

Disclosure of Generative AI use

The referee discloses that the following generative AI tools have been used in the preparation of this report:

formulating a text from an initial bullet point list of topics and thoughts. This text was then re-edited and the basis of the written report .

Strengths

The manuscript presents a conceptually ambitious and technically competent attempt to generalize hydrodynamic approaches to include the full counting statistics (FCS) of transported quantities in one-dimensional quantum systems far from equilibrium. The authors build on generalised hydrodynamics to construct a unified description that goes beyond average currents and incorporates the complete distribution of fluctuations, thereby addressing a central challenge in non-equilibrium many-body physics. The derivations are internally consistent and well grounded in conservation laws and local-equilibrium assumptions, and the results are formulated compactly in terms of large-deviation functionals that can, in principle, be applied to a broad class of integrable models.

The topic is timely and of significant interest to the community, as FCS and higher-order fluctuations provide the bridge between microscopic quantum correlations and emergent hydrodynamic behavior. The framework complements recent developments in ballistic fluctuation theory and stochastic GHD, and, if further elaborated with explicit examples, it could provide a useful theoretical tool for connecting hydrodynamic descriptions with experimentally measurable fluctuations in 1D quantum gases and spin chains.

Weaknesses

Link to previous work on FCS, both theory and experiment: While the manuscript develops an elegant hydrodynamic framework for non-equilibrium full counting statistics, it omits the seminal theoretical and experimental foundations of FCS, and the more general notion of Full Distribution Functions (FDF) associated with FCS, in 1D Bose systems. The concept of full distribution functions as a probe of cold atom experiments in time of flight to study many-body correlations was introduced by Demler and collaborators (Altman–Demler–Lukin, PRA 2004). And then later generalized for 1d systems and both density and phase: (Polkovnikov–Altman–Demler, PNAS 2006; Gritsev et al., Nat. Phys. 2006). The latter were experimentally realized in interference measurements of split 1D Bose gases for equilibrium: (Hofferberth et al., Nat. Phys. 2008) and non-equilibrium (Gring et al., Science 2012; Langen et al., Science 2015). These experiments directly accessed the full counting statistics (and the FDF) of phase fluctuations and for example revealed prethermalization as the emergent non-equilibrium steady state, and a GGE for its associated statistical ensemble. Schweigler et al., Nature 2017 then connected FCS and FDF to correlation functions and if they factorize or not, for a strongly correlated 1D quantum system. Since the present work aims to describe precisely such non-equilibrium FCS within a hydrodynamic framework, it should explicitly acknowledge this lineage and discuss how its results relate to those experimentally established full distribution functions.

Physical transparency and intuition: The hydrodynamic formalism is mathematically dense, and the physical mechanism linking microscopic correlations to higher cumulants remains opaque. A more intuitive discussion—ideally connecting to measurable observables such as noise, contrast, or phase fluctuations—would help. The assumptions of local equilibrium and Gaussian noise should be more clearly stated, with a discussion of their limitations in strongly interacting or non-integrable systems.

Explicit examples: The theory would benefit from a worked-out case (e.g., Lieb–Liniger gas, XXZ chain) showing how the hydrodynamic cumulant generating function reproduces known FCS results or predicts new features.

Early work by E. Demler and company:

  • Altman, Demler & Lukin, Phys. Rev. A 70, 013603 (2004)

“Probing many-body states of ultracold atoms via noise correlations.” → Introduced the idea that shot-to-shot density-fluctuation correlations in time-of-flight images encode the full counting statistics of the underlying quantum fields.

  • Polkovnikov, Altman & Demler, PNAS 103, 6125 (2006)

“Interference between independent fluctuating condensates.” → Explicitly connected interference fringe contrast distributions to the full counting statistics of relative-phase fluctuations—laying the foundation for later experiments.

  • Gritsev, Altman, Demler & Polkovnikov, Nature Phys. 2, 705 (2006)

“Full quantum distribution of contrast in interference experiments between interacting 1D Bose liquids.” → This paper presented the first theoretical prediction of full distribution functions (FDFs) of interference contrast, identifying the universal connection to 1D field correlations and Luttinger-liquid theory.

Experimental work directly using FCS/FDF to extract equilibrium and out of equilibrium physics:

  • Hofferberth et al., Nature Phys. 4, 489 (2008) – first measurement of FDFs of interference contrast.

  • Gring et al., Science 348, 207 (2015) –used FDF analysis to observe prethermalization — the emergence of quasi-stationary non-thermal states after a quench (splitting a 1D Bose gas)

  • Langen et al., Science 348, 207 (2015) – observation of the formation of a GGE.

  • Schweigler et al., Nature 545, 323 (2017) – complete reconstruction of full distribution functions and higher-order correlation functions in a 1D Bose gas, giving the first direct experimental realisation of FCS in a strongly correlated and topological quantum many-body system.

Those experiments empirically verified precisely the type of non-equilibrium FCS this SciPost manuscript seeks to describe hydrodynamically.

Report

The manuscript definitively warrants publication in SciPost Physics and will be a very valuable asset for studying 1d systems in and out of equilibrium. Before being accepted the authors should consider the above deficiencies, especially the reference to the earlier work on FCS/FDF works of Demler and collaborators and the corresponding experimental realizations and prethermalization studies.

Requested changes

  1. Explicitly cite and discuss the early FCS/FDF works of Demler and collaborators and the corresponding experimental realisations and prethermalization studies. 2. Discuss how the results of this manuscript relate to those former theoretical work and the experimentally established full distribution functions.
  2. Provide one explicit example or application demonstrating the predictive power of the formalism.

Recommendation

Ask for major revision

---

## Round 1 · Referee Report · Anonymous (Referee 2) · 2025-10-28

Report

This work belongs to the body of literature of generalized hydrodynamics, which aims at characterising the large-scale and late-time behavior of integrable systems. The authors study the dynamics of charge fluctuations after homogeneous quantum quenches in integrable one-dimensional systems. They derive an expression for the associated full counting statistics in the limit of short (but still hydrodynamic) times . They discuss the possibility to simplify the formula under certain assumptions. For certain integrable models and particular initial states, the derived formula specialises to a conjectured formula derived in the context of integrable spin chains by means of space-time dualities. The authors finally provide explicit calculations and derivations in the case of free fermions.

The problem of computing full counting statistics is very hard and is a long-standing problem in integrable systems. Therefore, the paper tackles an important question. However, to the best of my judgement, I think this paper belongs to a more specialized literature and so it would be better suited for a submission in Scipost Physics Core. There are a few reasons for this:

  • First, the style of the paper is really that of a technical, specialized work. In fact, I must say that I found the draft very heavy and difficult to process, even if I am familiar with some of the techniques used by the authors.

  • Second, I think the results are nice, but they feel incremental. By this, I do not intend to diminish the value of this work: I think it presents very thorough mathematica proofs and justifications for the derived formula. The latter, however, was already present in some form in the literature (as stated by the authors), at least in some cases. In addition, the present submission focuses mainly on non-interacting systems, where it is less surprising that fully rigorous derivations can be performed (and where numerical computations are efficient)

  • To clarify my point above, I think the paper is missing the study of some interacting case, where they can make predictions that were not possible using previous formulas (even if only conjectured). In this case, it would be important to substantiate the predictions numerically.

For these reasons, I would recommend submission in Scipost Physics Core.

Recommendation

Accept in alternative Journal (see Report)

---

## Round 1 · Referee Report · Anonymous (Referee 3) · 2025-11-17

Strengths

1-interesting hydrodynamic derivation of results
2-rigorous analytical checks of the results

Weaknesses

1-no numerical check in lattice models

Report

The authors study the charge fluctuations, and the full-counting statistics, after homogeneous quantum quenches in one-dimensional systems that show balistic quantum transport. At short times they derive by hydrodynamic arguments a formula for the full-counting statistics. It is interesting that this formula links the fluctuations of the charge to those of the current after an inhomogeneous quench. The latter can be expressed in some cases by using ballistic fluctuation theory. The authors state the result at the very beginning of the paper together with the three main assumptions that they employ to derive it. The three conditions under which the result holds are initial state clustering, local relaxation in the partitioning protocol, time-clustering. The hydrodynamic derivation is reported in section 2. Section 3 shows an ab initio derivation in a system of relativistic fermions. A crucial aspect that is investigated in the paper is the effect of long-range correlations that are know to develop after the quench, and that can spoil the usual hydrodynamic paradigm of independent mesoscopic cells.

The main result of the paper is quite interesting and in my opinion meets the high-quality standards of Scipost Physics. Although the result is quite technical, I think that the authors made an effort to make it understandable.

I have a few questions and suggestions:

1) Below Eq (10) is it correct Eq. (19)?

2) Below (182) there is a typo "equals".

3) The authors cite [20,21] as inspiration for some of the results. A similar approach based on the use of space-time duality was used in Phys. Rev. X 12, 031016 (2022) to derive the dynamics of the Renyi entropies in integrable systems. The authors could mention that.

4) Long range correlations other than the pair ones can develop in integrable systems. Is it possible to generalize the picture to those situations, for instance to the situation in which the quench produces entangled multiplets?

5) Is there a physics reason why the authors considered relativistic fermions, and not lattice models, such as the XX chain?

Recommendation

Publish (easily meets expectations and criteria for this Journal; among top 50%)

---

## Editorial Decision

awaiting_resubmission